# OPTIMIZING OPTIMIZERS FOR FAST GRADIENT-BASED LEARNING

## ABSTRACT

We lay the theoretical foundation for automating optimizer design in gradient-based learning. Based on the greedy principle, we formulate the problem of designing optimizers as maximizing the instantaneous decrease in loss. By treating an optimizer as a function that translates loss gradient signals into parameter motions, the problem reduces to a family of convex optimization problems over the space of optimizers. Solving these problems under various constraints not only recovers a wide range of popular optimizers as closed-form solutions, but also produces the optimal hyperparameters of these optimizers with respect to the problems at hand. This enables a systematic approach to design optimizers and tune their hyperparameters according to the gradient statistics that are collected during the training process. Furthermore, this optimization of optimization can be performed dynamically during training.

## 1 INTRODUCTION

We are interested in the problem of designing optimizers that maximize the utility of gradient-based learning for a given task. The objective of gradient-based learning is to minimize an expected scalar loss $\mathbb{E}[\mathcal{L}(\theta)]$ with respect to parameters $\theta \in \mathbb{R}^d$ using its (negative) gradient $g = -\nabla_\theta \mathcal{L} \in \mathbb{R}^d$. As learning takes time, all the parameters $\theta = \theta(t)$, the loss $\mathcal{L} = \mathcal{L}(\theta(t))$, and the gradients $g = g(t)$ are signals of time $t$, i.e., the training step. The process of learning manifests as the parameter *motion* $\dot\theta$ driven by the gradient *force* $g$ applied at each step $t$.

Physics requires a constitutive law that relates kinematic motion to its motive force. In gradient-based learning, optimizers take that role. We can represent an optimizer as a positive semidefinite operator $Q \succeq 0$ that linearly translates the gradients into the parameter updates,

$$\dot\theta = Q\, g. \tag{1}$$

Later sections will reveal that many existing optimizers fall into this category. By the chain rule, the instantaneous loss drop is a quadratic form:

$$-\dot{\mathcal{L}} = \nabla_\theta \mathcal{L}^\top \frac{\mathrm{d}\theta}{\mathrm{d}t} = g^\top \dot\theta = g^\top Q\, g. \tag{2}$$

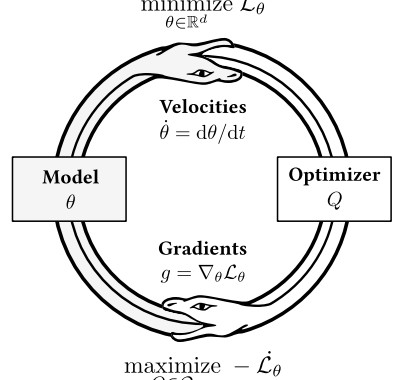

Figure 1: Just as optimizers train their models by feeding them parameter velocities $\dot\theta$, models can also fit the optimizers to the underlying tasks by feeding gradients $g$.

Adhering to the greedy paradigm, we turn our original problem of maximizing the utility of learning into a different optimization problem that maximizes this loss drop with respect to the optimizer $Q$:

$$\operatorname*{maximize}_{Q \in \mathcal{Q}} \mathbb{E}[g^\top Q\, g], \tag{P1}$$

where $\mathcal{Q} \subseteq \mathbb{S}_+^d$ is the design space of allowed optimizers.

Instantaneously, we notice that without any additional constraint, the maximum of the quadratic form $g^\top Q\, g$ is unbounded. Problem P1 reveals two design options that bound this maximum: (1)

the *trust region* implied by the feasible set $Q \in \mathcal{Q}$, and (2) the *gradient distribution* under the expectation $\mathbb{E}$. Our main focus is on how these two factors determine the *optimal optimizer* $Q^\star$.

Placing the optimizer itself as a subject of another optimization is interesting in several ways:

- Optimizers can be *systematically* designed with respect to individual problems (task and data), greatly reducing the need for tedious manual tuning of optimizer hyperparameters in practice.
- Optimizers and their hyperparameters can be *dynamically* tuned or even be replaced by better ones according to the intermediate probes from the gradients in the middle of training.
- By *reverse engineering* commonly used optimizers, we draw the landscape of optimizers that have driven the success of machine learning (Robbins & Monro, 1951; Kingma & Ba, 2015; Loshchilov & Hutter, 2019; Gupta et al., 2018; Martens & Grosse, 2015) into a single picture.
- Our unified framework uncovers the *underlying design principles* of those optimizers. This lets us better use the well-studied optimizers in practice and also suggest extensions to them.

## 2  OPTIMAL STATELESS OPTIMIZERS

Consider the following setup: Given a data distribution $\pi$, training samples $x \sim \pi$ produce the gradients $g = \nabla_\theta \mathcal{L}(\theta, x)$. The *gradient moment* is defined as:

$$\Sigma := \mathbb{E}[g\, g^\top], \tag{3}$$

where $\mathbb{E}$ denotes the expectation over the gradient distribution. Note that $\Sigma$ is a symmetric and positive semidefinite (PSD) matrix of shape $d \times d$. For any symmetric PSD optimizer $Q \in \mathbb{S}_+^d$ of shape $d \times d$, let us define its *learning power* as an expected quadratic form of the gradients:

$$P(Q) := \mathbb{E}[g^\top Q\, g] = \mathrm{Tr}(Q\, \Sigma) = \langle Q, \Sigma \rangle_{\mathrm{F}}. \tag{4}$$

From the chain rule of equation 2, the learning power is equal to the expected rate of loss drop: $-\mathbb{E}[\dot{\mathcal{L}}] = \mathbb{E}[g^\top \dot{\theta}] = P(Q)$. Problem P1 then becomes:

$$\operatorname*{maximize}_{Q \in \mathcal{Q}} P(Q) = \mathrm{Tr}(Q\, \Sigma), \tag{P2}$$

This is our main optimization problem.

Solving this without any additional constraint ends up with arbitrarily large eigenvalues of $Q$, which corresponds to arbitrarily large learning rates. This is certainly infeasible in practice. Real problems give us several reasons that make this "ideal solution" unrealizable: finite precision of our machines, curvature of non-convex loss landscapes, stochastic nature of subset gradients, etc. All of them restrict the ability of gradient estimates $g$ to represent the global geometry of the loss landscape over the parameter space. Taking a large step in the parameter space beyond the regions where $g$ remains explainable leads to unexpected, and usually fatal, behaviors.

This calls for additional constraints on the optimizer $Q$. AdaReg (Gupta et al., 2017) considers similar problem as P2 but with an indirect regularization term $\Phi(Q)$ on $Q$. Instead, we allow the engineer to choose the *trust region* $\mathcal{Q} \subseteq \mathbb{S}_+^d$ that circumscribes the feasible set directly, leading to a family of exact solutions. The following theorem makes this precise:

**Theorem 1** (Optimal stateless optimizers under convex constraints). [`proof`] *Let the* trust region $\{0\} \subseteq \mathcal{Q} \subseteq \mathbb{S}_+^d$ *be a nonempty, compact, convex set. Define (1) its* indicator $\delta_{\mathcal{Q}}(Q) = 0$ *if* $Q \in \mathcal{Q}$ *and* $+\infty$ *otherwise, (2) its* gauge *(Minkowski functional):* $\gamma_{\mathcal{Q}}(Q) = \inf\{\lambda > 0 : Q \in \lambda \mathcal{Q}\}$, *and (3) its* polar set $\mathcal{Q}^\circ = \{\Sigma \in \mathbb{S}^d : \sup_{Q \in \mathcal{Q}} \mathrm{Tr}(Q\Sigma) \leq 1\}$. *For any symmetric matrix* $\Sigma \in \mathbb{S}^d$,

*(i)* (Existence and sublinearity): *The maximum of* P2 *is attained:* $P^\star(\Sigma) := \max_{Q \in \mathcal{Q}} \mathrm{Tr}(Q\Sigma)$. *Furthermore,* $P^\star$ *is sublinear (convex and positively homogeneous) and finite everywhere.*

*(ii)* (Conjugacy identities): *The maximum* $P^\star(\Sigma)$ *satisfies the following identity:*

$$P^\star = \delta_{\mathcal{Q}}^* = \gamma_{\mathcal{Q}^\circ} \qquad \text{and} \qquad \gamma_{\mathcal{Q}}^* = \delta_{\mathcal{Q}^\circ}, \tag{5}$$

*i.e., the* optimal learning power *is the convex conjugate of the indicator and also the gauge of the polar, while the conjugate of the gauge is the indicator of the polar.*

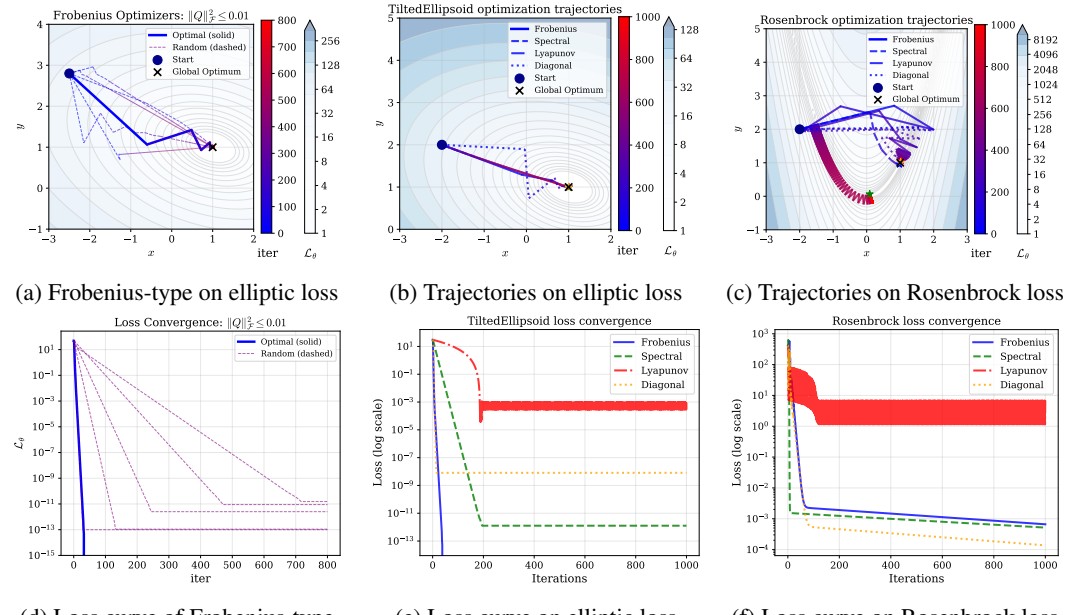

(a) Frobenius-type on elliptic loss (b) Trajectories on elliptic loss (c) Trajectories on Rosenbrock loss

(d) Loss curve of Frobenius-type (e) Loss curve on elliptic loss (f) Loss curve on Rosenbrock loss

Figure 2: Behavior of optimal optimizers under different types of trust regions. (a, d) Dotted lines are suboptimal optimizers with random $\Sigma$ in an equal-power Frobenius family; the straight line shows the optimal optimizer found by our theory, achieving fastest convergence. (b, c, e, f) No free lunch theorem: Frobenius family excels for simple elliptic losses, while spectral and diagonal families excel for nonconvex loss geometries. Each line indicates the best result from dense search among all trust region parameters, e.g, $B$ for Frobenius family, etc.

*(iii)* (Construction): *An optimal optimizer $Q^\star \in \arg\max_{Q \in \mathcal{Q}} \mathrm{Tr}(Q\Sigma)$ is a subgradient of $P^\star$ at $\Sigma$: $Q^\star \in \partial_\Sigma P^\star(\Sigma)$. If the maximizer is unique, $P^\star$ is differentiable at $\Sigma$ and $Q^\star = \nabla_\Sigma P^\star(\Sigma)$.*

*(iv)* (Order preservation on $\mathbb{S}_+^d$): *If $\Sigma_1 \succeq \Sigma_2$, then $P^\star(\Sigma_1) \geq P^\star(\Sigma_2)$. If $\Sigma \succeq 0$, then $P^\star(\Sigma) \geq 0$.*

*(v)* (Lipschitz continuity in symmetrized polar gauge): *Define $\|\cdot\|_{\mathcal{Q}^\circ}^{\mathrm{sym}} := \max\{\gamma_{\mathcal{Q}^\circ}(\cdot), \gamma_{\mathcal{Q}^\circ}(-\cdot)\}$. Then, for any $\Sigma, \hat{\Sigma} \in \mathbb{S}^d$, $|P^\star(\Sigma) - P^\star(\hat{\Sigma})| \leq \|\Sigma - \hat{\Sigma}\|_{\mathcal{Q}^\circ}^{\mathrm{sym}}$. Hence, the optimal learning power is Lipschitz-bounded by the difference between the two data-induced moments.*

The proof is in Appendix H. From items (i), (ii), and (iii), we have a principled way to construct the optimal (stateless) optimizer $Q^\star$ from any given gradient statistics $\Sigma$ and any nicely conditioned feasible set $\mathcal{Q}$. In practice, full gradients rarely appear, and stochastic gradients drift throughout non-convex loss landscapes, making the moments $\Sigma$ themselves drift as well. Items (iv) and (v) then provide a bound on the estimation error of the optimal learning power from this perturbation.

Theorem 1 states that setting the family of available trust regions $\mathcal{Q}$ *determines* the optimal optimizer $Q^\star$ and its associated hyperparameters through the solutions to the convex optimization problem P2. In particular, consider the following four types of trust regions:

- *Frobenius ball type* $\mathcal{Q}_{\mathrm{F}}(B) = \{Q \succeq 0 : \|Q\|_{\mathrm{F}}^2 \leq B\}$ is the simplest and the largest family that does not favor any particular direction in the parameter space, but requires larger memory to store its hyperparameters.

- *Spectral type* $\mathcal{Q}_{\mathrm{S}}(\tau, \lambda) = \{Q \succeq 0 : \mathrm{Tr}(Q) \leq \tau, Q \preceq \lambda I\}$ upper limits the (1) per-direction spectrum for safety and the (2) trace for total update budget, simultaneously.

- *Lyapunov type* $\mathcal{Q}_{\mathrm{L}}(B) = \{Q \succeq 0 : \mathrm{Tr}(Q^2\Sigma) \leq B\}$ utilizes the gradient moment $\Sigma$ as the metric, leading to a natural Lyapunov-like stability condition.

- *Diagonal type* $\mathcal{Q}_{\mathrm{D}}(B, c) = \{Q = \mathrm{diag}(q_j) \succeq 0 : \sum_j c_j q_j^2 \leq B\}$ represents element-wise optimizers, a memory-efficient family that are commonly used in large-scale machine learning.

Instantiating the construction from Theorem 1 on each of these families, we obtain the closed-form optimal optimizer $Q^\star$ and the corresponding optimal learning power $P^\star$.

**Corollary 2** (Closed-form solutions for common optimizer families). [`proof`] *Given a PSD moment $\Sigma$, we have its eigendecomposition $\Sigma = U \operatorname{diag}(\sigma_1 \geq \cdots \geq \sigma_d) U^\top$. Then the closed-form solutions are:*

*(i)* (Frobenius ball): $Q_F^\star = \sqrt{B}\, \Sigma/\|\Sigma\|_F$. *This gives* $P_F^\star(\Sigma) = \sqrt{B}\, \|\Sigma\|_F$.

*(ii)* (Spectral): $Q_S^\star = U \operatorname{diag}(q_i^\star) U^\top$, *where (1)* $q_i^\star = \lambda$ *for* $i \leq k$, *(2)* $q_{k+1}^\star = \tau - k\lambda$, *and (3)* $q_i^\star = 0$ *for* $i > k+1$ *with* $k = \lfloor \tau/\lambda \rfloor$. *This gives* $P_S^\star(\Sigma) = \lambda \sum_{i \leq k} \sigma_i + (\tau - k\lambda)\sigma_{k+1}$.

*(iii)* (Lyapunov): $Q_L^\star = \alpha\, \Pi_\Sigma$, *where* $\Pi_\Sigma$ *is the orthogonal projection onto the support of* $\Sigma$, *and* $\alpha = \sqrt{B}\,(\sum_{i:\sigma_i > 0} \sigma_i)^{-1/2}$. *This gives* $P_L^\star(\Sigma) = \sqrt{B}\,(\sum_i \sigma_i)^{1/2}$.

*(iv)* (Diagonal): $[Q_D^\star]_{jj} \propto \sigma_j/c_j$, *where* $\sigma_j$ *is the $j$-th singular value and also the $j$-th diagonal element of* $\Sigma$, *and* $U = I$. *This gives* $P_D^\star(\Sigma) = \sqrt{B}\,(\sum_j \sigma_j^2/c_j)^{1/2}$.

Again, the proof is in Appendix H. These analytic solutions reveal how different types of optimal optimizers $Q^\star$ emerge from the choice of trust region geometry $\mathcal{Q}$. Specifically, we see that:

**Frobenius family $\leftrightarrow$ Proportional optimizers.** These optimal optimizers are *proportional* to the gradient moment $Q^\star \propto \Sigma$. We can further project this general class into special geometries to obtain the optimal hyperparameters for various types of optimizers as in Corollary 5.

**Spectral family $\leftrightarrow$ Water-filling optimizers $\sim$ gradient clipping & LR scheduling.** The spectral trust region $\mathcal{Q}_S(\tau, \lambda)$ results in a *water-filling* optimizer that concentrates the parameter update rates into the largest available principal component of the data moment $\Sigma$ up to a per-mode cap $\lambda$ (similar to gradient clipping), sequentially, until the total budget $\tau$ is reached (similar to learning rate scheduling). This shows how algorithmic tricks can be represented as trust regions.

**Lyapunov family $\leftrightarrow$ Equal-power optimizers $\supset$ {AdaGrad, natural gradient}.** The optimal solutions of this family allocate uniform power across eigendirections of the gradient moment $\Sigma$, whitening the gradient statistics by projecting onto the support of $\Sigma$ with a constant scaling factor. When $\Sigma$ is a Fisher information matrix, this represents *natural gradient descent* (Amari, 1998). Generally, this encompasses full-matrix AdaGrad (Duchi et al., 2011; Agarwal et al., 2019), K-FAC (Martens & Grosse, 2015), and Shampoo (Gupta et al., 2018).

**Diagonal family $\leftrightarrow$ Coordinate-wise optimizers $\supset$ {Adam, GD}.** Corollary 2 shows that these coordinate-wise optimal optimizers set up their weights that scale with the coordinate-wise gradient variance $\sigma_j = \Sigma_{jj}$ and inversely with the pre-defined costs $c_j$, leading to optimal hyperparameters for various types of existing optimizers such as Adam optimizer (Kingma & Ba, 2015) as in Corollary 6. Various types of well-used optimizers are categorized by the choice of costs $c_j$, e.g., $c_j = 1$ gives simple gradient descent, $c_j = \operatorname{EMA}(g_j^2)^{1/2}$ gives diagonal AdaGrad (Duchi et al., 2011) or RMSProp-style optimizer (Tieleman & Hinton, 2012).

The behaviors of different types of optimizers are visualized in Figure 2. Figure 2a and 2d show that our analytically found optimal optimizer is the fastest among all hyperparameter settings under the same Frobenius family. This demonstrates that we can systematically choose the optimal hyperparameters according to the gradient statistics as the theory suggests. On the other hand, Figure 2b, 2c, 2e, and 2f highlight how optimizers from different families can outperform others in their specialized domains. This reminds us of the notorious *no free lunch theorem* in optimization (Wolpert & Macready, 1997): "no single algorithm is universally superior". With our convex optimization framework which associates the best algorithm that matches a given task, we can now update this catchphrase into: "Under the engineer's choice of trust region $\mathcal{Q}$, the optimal optimizer $Q^\star$ is *determined* by the task's gradient statistics $\Sigma$".

## 3 OPTIMAL DYNAMIC OPTIMIZERS WITH STATE VARIABLES

Up to this point, we have focused on the simple stateless optimizers, which are less used in real-world applications. In practice, optimizers have memory, often in the form of momentum, in order to stabilize the learning process from stochastic gradients and non-convex loss landscapes. We now extend our framework by letting the optimizer $Q[n]$ be a *causal dynamical* operator: a *filter* that translates gradient history $g[n]$ into instantaneous parameter velocity $\dot{\theta}[n]$, where $n \in \{0, 1, 2, \ldots\}$

representing (discrete) training steps. Define a *dynamic* optimizer as an LTI filter with a symmetric matrix impulse response $Q[n] \in \mathbb{R}^{d \times d}$. This operates as a causal convolution:

$$\dot{\theta}[n] = (Q * g)[n] := \sum_{k=0}^{\infty} Q[k]\, g[n-k]. \tag{6}$$

We use the Hilbert norm $\langle \cdot, \cdot \rangle_{\mathcal{H}}$ instead of the Frobenius norm $\langle \cdot, \cdot \rangle_F = \mathrm{Tr}(\cdot^\top \cdot)$ we used in Section 2.

$$\|Q\|_{\mathcal{H}}^2 := \sum_{n=0}^{\infty} \mathrm{Tr}(Q[n]^\top Q[n]) < \infty, \qquad \langle Q_1, Q_2 \rangle_{\mathcal{H}} := \sum_{n=0}^{\infty} \mathrm{Tr}(Q_1[n]^\top Q_2[n]). \tag{7}$$

In addition, for theoretical analysis, gradient processes $g[n] \in \mathbb{R}^d$ are assumed to be wide-sense stationary (WSS). The *moments* are then the autocorrelations:

$$R[k] := \mathbb{E}[g[n]\, g[n-k]^\top], \tag{8}$$

for $k \geq 0$. The *instantaneous learning power* is, again, an inner product (details in Appendix G.1):

$$P(Q;n) := \mathbb{E}\big[g[n]^\top \dot{\theta}[n]\big] = \mathbb{E}\left[g[n]^\top \sum_{k=0}^{\infty} Q[k]\, g[n-k]\right] = \sum_{k=0}^{\infty} \mathrm{Tr}\big(Q[k]^\top R[k]\big) = \langle Q, R \rangle_{\mathcal{H}}. \tag{9}$$

Define a nonempty, convex, and weakly compact trust region $\mathcal{Q} \subset \mathcal{H}$. The *indicator* $\delta_{\mathcal{Q}}(Q)$ and the *gauge* $\gamma_{\mathcal{Q}}(Q)$ are defined the same as in Section 2. Also, the *polar set* $(\mathcal{Q})^\circ := \{R \in \mathcal{H} \mid \sup_{Q \in \mathcal{Q}} \langle Q, R \rangle_{\mathcal{H}} \leq 1\}$ is defined likewise. Then, we have every notation the same as in Section 2.

In this dynamic setting, problem P2 is lifted to:

$$\underset{Q \in \mathcal{Q}}{\text{maximize}}\; P(Q) = \langle Q, R \rangle_{\mathcal{H}}. \tag{P3}$$

This has exactly the same form as in Section 2. Unsurprisingly, we arrive at the same results. The only difference is the type of inner product that defines the instantaneous learning power $P(Q)$.

**Theorem 3** (Optimal dynamic optimizers under convex constraints). [`proof`] *Given the definitions above, the following hold for any nonempty, convex, and weakly compact trust region $\mathcal{Q} \subset \mathcal{H}_+$ with $0 \in \mathcal{Q}$:*

*(i)* (Existence and sublinearity): *The maximum of P3 is attained: $P^\star(R) := \max_{Q \in \mathcal{Q}} \langle Q, R \rangle_{\mathcal{H}}$. Furthermore, $P^\star$ is sublinear and finite everywhere.*

*(ii)* (Conjugacy identities): *$P^\star = \delta_{\mathcal{Q}}^* = \gamma_{\mathcal{Q}^\circ}$ and $\gamma_{\mathcal{Q}}^* = \delta_{\mathcal{Q}^\circ}$.*

*(iii)* (Construction): *Any optimal optimizer $Q^\star \in \arg\max_{Q \in \mathcal{Q}} \langle Q, R \rangle_{\mathcal{H}}$ is a subgradient of $P^\star$ at $R$: $Q^\star \in \partial_R P^\star(R)$. If the maximizer is unique, $P^\star$ is differentiable at $R$ and $\nabla_R P^\star(R) = Q^\star$.*

*(iv)* (Order preservation on $\mathcal{H}_+$): *If $R \in \mathcal{H}_+$ (Hermitian PSD a.e.), then $P^\star(R) \geq 0$. Moreover, if $R_1 - R_2 \in \mathcal{H}_+ \setminus \{0\}$ and $\exists Q \in \mathcal{Q}$ with $\langle Q, R_1 - R_2 \rangle_{\mathcal{H}} > 0$ (e.g., if $\mathcal{Q}$ contains a positive definite element), then $P^\star(R_1) > P^\star(R_2)$.*

*(v)* (Lipschitz continuity in the symmetrized polar gauge): *Define the symmetrized polar gauge $\|u\|_{\mathcal{Q}^\circ}^{\mathrm{sym}} := \max\{\gamma_{\mathcal{Q}^\circ}(u), \gamma_{\mathcal{Q}^\circ}(-u)\}$. Then $\forall R, \hat{R} \in \mathcal{H}$, $|P^\star(R) - P^\star(\hat{R})| \leq \|R - \hat{R}\|_{\mathcal{Q}^\circ}^{\mathrm{sym}}$.*

The proof is similar to the stateless case, and is provided in Appendix H like all other proofs for this section. Theorem 3 formalizes how the optimal dynamic optimizer $Q^\star$ *equalizes* the learning power *across different frequencies* as a function of the convex trust region $\mathcal{Q}$. All the closed-form solutions from Corollary 2 can also be directly lifted to the dynamic framework, as elaborated in Appendix G.3. Instead of redundantly repeating these closed-form solutions, we discuss how they are connected to well-used optimizers in practice. As we will see in Corollaries 5 and 6, solving P3 using Theorem 3 often produces general dynamic optimizers with infinite impulse responses (IIR) $Q[n]$, whose implementations require infinite memory for the optimizer states. In practice, we often restrict ourselves to simpler, realizable families of optimizers, e.g., those with EMA-based momenta. The following lemma justifies this *post-projection* of optimal optimizers from general trust regions to more restrictive geometries with fewer controllable hyperparameters.

**Lemma 4** (Conservation of optimality under projection). [`proof`] *Let $\mathcal{H}$ be a real Hilbert space. Given a nonzero moment $R \in \mathcal{H}$, let $\mathcal{Q} \subset \mathcal{H}$ be nonempty, closed, convex, with $0 \in \mathcal{Q}$. Let $\mathcal{C} \subset \mathcal{H}$ be a cone (closed under positive scaling). Then the normal cone of $\mathcal{Q} \cap \mathcal{C}$ at $Q$ is: $N_{\mathcal{Q} \cap \mathcal{C}}(Q) := \{M \in \mathcal{H} : \langle M, Q' - Q \rangle_{\mathcal{H}} \leq 0 \; \forall Q' \in \mathcal{Q} \cap \mathcal{C}\}$. The solution set of the optimization problem and its restriction to $\mathcal{C}$ are:*

$$\mathcal{Q}^\star(R) := \arg \max_{Q \in \mathcal{Q}} \langle Q, R \rangle_{\mathcal{H}}, \qquad \mathcal{Q}_{\mathcal{C}}^\star(R) := \arg \max_{Q \in \mathcal{Q} \cap \mathcal{C}} \langle Q, R \rangle_{\mathcal{H}}. \tag{10}$$

*Let $\Pi_{\mathcal{C}}$ be the Hilbert metric projection onto $\mathcal{C}$. For any $Q^\star \in \mathcal{Q}^\star(R)$, the following are equivalent:*

*(i) (Commutativity) $\Pi_{\mathcal{C}}(Q^\star) \in \mathcal{Q}_{\mathcal{C}}^\star(R)$. That is, projecting the unconstrained optimal solution $Q^\star \in \mathcal{Q}^\star(R)$ onto the cone $\mathcal{C}$ yields the constrained optimal solution $\Pi_{\mathcal{C}}(Q^\star)$.*

*(ii) (Normal-cone alignment) There exists $Q_{\mathcal{C}}^\star \in \mathcal{Q}_{\mathcal{C}}^\star(R)$ such that $\{R, Q^\star - Q_{\mathcal{C}}^\star\} \subset N_{\mathcal{Q} \cap \mathcal{C}}(Q_{\mathcal{C}}^\star)$,*

*Moreover, if $N_{\mathcal{Q} \cap \mathcal{C}}(Q_{\mathcal{C}}^\star)$ is a ray $\{\lambda M : \lambda \geq 0\}$, then commutativity holds if and only if $R$ and $Q^\star - Q_{\mathcal{C}}^\star$ are positive multiples of the same direction $M$.*

The projection cone $\mathcal{C}$ represents a desired property of the set of optimizers. If $\mathcal{C}$ is in a nice shape, e.g., a set of optimizers with a momentum $\mathcal{C} = \mathcal{C}_{1p} := \{Q[n] = \eta(1-\beta)\beta^n I : \eta \geq 0, 0 < \beta < 1\}$, then we can first solve the easier unconstrained problem for general $\mathcal{Q}$ as in Corollary 2 and then project the resulting solutions to the cone $\mathcal{C}$ to obtain the final optimal solution over the target constrained problem $\mathcal{Q} \cap \mathcal{C}$. This extends our framework to many practical optimizers. Now we are ready to find the optimal hyperparameters for real optimizers in use.

**Corollary 5** (Instantaneously optimal SGD+Momentum). [`proof`] *Consider the general family of Frobenius trust regions $\mathcal{Q}_F(B)$ and a cone $\mathcal{C}_{1p}$ of isotropic 1-pole optimizers:*

$$\mathcal{Q}_F(B) := \{Q : \|Q\|_{\mathcal{H}} \leq \sqrt{B}\}, \quad \mathcal{C}_{1p} := \{Q_{\eta,\beta}[n] = \eta(1-\beta)\beta^n I : \eta \geq 0, 0 < \beta < 1\}. \tag{11}$$

*Given gradients $g[n]$, define $m_\beta[n] := \sum_{k=0}^{\infty} \beta^k g[n-k]$ as the unnormalized momentum at time $n$ with momentum parameter $\beta$. Then the optimal solution of problem P3 under the trust region $\mathcal{Q}_F(B) \cap \mathcal{C}_{1p}$ is an SGD+Momentum optimizer with optimal hyperparameters:*

$$\beta^\star[n] = \arg \max_{\beta \in (0,1)} \sqrt{1-\beta^2} \, \mathbb{E}[g[n]^\top m_\beta[n]], \quad \eta^\star = \left( \frac{B(1+\beta^\star)}{d(1-\beta^\star)} \right)^{1/2}, \tag{12}$$

*where $d$ is the dimension of the parameter space.*

The *learning rate* $\eta^\star$ scales to saturate the budget $B$.

Corollary 6 does the same to Adam (Kingma & Ba, 2015). For Adam, the existence of a time-varying divisor $\text{EMA}(g^2, \beta_2)^{-1/2}$ slightly complicates the derivation by making the optimizer time-varying.

**Corollary 6** (Instantaneously optimal Adam/AdamW). [`proof`] *Consider the general family of diagonal trust regions $\mathcal{Q}_D(B, c)$ and a cone $\mathcal{C}_{1p}(c)$ of diagonal 1-pole optimizers of a given cost vector $c$:*

$$\mathcal{Q}_D(B, c) := \{\text{diag}(q_j) : \sum_j c_j \sum_{k \geq 0} |q_j[k]|^2 \leq B\}, \tag{13}$$

$$\mathcal{C}_{1p}(c) := \{Q_{\eta,\beta_1}[n] = \text{diag}(\eta(1-\beta_1)\beta_1^n/c_j) : \eta \geq 0, 0 < \beta_1 < 1\}. \tag{14}$$

*Given gradients $g[n]$, define the first $m$ and second moment $v$ for coordinate $j$ at time $n$ as*

$$m_{\beta_1,j}[n] = \beta_1 m_{\beta_1,j}[n-1] + (1-\beta_1)g_j[n], \quad v_{\beta_2,j}[n] = \beta_2 v_{\beta_2,j}[n-1] + (1-\beta_2)g_j^2[n], \tag{15}$$

*where $\beta_1, \beta_2 \in (0,1)$ are hyperparameters. Define the cost $c_j = v_{\beta_2,j}^{1/2}$. Then the optimal solution of problem P3 under $\mathcal{Q}_D(B, c) \cap \mathcal{C}_{1p}(c)$ is an Adam optimizer with optimal hyperparameters:*

$$(\beta_1^\star[n], \beta_2^\star[n]) = \arg \max_{\beta_1 \in (0,1), \beta_2 \in (0,1)} a(\beta_1, \beta_2) \, \mathbb{E}[g[n]^\top u_{\beta_1, \beta_2}[n]], \quad \eta^\star = \sqrt{B} \, a(\beta_1^\star, \beta_2^\star), \tag{16}$$

*where $u_{\beta_1,\beta_2} := m_{\beta_1}/v_{\beta_2}^{1/2}$ is the Adam update and $a(\beta_1, \beta_2) := \sqrt{(1+\beta_1)/((1-\beta_1)\sum_j(1/c_j))}$ is the normalization factor.*

Table 1: Demonstration of Corollary 5 for SGD with momentum. Best baseline at $\beta = 0.9$. mean $\pm$ std.

| Method | Test acc. % | Train loss |
|--------|-------------|------------|
| Best baseline | $77.57 \pm 0.09$ | $0.0078 \pm 0.0001$ ● |
| **Ours** | $78.06 \pm 0.07$ ● | $0.0080 \pm 0.0001$ |

Table 2: Demonstration of Corollary 6 for Adam. Best baseline at $(\beta_1, \beta_2) = (0.8, 0.999)$. mean $\pm$ std.

| Method | Test acc. % | Train loss |
|--------|-------------|------------|
| Best baseline | $73.20 \pm 0.21$ | $0.0324 \pm 0.0042$ |
| **Ours** | $73.26 \pm 0.31$ ● | $0.0115 \pm 0.0010$ ● |

(a) Comparison of SGD+M optimizers.

(b) Comparison of Adam optimizers.

Figure 3: Demonstration of Corollaries 5 and 6. Our instantiations of optimal optimizers are compared with baselines having fixed hyperparameters on the CIFAR-100 dataset (Krizhevsky, 2009) with ResNet-18 (He et al., 2016), following the standard settings of He et al. (2016). The error bars indicate the mean and standard deviation over 10 runs. Our instantiation shows better performance than every baseline optimizer with fixed hyperparameters, without relying on heavy workload of manual hyperparameter tuning.

Corollaries 5 and 6 show that the optimal hyperparameters for both SGD+Momentum and Adam are the ones that maximize the expected weighted cosine similarity between the gradients and the optimizer responses. The optimal hyperparameters are achieved when the corresponding optimizer *aligns best* with the gradient distribution for each time step $n$.

Furthermore, by deriving the SGD and Adam/AdamW optimizers from our framework, we gain additional insights into these well-known optimizers. For example, Adam/AdamW can be interpreted as an *optimal 1-pole approximation of the dynamic diagonal optimizer with cost $c = v^{1/2}$*. Similarly, we can reverse engineer various other optimizers, including Gauss-Newton, natural gradient descent (Amari, 1998), K-FAC (Martens & Grosse, 2015), Shampoo (Gupta et al., 2018), and Muon (Liu et al., 2025), into our framework, as shown in Table 7 and in Appendix F. This classifies practical optimizers, reveals their hidden design principles, and provides a systematic way to determine optimal hyperparameters for these optimizers.

## 4 EMPIRICAL DEMONSTRATION

One of the practical advantages of our framework is the *automatic* determination of optimal hyperparameters, as shown in Corollaries 5 and 6. The theory clarifies that optimal hyperparameters depend on the gradient distribution, as illustrated in Figure 1. Here, we provide simple instantiations of these theoretical frameworks and demonstrate their validity and practical usefulness. The implementation of the argmax in equations 12 and 16 can vary. However, the key idea is to maximize the cosine similarity *among available options*. The simplest way is to maintain two fixed optimizers with different hyperparameters $\beta^{(1)}$ and $\beta^{(2)}$, compute the argmax operands for each option, and select the optimizer with the largest value *dynamically* to obtain the parameter update.

Using this simple instantiation, we empirically demonstrate the theory by training a ResNet-18 (He et al., 2016) model on the CIFAR-100 dataset (Krizhevsky, 2009). We provide the full algorithm in Appendix C. Baseline optimizers are trained with fixed hyperparameters, following typical machine learning practices. For the momentum of SGD+Momentum, we tried $\beta \in [0.01, 0.999]$ and reported the best one. For the Adam, we tried $\beta_1 \in [0.1, 0.99]$ while keeping $\beta_2 = 0.999$ fixed, and reported

Table 3: AdamW on Gemma-2B with MetaMathQA-395K validated on GSM8K. mean ± std.

| Method | Test acc. % | Train loss |
|---|---|---|
| Best baseline | 52.57 ± 1.10 | 0.2080 ± 0.0004 ● |
| **Ours** | 52.77 ± 0.93 ● | 0.2084 ± 0.0003 |

Table 4: AdamW on Llama-3-8B with MetaMathQA-395K dataset. mean ± std.

| Method | Test acc. % | Train loss |
|---|---|---|
| Best baseline | 76.20 ± 0.33 | 0.1927 ± 0.0005 |
| **Ours** | 76.30 ± 0.31 ● | 0.1925 ± 0.0005 ● |

Table 5: AdamW on Gemma-2B with Commonsense-170K dataset. mean ± std.

| Gemma-2B (LoRA) | BoolQ | PIQA | Social IQA | HellaSwag | Winogrande | OBQA | Avg |
|---|---|---|---|---|---|---|---|
| Best baseline | 65.31 ± 0.27 ● | 78.87 ± 0.67 | 73.66 ± 0.37 ● | 72.97 ± 1.47 | 71.40 ± 0.30 | 73.20 ± 0.65 | 71.99 ± 0.24 |
| **Ours** | 65.31 ± 0.04 ● | 79.00 ± 0.36 ● | 73.58 ± 0.06 | 75.09 ± 1.02 ● | 71.80 ± 0.39 ● | 73.27 ± 1.15 ● | 72.12 ± 0.21 ● |

| Gemma-2B (Full FT) | BoolQ | PIQA | Social IQA | HellaSwag | Winogrande | OBQA | Avg |
|---|---|---|---|---|---|---|---|
| Best baseline | 62.79 ± 0.27 | 74.12 ± 0.26 | 66.63 ± 0.33 | 40.50 ± 1.15 | 61.48 ± 0.32 | 62.60 ± 1.02 | 61.86 ± 0.16 |
| **Ours** | 63.29 ± 0.78 ● | 75.70 ± 0.22 ● | 68.41 ± 0.69 ● | 42.47 ± 1.06 ● | 62.46 ± 4.64 ● | 64.40 ± 0.86 ● | 63.36 ± 0.93 ● |

the best one. Figure 3 and Tables 1 and 2 summarize the results. Our automatic hyperparameter tuning shows comparable and often better performance than the baseline optimizers with fixed hyperparameters in both final accuracy and convergence speed. This demonstrates that tedious manual hyperparameter tuning is unnecessary and can be replaced by our framework.

We also demonstrate our framework in more practical scenarios training large language models (LLMs) and vision transformers (ViTs). We train a Gemma-2B (Gemma Team et al., 2023) and Llama-3-8B (Grattafiori et al., 2024) using low rank adaptation (LoRA) (Hu et al., 2022) with standard settings on the MetaMathQA-395K dataset (Yu et al., 2024a) and compare the results with baseline optimizers of fixed hyperparameters on GSM8K (Cobbe et al., 2021). Table 3 and 4 summarize the results of ten runs for each model. Furthermore, we also train a Gemma-2B with both LoRA and full fine-tuning on the Commonsense-170K dataset (Hu et al., 2023) using ours and baseline optimizers of fixed hyperparameters. We then compare the results on various reasoning tasks such as BoolQ (Clark et al., 2019), PIQA (Bisk et al., 2020), Social IQA (Sap et al., 2019), HellaSwag (Zellers et al., 2019), Winogrande (Sakaguchi et al., 2021), and OBQA (Hu et al., 2023), and summarize the results in Table 5. Finally, we train ViT-B and ViT-L models (Dosovitskiy et al., 2021) by LoRA on various image classification tasks including Cars (Krause et al., 2013), CIFAR-100 (Krizhevsky, 2009), CUB-200 (Wah et al., 2011), DTD (Cimpoi et al., 2014), Food-101 (Bossard et al., 2014), RESISC45 (Cheng et al., 2017), and SUN397 (Xiao et al., 2010). Table 6 summarizes the results. Detailed settings and extended results are provided in Appendix D.

Our automatic hyperparameter tuning, again, shows comparable and often better performance than the baseline optimizers with fixed hyperparameters. This improvement is consistent across various architectures and tasks. We emphasize that the overhead of our method is only less than 5% of the training time, and even less than what is measurable for ViT training, where we encountered speed *improvement* instead of slowdown. Given the huge amount of time required for hyperparameter tuning, this additional cost is acceptable. In summary, our theory enables us to greatly reduce the workload of manual hyperparameter tuning in practical scenarios, where computations are scarse resources. In addition to this, we notice that our automatic framework can be combined with validation-aware tuning of optimizing environments, which is typically done by human engineers by manual inspection of validation curves. We provide a proof-of-concept in Appendix E.

## 5 CONVERGENCE ENDPOINT OF GREEDY OPTIMAL OPTIMIZERS

Despite the greedy objective for instantaneous progress of learning, the previous experiments reveal that the resulting solutions often yield better test accuracy as well. This section provides a theoretical foundation for this empirical observation. Note that our primary goal here is to enhance the usability of existing families of optimizers that are already verified both theoretically and empirically. That is, we assume that the optimizers under study has a well-established convergence analysis in its general form, i.e., regardless of the choice of hyperparameters. For readers who are interested in these aspects, we refer to works of Ghadimi & Lan (2013); Yang et al. (2016); Reddi et al. (2018); Zhou et al. (2024); Assran & Rabbat (2020); Li & Orabona (2020); Cutkosky & Mehta (2020); Defóssez et al. (2022). We instead focus on the dependence of the learning dynamics and its target endpoint on the greedy selection of optimizer from the predefined family of optimizers.

Table 6: AdamW on Vision Transformer fine-tuning tasks with LoRA. mean ± std.

| ViT-B (rank = 32) | Cars | CIFAR-100 | CUB-200 | DTD | Food-101 | RESISC45 | SUN397 | Avg |
|---|---|---|---|---|---|---|---|---|
| Best baseline | 77.56 ± 0.09 | 91.74 ± 0.07 ● | 84.67 ± 0.06 ● | 78.32 ± 0.38 ● | 88.13 ± 0.03 | 94.54 ± 0.01 ● | 72.72 ± 0.08 ● | 83.95 ± 0.11 |
| Ours | 77.95 ± 0.38 ● | 91.87 ± 0.02 ● | 84.56 ± 0.13 | 78.23 ± 0.48 | 88.16 ± 0.09 ● | 94.24 ± 0.09 | 72.71 ± 0.21 | 83.96 ± 0.20 ● |

| ViT-L (rank = 8) | Cars | CIFAR-100 | CUB-200 | DTD | Food-101 | RESISC45 | SUN397 | Avg |
|---|---|---|---|---|---|---|---|---|
| Best baseline | 84.89 ± 0.12 | 93.20 ± 0.08 ● | 87.08 ± 0.21 ● | 80.04 ± 0.18 | 89.98 ± 0.07 | 95.13 ± 0.08 ● | 75.18 ± 0.10 ● | 86.50 ± 0.12 |
| Ours | 85.40 ± 0.11 ● | 93.05 ± 0.01 | 86.80 ± 0.12 | 80.74 ± 0.44 ● | 90.04 ± 0.15 ● | 95.07 ± 0.02 | 74.87 ± 0.08 | 86.57 ± 0.13 ● |

We first briefly discuss within the context of least squares case, which can be extended to more general problems.

**Proposition 7** (Convergence endpoint of commutative optimizers for least squares). [proof] *Let $Q \succeq 0$ be an optimizer. For least squares loss $\mathcal{L}(\theta) = \frac{1}{2}\|J\theta - y\|^2$, the parameter motion is $\dot{\theta} = -QJ^\top(J\theta - y)$. The convergence endpoint $\theta^\infty$ is the minimum $Q^{-1}$-norm solution:*

$$\theta^\infty := \arg\min_{J\theta=y} \|\theta\|^2_{Q^{-1}} = QJ^\top(JQJ^\top)^{-1}y. \tag{17}$$

*Moreover, if $Q$ commutes with the gradient moment $R = J^\top J$, i.e., $QJ^\top J = J^\top JQ$, then*

$$\theta^\infty = J^\top(JJ^\top)^{-1}y = \arg\min_{J\theta=y} \|\theta\|^2 =: \theta^\star, \tag{18}$$

*where $\theta^\star$ is the minimum (Euclidean) norm solution, or the canonical pseudoinverse solution.*

The proof is in Appendix H. Two important observations on the convergence endpoint can be made: First, since the optimizer $Q$ is symmetric PSD, the least squares problem *always* achieves its minimum training loss. Second, if $Q$ commutes with the gradient moment $R = J^\top J$, then the endpoint is independent of the value of the optimizer $Q$, and equals the canonical pseudoinverse solution $\theta^\star = J^\dagger y$. In particular, the second observation implies that we can further achieve *implicit regularization* by tuning the optimizer $Q$ to achieve certain alignment condition with the gradient moment $R = J^\top J$. The commutativity condition can preserve the implicit bias of vanilla gradient descent. The following lemma shows that our greedy formulation achieves this.

**Lemma 8** (Commutativity). [proof] *For the four families of optimizers in Section 2, i.e., Frobenius ball, spectral, data-metric, and diagonal families, the optimal optimizers $Q^\star \in \arg\max_{Q \in \mathcal{Q}} \text{Tr}(QR)$ commute with any symmetric PSD matrix $R$, including those with the form $R = J^\top J$:*

$$Q^\star R = RQ^\star. \tag{19}$$

If the loss function is not a sum of squares, e.g., a nonparametric loss function $\ell(f, y)$ with gradient $s = \nabla_\theta \ell(f, y)$, then $R = \mathbb{E}[gg^\top] = \mathbb{E}[J^\top ss^\top J] =: J^\top \Sigma_s J$ is the gradient moment, where $\Sigma_s := \mathbb{E}[ss^\top]$ is the gradient moment of the criterion $\ell$. Even in this case, Lemma 8 shows that $Q^\star$ does commute with $R$. However, $Q^\star$ may not commute with the Gram matrix $J^\top J$ in general, which prevents achieving the canonical pseudoinverse solution of Theorem 7. Nevertheless, by carefully choosing the desired families of optimizers, we can also achieve this commutativity condition $[Q, J^\top J] = 0$. Notable examples of such commuting optimizers are the families of simple gradient descent (with or without momentum) that are scalar multiples of the identity matrix, and the families of Adam optimizers sharing the same $\beta_2$ value, which fixes the diagonal structure of the optimizer. These are the families that used in the demonstrations in Section 3.

In short, our greedy optimal optimizer does minimal disruption to the convergence endpoint for the least squares problem by maintaining the commutativity property of Lemma 8. This effect can be seen in Figure 2(a,d), where the greedy optimal optimizer achieves the fastest convergence without suffering from the nonisotropic curvature of the problem, leading to precise minimization.

Extending this to more general problems requires a concept of *kernels*. Define the *optimizer-augmented kernel* (OAK) $K_Q$ induced by the optimizer $Q$ (Jacot et al., 2018; Geifman et al., 2024):

$$K_Q(x, x') := J(x)QJ(x')^\top. \tag{20}$$

The kernel-induced dynamics in the function space near a given interpolation point $X$ is governed by the OAK $K_Q$:

$$\dot{f}(\cdot; t) = -K_Q(\cdot, X; t)(f(X; t) - y), \tag{21}$$

with time $t$. This is analogous to the parameter motion formula in Proposition 7, but now in the function space. The solution to this ODE at training points $X$ is:

$$f(X;t) = y - e^{-K_Q t}(y - f(X;0)). \tag{22}$$

Setting $t \to \infty$ in the above equation, we have the following theorem, which shows that our greedy optimal optimizer reduces the RKHS norm of the fitted function $f^\infty$ globally without altering the convergence endpoint. The proof is in Appendix H.

**Theorem 9** (Convergence endpoint of greedy optimal optimizers). [`proof`] *Let $Q^\star$ solve* $\max_{Q \in \mathcal{Q}} \langle Q, R \rangle$ *for a Frobenius, spectral, data-metric, or diagonal families, where $R = \mathbb{E}[gg^\top]$. Then, the convergence endpoint $f_{Q^\star}^\infty$ is, under squared loss and OAK dynamics with small initialization, the unique minimum-norm interpolant in the RKHS $\mathcal{H}_{K_{Q^\star}}$:*

$$f_{Q^\star}^\infty(\cdot) = \arg \min_{f(X)=y} \|f\|_{\mathcal{H}_{K_{Q^\star}}}^2 = K_{Q^\star}(\cdot, X) K_{Q^\star}^\dagger y, \quad \|f_{Q^\star}^\infty\|_{\mathcal{H}_{K_{Q^\star}}}^2 = y^\top K_{Q^\star}^\dagger y \tag{23}$$

*where $K_{Q^\star}^\dagger$ is the Moore-Penrose pseudoinverse of $K_{Q^\star}$. In particular, whenever interpolation is possible, all such $Q$ achieve the same minimal training loss in this OAK dynamics.*

The theorem shows that our greedy optimal optimizers $Q^\star$ reduce the RKHS norm of the fitted function $f_{Q^\star}^\infty$ globally. Least squares problem is a special case that pins the endpoint to the canonical pseudoinverse solution. This explains why maximizing instantaneous progress of learning often leads to the best long-horizon convergence endpoint, and in turn, better validation performance. The results further justifies our automatic hyperparameter tuning framework.

## 6 SCOPE AND LIMITATIONS

**Long-horizon objective from greedy paradigm.** In order to simplify the analysis, this work resorts to the greedy paradigm, primarily focusing on instantaneous progress of learning. As a trade-off, global optimality guarantee requires further investigation under this greedy paradigm. We have provided a theoretical analysis on the convergence targets near the linearized dynamics regime in Section 5, and rely on more sophisticated guarantees on the families of optimizers from optimizer-specific prior works. It is, therefore, recommended to combine our theoretical results with the optimizer-specific guarantees in order to use our theory to devise a new type of optimizers.

**Choice of trust region types.** Rather than determining which optimizer class is optimal for a given task, this work provides an optimization framework *within* user-defined feasible sets. Our framework helps engineers by reducing hyperparameter search effort; however, it still requires intelligent choice of trust region types, i.e., which optimizer *class* fits the task.

**Renovating existing optimizers.** In Theorems 1 and 3, we provide general construction of optimal optimizers from convex constraints. However, in this work, we focus on the well-established optimizers, and supplement them with a systematic methodology to find the right hyperparameters for a given task. Designing a new class of optimizers will be a natural extension of this work.

## 7 CONCLUSION

We established a firm theoretical grounding for systematically achieving optimal optimizers in a greedy sense. Our convex optimization framework connects commonly used optimizers to convex constraint sets, merging those independently developed techniques into a single unified framework in Table 7. Our main results, Theorems 1 and 3, and Lemma 4 are general tools that can be extended to arbitrary trust region to invent new families of optimizers for specific uses. Our theory, therefore, does not disprove the *no free lunch theorem*; rather, it provides a principled way to *leverage* this wisdom to flexibly design and adapt optimizers for our own problems at hand.

ETHICS STATEMENT

We acknowledge the ICLR Code of Ethics.

REPRODUCIBILITY STATEMENT

All the proofs of the theoretical part of this paper, including every Lemma, Theorem, Proposition, and Corollary, are provided in Appendix H with detailed derivations, starting from the basic definitions and assumptions made in the main text. Moreover, omitted theoretical results are elaborated in Appendix G. Regarding the implementation, Appendix C gives the algorithm to realize our theoretically justified optimal optimizers.

LLM USAGE STATEMENT

We deeply acknowledge the usefulness of LLMs in revising the manuscript, especially for fixing vocabulary and grammar-related issues. We also used LLMs to check the correctness and coherence of the proofs and notations. This greatly helped us in identifying awkward mistakes we had been making all along.

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

Table 7: **Unified table of optimizers:** moment budgets, cones, and optimal hyperparameter expressions. For all rows, the learning rate parameter $\eta$ is determined by the trust region or budget (see text for details) in theory, and can be empirically maximized in practice. Here, $a(\cdot)$ denotes a normalization factor that depends on the hyperparameters of the optimizer family $\mathcal{Q}$ and (especially) the cone $\mathcal{C}$, $m_\beta$ is the running average momentum operator with parameter $\beta$ defined as $m_\beta[n] = g[n] + \beta m_\beta[n-1]$, and $u = Qg = -\Delta\theta$ is the parameter update operator. For detailed derivations of these results, see the links for each optimizer name below.

| Optimizer | Budget $\mathcal{Q}$ | Cone $\mathcal{C}$ | Optimal Hyperparameters / Formulas |
|---|---|---|---|
| Gradient Descent | $\|Q\|_{\mathcal{H}}^2 \leq B$ | $\eta I \delta[n]$ | $\eta^\star = \sqrt{B/d}, \ d = \dim(\theta)$ |
| SGD+Momentum | $\|Q\|_{\mathcal{H}}^2 \leq B$ | $(1-\beta)\beta^n I$ | $\beta^\star = \arg\max_\beta a(\beta)\, \mathbb{E}[g^\top m_\beta]$ |
| Precond. GD + Momentum | $\|Q\|_{\mathcal{H},P^{-1}}^2 \leq B$ | $Q[n] = \eta(1-\beta)\beta^n P$ | $\beta^\star = \arg\max_\beta a(\beta,P)\, \mathbb{E}[g^\top P m_\beta]$ |
| Newton's Method | $\|Q\|_{\mathcal{H},H}^2 \leq B$ | $\eta(1-\beta)\beta^n H^{-1}$ | $\beta^\star = \arg\max_\beta a(\beta,H)\, \mathbb{E}[g^\top H^{-1} m_\beta]$ |
| Natural Gradient Descent | $\|Q\|_{\mathcal{H},F}^2 \leq B$ | $\eta(1-\beta)\beta^n F^{-1}$ | $\beta^\star = \arg\max_\beta a(\beta,F)\, \mathbb{E}[g^\top F^{-1} m_\beta]$ |
| K-FAC | $\|Q\|_{\mathcal{H},\mathrm{bdiag}}^2 \leq B$ | $\eta(1-\beta)\beta^n \, \mathrm{bdiag}(F_\ell^{-1})$ | $\beta^\star = \arg\max_\beta a(\beta,\{F_\ell\})\, \mathbb{E}[g^\top \mathrm{bdiag}(F_\ell^{-1}) m_\beta]$ |
| Shampoo | $\|Q\|_{\mathcal{H},\mathrm{Kron}}^2 \leq B$ | $\eta(1-\beta)\beta^n \bigotimes_i G_i^{-1/2}$ | $\beta^\star = \arg\max_\beta a(\beta,\{G_i\})\, \mathbb{E}[g^\top (\bigotimes_i G_i^{-1/2}) m_\beta]$ |
| Full-matrix AdaGrad + Mom | $\|Q\|_{\mathcal{H},G^{-1/2}}^2 \leq B$ | $\eta(1-\beta)\beta^k G^{-1/2}$ | $\beta^\star = \arg\max_\beta a(\beta,G)\, \mathbb{E}[g^\top G^{-1/2} m_\beta]$ |
| Diagonal AdaGrad + Mom | $\sum_j c_j \sum_k |q_{j,k}|^2 \leq B$ | $\eta(1-\beta)\beta^k \mathrm{diag}(1/c_j)$ | $\beta^\star = \arg\max_\beta a(\beta)\, \mathbb{E}[g^\top u_{\beta,\mathrm{diag}}]$ |
| RMSProp-style | $\sum_j c_j |q_j|^2 \leq B$ | $\eta \, \mathrm{diag}(1/c_j)$ | $\eta^\star = |g|/\sqrt{\mathbb{E}[g^2]}$ (coordinate-wise) |
| Adam/AdamW | $\sum_j c_j \sum_k |q_{j,k}|^2 \leq B$ | $\mathrm{diag}((1-\beta_1)\beta_1^n/c_j)$ | $\beta_1^\star, \beta_2^\star = \arg\max_{\beta_1,\beta_2} a(\beta_1,\beta_2)\, \mathbb{E}[g[n]^\top u_{\beta_1,\beta_2}[n]]$ |
| Muon | $\|\Delta\Theta\|_{\mathrm{op}} \leq \gamma$ | $\eta \, \mathrm{Ortho}(B_\mu[n])$ | $\mu^\star = \arg\max_\mu \, \mathbb{E}\big[\langle G_n, \mathrm{Ortho}\big(\sum_{k=0}^\infty \mu^k G_{n-k}\big)\rangle\big]$ |

# A  MASTER TABLE OF OPTIMIZERS

Throughout this work, we have derived various types of optimizers from our convex optimization framework. We can now register various optimizers under a single unified table as shown in Table 7. For detailed derivation of each optimizer family, *click the first column items* of the table to jump to the corresponding section. For SGD with momentum, Adam (Kingma & Ba, 2015), and AdamW (Loshchilov & Hutter, 2019), see Corollaries 5 and 6 in the main text. For other optimizers, we have provided the detailed derivation in Appendix F.

# B  RELATED WORK

**Categorization and unification of optimizers.** The closest work to ours is AdaReg (Gupta et al., 2017), which presents a minimization framework for selecting the best optimizer adaptively. Specifically, AdaReg generalizes AdaGrad (Duchi et al., 2011) and Online Newton Step (ONS) (Hazan et al., 2007) into solutions of a single convex optimization problem that resembles our problem P2 for the stateless case. However, this is done with a regularization term $\Phi(Q)$ that penalizes the complexity of the optimizer. Our framework extends this idea to significantly broader family of optimizers, including stateless *and* dynamic optimizers, with respect to general families of *trust regions*. This allows us to unify many existing optimizers in practice and find their optimal hyperparameters. On the other hand, Frank-Wolfe methods (Frank & Wolfe, 1956; Garber & Wolf, 2021) considers finding the most aligned optimization step for the constrained convex optimization problem. This alignment principle is similar to our analysis in Section 5, where we generalize this idea to the practical algorithms for deep learning.

**Performance-guided discovery of optimizers.** Another line of work is the performance estimation problem (PEP) framework (Drori & Teboulle, 2014; Kim & Fessler, 2017; Taylor et al., 2017; Goujaud et al., 2024), where first order methods for convex optimization are categorized, compared, and suggested based on their worst-case performance. Although we share a general philosophy to algorithmically suggest the best optimizer for each task, our greedy paradigm is orthogonal to the PEP framework, as we focus on the instantaneous performance of the optimizers in general gradient-based learning. We also encompass a broader family of optimizers, unifying existing widely-used optimizers in deep learning such as SGD with Nesterov momentum (Nesterov,

1983), AdamW (Loshchilov & Hutter, 2019), LAMB (You et al., 2020), K-FAC (Martens & Grosse, 2015), Shampoo (Gupta et al., 2018), and Lion (Chen et al., 2023).

**Symbolic discovery of optimizers.** Techniques like symbolic discovery (Chen et al., 2023; Zheng et al., 2022), non-parametric optimizer search (Wang et al., 2022), and neural optimizer search (Bello et al., 2017) are also related to our work, as their objective is to discover the optimal optimizer for a given task. In their framework, symbolic optimizers are obtained by a tree-based search of a predefined set of optimizers. Ours instead lets the engineer select the broader family of optimizers, and then provides a mathematical tool to find the optimal solution among them. Therefore, these works are also orthogonal to ours.

**Hyperparameter optimization.** Many works have proposed to automatically tune the hyperparameters governing optimization. Most of them adopt a learning framework to find a good set of hyperparameters including learning rates (Daniel et al., 2016), their schedules (Xu et al., 2017; 2019), and other optimizer parameters (Shaban et al., 2019). Hypergradient methods (Maclaurin et al., 2015; Baydin et al., 2017; Grazzi et al., 2020; Moskovitz et al., 2019) are also proposed to find the optimal hyperparameters. Instead of resorting to learning-based methods, we establish a theoretical framework through the lens of convex optimization problems (Boyd & Vandenberghe, 2004). By doing so, we can classify well-used optimizers such as SGD with momentum and Adam (Kingma & Ba, 2015) as special cases of our framework, and provide a systematic way to determine the optimal hyperparameters for these optimizers.

**Learning to optimize.** Learning to optimize (Li & Malik, 2016) aims to adapt the optimizer to a given task by treating optimizers as learnable parametric models (Andrychowicz et al., 2016). Various architectures have been explored, including RNNs (Andrychowicz et al., 2016; Wichrowska et al., 2017; Lv et al., 2017), Transformers (Chen et al., 2022; Moudgil et al., 2023; Jain et al., 2024), and per-tensor HyperNetworks (Ha et al., 2016; Metz et al., 2022). Their primary focus is on meta-training these optimizer-networks for stability and adaptability. These works represent a nontraditional, network-based family of generally nonconvex optimizers, which is not generally compatible with our framework which is based on convex optimization.

**Learning to learn.** Rooted in the human-inspired philosophy (Schmidhuber, 1987; Bengio et al., 1990), meta-learning is another line of work that shares a similar spirit with learning to optimize (Gharoun et al., 2023). A large proportion of works on meta-learning target few-shot learning tasks, which prepare the model, not the optimizer, for downstream tasks (Vinyals et al., 2016; Finn et al., 2017; Yu et al., 2024b; Sun et al., 2019). Among them, Meta-SGD (Li et al., 2017) is noteworthy, as it prepares the optimizer. However, the problem set we address is general gradient-based learning, which differs from the tasks of concern in meta-learning.

## C    INSTANTIATION OF OPTIMAL OPTIMIZERS

In Section 3 of the main manuscript, we introduced a practical instantiation of the theoretically derived optimal conditions for the SGD+Momentum and Adam (Kingma & Ba, 2015; Loshchilov & Hutter, 2019) families of optimizers. This section gives the instantiation of the algorithms for these optimal conditions. Practical considerations are also suggested.

### C.1    GENERAL META-ALGORITHM FOR OPTIMAL OPTIMIZER BY SELECTION

In the unified table of optimizers in Table 7, we see similar structures for the optimal conditions reappearing in different optimizer families. Specifically, the greedy optimality framework suggests that we select the optimizer among available options based on the inner product of (1) the instantaneous gradient $g = -\nabla_\theta \mathcal{L}$ from the gradient calculation and (2) the parameter update $u = Qg = -\Delta\theta$ from each optimizer. This leads to the following meta-algorithm for selecting the optimal optimizer, which is the simplest instantiation of this greedy optimality.

In this instantiation, we dynamically select the optimal optimizer from a predefined set of candidate optimizers based on the analytically calculated objective function, i.e., the operand of argmax in

the last column of Table 7. For example, for the SGD+Momentum optimizer family, the objective function is

$$J(\beta) = a\beta \cdot \mathbb{E}[g^\top m_\beta], \tag{24}$$

where $a(\beta)$ is a scalar normalization factor and $m_\beta$ is the running average momentum operator with parameter $\beta$ defined as $m_\beta[n] = g[n] + \beta m_\beta[n-1]$. For the Adam optimizer family, the objective function is

$$J(\beta_1, \beta_2) = a(\beta_1, \beta_2) \cdot \mathbb{E}[g^\top u_{\beta_1, \beta_2}], \tag{25}$$

where $a(\beta_1, \beta_2)$ is a scalar normalization factor, $u_{\beta_1, \beta_2} = m_{\beta_1}/v_{\beta_2}^{1/2}$ is the Adam update and $v_{\beta_2}$ is the running average second-moment operator with parameter $\beta_2$ defined as $v_{\beta_2}[n] = \beta_2 v_{\beta_2}[n-1] + (1 - \beta_2)g_j^2[n]$. During optimization, the distributions of gradients $g$, the momenta $m_\beta$, and the parameter updates $u$ are all time-varying, and the exact value of the expectation $\mathbb{E}$ in the equations 24 and 25 are unknown. Therefore, we approximate this by an immediate dot product between the instantaneous gradient $g[n]$ and the instantaneous parameter update $m[n]$ or $u[n]$.

This leads to Algorithm 1. The lines highlighted in blue are the ones that are different from the standard training loop. In all our experiments, we use maximal of $K = 2$ candidate optimizers. In practice, this only adds less than 5% computational overhead as elaborated in Table 17.

---

**Algorithm 1** Optimal $K$-Choice Switch Optimizer

---

**Require:** Candidate optimizers $\{Q_1, Q_2, \ldots, Q_K\}$ with hyperparameters $\{\beta_1, \beta_2, \ldots, \beta_K\}$
 1: Initialize model and all candidate optimizers.
 2: **for** each training step $n$ **do**
 3:     Compute forward pass: $\mathcal{L}(\theta[n])$.
 4:     Compute current gradient: $g[n] = -\nabla_\theta \mathcal{L}(\theta[n])$.
 5:     **for** each candidate optimizer $Q_k$ with hyperparameter $\beta_k$ **do**
 6:         Update internal state: $Q_k \leftarrow Q_k(g[n])$.
 7:         Evaluate objective function: $J(\beta_k, g[n])$.
 8:     **end for**
 9:     Get the optimal optimizer: $Q^\star = Q_k$ where $k = \arg\max_k J(\beta_k, g[n])$.
 10:     Update parameters: $\theta[n+1] \leftarrow \theta[n] - Q^\star g[n]$.
 11:     Apply hysterisis reset for optimizer internal states to stabilize selection.
 12: **end for**

---

### C.2   PRACTICAL CONSIDERATIONS.

There are several practical considerations to keep in mind when applying the above algorithm. First, selection stability can be improved by maintaining an EMA of the objective function values $J(\beta_k, g[n])$ and choosing the optimizer with the highest EMA. Additionally, a hysteresis threshold can be used to prevent frequent switching between different optimizers: only update the chosen optimizer if the new selection $k$ remains consistently different from the previous selection $k'$ for several consecutive steps. Finally, particularly in the initial stages of training when gradients change rapidly, we often observe that the objective $J$—that is, the inner product of the gradient and the proposed parameter update—becomes negative. This suggests that the parameter update has moved outside the local region where the gradient meaningfully reflects the underlying geometry of the loss function. In such cases, instability can be mitigated by manually reducing the internally stored optimizer states, effectively providing a soft reset for the algorithm. Throughout our experiments, we found that decaying the internal optimizer states by half when the objective $J$ is negative for five consecutive steps is effective in mitigating instability. We use *only* this state decay technique across all experiments in this work. We did not use other practical stabilization techniques when reporting the results in this work, although we have observed consistent benefits from all the aforementioned techniques. This is to keep the presentation of the main manuscript focused on the core ideas and results. We will publish the code upon the publication of this work.

### C.3   OPTIMAL SGD+MOMENTUM AND ADAM

To gain further insightes on the actual instantiation of this meta-algorithm for optimal optimizer selection, we provide the following two algorithms for the implementation of the optimal

SGD+Momentum and Adam. Note how the hysterisis reset is applied in the last lines of the algorithms. We did not use other practical stabilization techniques that are not mentioned in the main manuscript.

---

**Algorithm 2** Optimal SGD+Momentum by $K$-choice switch

---

**Require:** Learning rate $\eta$, number of candidate optimizers $K$, candidate optimizers $\{Q_1, Q_2, \ldots, Q_K\}$ with hyperparameters $\{\beta_1, \beta_2, \ldots, \beta_K\}$, respectively.
1: Initialize optimizer states $\mu_k \leftarrow 0, \forall k \in \{1, 2, \ldots, K\}$, hysteresis counter $H \leftarrow 0$.
2: **for** each training step $n$ **do**
3:     $g \leftarrow \nabla_\theta \mathcal{L}(\theta)$                                    ▷ Standard forward-backward pass
4:     **for** $k = 1, \ldots, K$ **do**
5:         $\mu_k \leftarrow \beta_k \mu_k + g$                              ▷ Update optimizer states
6:         $J_k \leftarrow \sqrt{1 - \beta_k^2}\, g^\top \mu_k$               ▷ Objective function for $k$-th optimizer
7:     **end for**
8:     $k^\star \leftarrow \arg\max_k J_k$                          ▷ Dynamic selection of optimizer
9:     $\theta \leftarrow \theta - \eta \mu_{k^\star}$                ▷ Parameter update with $k^\star$-th optimizer
10:    **if** $J_{k^\star} < 0$ for consecutive 5 steps **then**       ▷ Hysteresis reset (stabilization trick)
11:        **for** $k = 1, \ldots, K$ **do**
12:           $\mu_k \leftarrow 0.5\mu_k$
13:        **end for**
14:    **end if**
15: **end for**
16: **return** $\theta$

---

**Algorithm 3** Optimal Adam by $K$-choice switch

---

**Require:** Learning rate $\eta$, number of candidate optimizers $K$, candidate optimizers $\{Q_1, Q_2, \ldots, Q_K\}$ with hyperparameters $\{(\beta_1^{(1)}, \beta_2^{(1)}), (\beta_1^{(2)}, \beta_2^{(2)}), \ldots, (\beta_1^{(K)}, \beta_2^{(K)})\}$, respectively.
1: Initialize optimizer states $\mu_k, v_k \leftarrow 0, \forall k \in \{1, 2, \ldots, K\}$, hysteresis counter $H \leftarrow 0$
2: **for** each calibration step $t$ **do**
3:     $g \leftarrow \nabla_\theta \mathcal{L}(\theta)$                                   ▷ Standard forward-backward pass
4:     **for** $k = 1, \ldots, K$ **do**
5:         $v_k \leftarrow \beta_2^{(k)} v_k + (1 - \beta_2^{(k)})g^2$                 ▷ Update optimizer states
6:         $\mu_k \leftarrow \beta_1^{(k)} \mu_k + (1 - \beta_1^{(k)})g$
7:         $c_k \leftarrow \sqrt{v_k}$                                   ▷ Coordinate-wise costs
8:         $u_k \leftarrow \mu_k/(c_k + \epsilon)$                             ▷ Adam update
9:         $a_k \leftarrow \sqrt{1 + \beta_1^{(k)}}/\sqrt{(1 - \beta_1^{(k)})\sum_j (1/c_k)}$     ▷ Normalization factor
10:       $J_k \leftarrow a_k\, g^\top u_k$                        ▷ Objective function for $k$-th optimizer
11:    **end for**
12:    $k^\star \leftarrow \arg\max_k J_k$                         ▷ Dynamic selection of optimizer
13:    $\theta \leftarrow \theta - \eta u_{k^\star}$               ▷ Parameter update with $k^\star$-th optimizer
14:    **if** $J_{k^\star} < 0$ for consecutive 5 steps **then**       ▷ Hysteresis reset (stabilization trick)
15:        **for** $k = 1, \ldots, K$ **do**
16:           $\mu_k \leftarrow 0.5\mu_k$
17:        **end for**
18:    **end if**
19: **end for**
20: **return** $\theta$

---

# D   Implementation details and more experiments

This section provides implementation details and full results of the experiments conducted to validate the theory in Section 3. First, we provide the hyperparameters and settings for the experiments in Table 8, Table 9, Table 10, Table 11, and Table 12 of Appendix D.1. Additional experimental results are provided in Table 13, Table 14, Table 16, Table 15 of Appendix D.2. Our automatic hyperparameter tuning shows comparable performance across all datasets and models, including conventional residual networks (He et al., 2016), vision transformers (Dosovitskiy et al., 2021), and modern large language models (Gemma Team et al., 2023; Grattafiori et al., 2024) with

or without using parameter-efficient fine-tuning methods like low-rank adaptation (LoRA) (Hu et al., 2022). This demonstrates the practical usefulness of our framework.

### D.1 IMPLEMENTATION DETAILS

**ResNet-18 on CIFAR-100.** For ResNet-18 (He et al., 2016) on CIFAR-100 (Krizhevsky, 2009) experiments, we follow the standard settings of He et al. (2016): 300 epochs with a learning rate decay of 0.1 at epochs 60, 120, and 160. All hyperparameters other than momentum are held fixed. We use a weight decay of $5 \times 10^{-4}$, batch size of 128, and a base learning rate of 0.1 for SGD and 0.01 for Adam (Kingma & Ba, 2015). For our optimal Adam-type optimizers using the *two-option switch*, we use $\beta_1$ endpoints of 0.8 and 0.99, to ensure that these endpoints enclose the typical range of $\beta_1$ values used in practice. For our optimal SGD+Momentum-type optimizers using the *two-option switch*, we use momentum endpoints of 0.01 and 0.99, again to ensure that the endpoints enclose the typical range of momentum values in practice. For optimal SGD+Momentum-type optimizers using the *five-option switch*, we use momentum endpoints of 0.9, 0.95, 0.98, 0.99, and 0.995, to demonstrate the effectiveness of fine-grained control in dynamic hyperparameter tuning. In the main manuscript, we show only the results of the *two-option switch* for SGD+Momentum and Adam, since these do not exceed 10% of the computation time of the baseline. Our *five-option switch* for SGD+Momentum demonstrates that we can achieve *significantly better performance* than the baseline optimizer with fixed hyperparameters ($77.57\% \rightarrow 78.33\%$ test accuracy). These extended results are summarized in Tables 13 and 14 in the next section. However, current implementation of multi-option switch larger than two requires a significant amount of computation time (around +100% compared to the baseline) and memory usage. Therefore, we did not include the results in the main manuscript, opening up a future direction for more efficient implementation.

Table 8: Hyperparameters and settings for math finetuning experiments on Gemma-2B (Gemma Team et al., 2023) with LoRA (Hu et al., 2022). Values reflect the experimental script.

| Parameter | Value(s) / Description |
|---|---|
| Dataset | MetaMathQA-395K |
| Training subset size | 100,000 |
| Models tested | Gemma-2B (`google/gemma-2b`) |
| Hardware | $1 \times$ A100-80GB |
| Precision | Bfloat16 (BF16) |
| Optimizer | AdamW (Kingma & Ba, 2015; Loschilov & Hutter, 2019) |
| Optimal AdamW-type optimizer | *two-option switch* with $\beta_1$ endpoints of 0.8 and 0.99 |
| Epochs | 1 |
| Batch size ($bs$) | 32 |
| Learning rate ($lr$) | $2 \times 10^{-4}$ |
| Weight decay | 0 |
| Warmup ratio | 0 |
| Adapter type | LoRA (Hu et al., 2022) |
| LoRA Rank ($r$) | 32 |
| LoRA Scaling ($\alpha$) | 4 |
| LoRA Dropout | 0.05 |
| Cutoff length | 256 |
| Adapter target modules | `q_proj, k_proj, v_proj, o_proj` `down_proj, up_proj, gate_proj` |

**Gemma-2B and Llama-3-8B on MetaMathQA-395K.** We summarize the hyperparameters and training settings for these experiments in Table 3 and Table 4 in Section 3 of the main manuscript. We use low-rank adaptation (LoRA) (Hu et al., 2022) with a rank of 32 and a scaling factor of 4 for both Gemma-2B (Gemma Team et al., 2023) and Llama-3-8B (Grattafiori et al., 2024). We truncate the MetaMathQA-395K (Yu et al., 2024a) training dataset to 100,000 examples for both models. The reported test accuracy is based on a separate, validation-only dataset, GSM8K (Cobbe et al., 2021). Additional experimental results are provided in Table 15, where we show results for baseline optimizers with different $\beta_1$ values we have tested. In the main manuscript, only the best baseline optimizer results are shown for brevity: $\beta_1 = 0.5$ for Gemma-2B and $\beta_1 = 0.9$ for Llama-3-8B. For our optimal AdamW-type optimizers using the *two-option switch*, we use $\beta_1$ endpoints of 0.8 and 0.99, to ensure that these endpoints enclose the typical range of $\beta_1$ values used in practice.

Table 9: Hyperparameters and settings for math finetuning experiments on Llama-3-8B (Grattafiori et al., 2024) with LoRA (Hu et al., 2022). Values reflect the experimental script.

| Parameter | Value(s) / Description |
|---|---|
| Dataset | MetaMathQA-395K |
| Training subset size | 100,000 |
| Models tested | Llama-3-8B (`meta-llama/llama-3-8b`) |
| Hardware | $1 \times$ A100-80GB |
| Precision | Bfloat16 (BF16) |
| Optimizer | AdamW (Kingma & Ba, 2015; Loshchilov & Hutter, 2019) |
| Optimal AdamW-type optimizer | *two-option switch* with $\beta_1$ endpoints of 0.8 and 0.99 |
| Epochs | 1 |
| Batch size ($bs$) | 32 |
| Learning rate ($lr$) | $1 \times 10^{-4}$ |
| Weight decay | 0 |
| Warmup ratio | 0 |
| Adapter type | LoRA (Hu et al., 2022) |
| LoRA Rank ($r$) | 32 |
| LoRA Scaling ($\alpha$) | 4 |
| LoRA Dropout | 0.05 |
| Cutoff length | 256 |
| Adapter target modules | `q_proj, k_proj, v_proj, o_proj` |
| | `down_proj, up_proj, gate_proj` |

Table 10: Hyperparameters and settings for the main commonsense finetuning experiments on Gemma-2B (Gemma Team et al., 2023) with LoRA (Hu et al., 2022).

| Parameter | Value(s) / Description |
|---|---|
| Dataset | Commonsense-170K (Hu et al., 2023) |
| Models tested | Gemma-2B (Gemma Team et al., 2023) |
| Hardware | $1 \times$ A100-80GB |
| Precision | Bfloat16 (BF16) |
| Optimizer | AdamW (Kingma & Ba, 2015; Loshchilov & Hutter, 2019) |
| Optimal AdamW-type optimizer | *two-option switch* with $\beta_1$ endpoints of 0.8 and 0.95 |
| Epochs | 1 |
| Batch size ($bs$) | 32 |
| Learning rate ($lr$) | $2 \times 10^{-4}$ |
| Weight decay | 0 |
| Warmup ratio | 0 |
| Adapter type | LoRA (Hu et al., 2022) |
| LoRA Rank ($r$) | 32 |
| LoRA Scaling ($\alpha$) | 4 |
| LoRA Dropout | 0.05 |
| Cutoff length | 256 |
| Adapter target modules | `q_proj, k_proj, v_proj, o_proj` |
| | `down_proj, up_proj, gate_proj` |

**Gemma-2B on Commonsense-170K.** We summarize the hyperparameters and training settings for these experiments in Table 5 in Section 3 of the main manuscript. We use low-rank adaptation (LoRA) (Hu et al., 2022) with a rank of 32 and a scaling factor of 4 for Gemma-2B (Gemma Team et al., 2023). We also demonstrate our optimizer with full fine-tuning on the Commonsense-170K (Hu et al., 2023) dataset. Additional experimental results are provided in Table 16 in the next section, where we show results for baseline optimizers with different $\beta_1$ values we have tested. In the main manuscript, only the best baseline optimizer results are shown for brevity: $\beta_1 = 0.95$ for LoRA and $\beta_1 = 0.5$ for full fine-tuning. For our optimal AdamW-type optimizers using the *two-option switch*, we use $\beta_1$ endpoints of 0.1 and 0.99 for full fine-tuning and $\beta_1$ endpoints of 0.8 and 0.95 for LoRA, to ensure that these endpoints enclose the typical range of $\beta_1$ values used in each type of experiment in practice. After fitting the Commonsense-170K (Hu et al., 2023) dataset, we evaluate performance on various reasoning datasets that are commonly used in the LLM literature. These include BoolQ (Clark et al., 2019), PIQA (Bisk et al., 2020), Social IQA (Sap et al., 2019),

Table 11: Hyperparameters and settings for the main commonsense finetuning experiments on Gemma-2B (Gemma Team et al., 2023) with full fine-tuning.

| Parameter | Value(s) / Description |
|---|---|
| Dataset | Commonsense-170K (Hu et al., 2023) |
| Models tested | Gemma-2B (Gemma Team et al., 2023) |
| Hardware | $1 \times$ A100-80GB |
| Precision | Bfloat16 (BF16) |
| Optimizer | AdamW (Kingma & Ba, 2015; Loshchilov & Hutter, 2019) |
| Optimal AdamW-type optimizer | *two-option switch* with $\beta_1$ endpoints of 0.1 and 0.99 |
| Epochs | 1 |
| Batch size ($bs$) | 32 |
| Learning rate ($lr$) | $1 \times 10^{-5}$ |
| Weight decay | 0 |
| Warmup ratio | 0 |
| Adapter type | Full fine-tuning |
| Cutoff length | 256 |

HellaSwag (Zellers et al., 2019), Winogrande (Sakaguchi et al., 2021), and OBQA (Hu et al., 2023). We also report the average performance across all datasets.

Table 12: Common hyperparameters and settings for ViT-Base and ViT-Large (Dosovitskiy et al., 2021) finetuning experiments across various classification datasets.

| Parameter | Value(s) / Description |
|---|---|
| Models tested | ViT-B / ViT-L (Dosovitskiy et al., 2021) |
| Datasets | Stanford Cars, CIFAR-100, CUB-200, DTD, Food-101, RESISC45, SUN397 |
| Hardware | $1 \times$ RTX 5090-32GB |
| Precision | Bfloat16 (BF16) |
| Optimizer | AdamW (Kingma & Ba, 2015; Loshchilov & Hutter, 2019) |
| Optimal AdamW-type optimizer | *two-option switch* with $\beta_1$ endpoints of 0.5 and 0.99 |
| Batch size ($bs$) | 256 |
| Epochs | Stanford Cars: 20 |
| | CIFAR-100: 7 |
| | CUB-200: 25 |
| | DTD: 25 |
| | Food-101: 10 |
| | RESISC45: 10 |
| | SUN397: 15 |
| Learning rate ($lr$) | Stanford Cars: $5 \times 10^{-3}$ |
| | CIFAR-100: $1 \times 10^{-3}$ |
| | CUB-200: $2 \times 10^{-3}$ |
| | DTD: $2 \times 10^{-3}$ |
| | Food-101: $2 \times 10^{-3}$ |
| | RESISC45: $2 \times 10^{-3}$ |
| | SUN397: $1 \times 10^{-3}$ |
| Weight decay | 0 |
| Warmup ratio | 0 |
| Adapter type | LoRA (Hu et al., 2022) |
| LoRA Rank ($r$) | 8, 32 |
| LoRA Scaling ($\alpha$) | 4 |
| LoRA Dropout | 0 |
| Target modules | `query, value` |

**ViT-B and ViT-L on various classification datasets.** We summarize the hyperparameters and training settings for these experiments in Table 6 in Section 3 of the main manuscript. We tested four different configurations: ViT-Base (Dosovitskiy et al., 2021) with rank-8 LoRA, ViT-Base with rank-32 LoRA, ViT-Large with rank-16 LoRA, and ViT-Large with rank-32 LoRA. The same pre-trained weights are fine-tuned for various task-specific datasets, including Stanford Cars (Krause et al., 2013), CIFAR-100 (Krizhevsky, 2009), CUB-200 (Wah et al., 2011), DTD (Cimpoi et al., 2014), Food-101 (Bossard et al., 2014), RESISC45 (Cheng et al., 2017), and SUN397 (Xiao et al.,

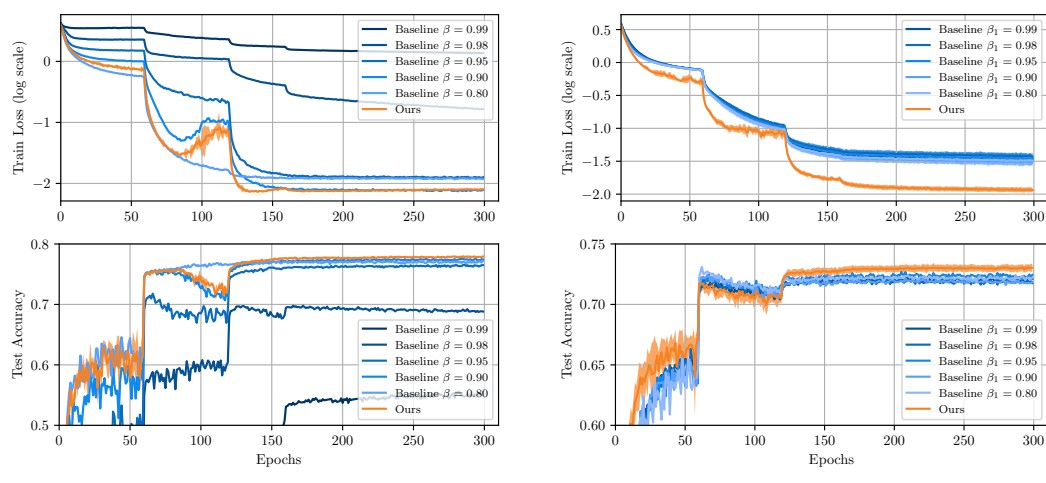

(a) Training curve of SGD+M optimizers.     (b) Training curve of Adam optimizers.

Figure 4: Demonstration of Corollaries 5 and 6. Our instantiations of optimal optimizers are compared with baselines having fixed hyperparameters on the CIFAR-100 dataset (Krizhevsky, 2009) with ResNet-18 (He et al., 2016), following the standard settings of He et al. (2016). The line and shaded area indicate the mean and standard deviation over 10 runs. For clear visualization, each baseline plot shows only the best run. For SGD+Momentum, momentum below 0.8 showed suboptimal performance.

Table 13: Full test results of SGD+Momentum on CIFAR-100 with ResNet-18. mean $\pm$ std.

| Method | Test acc. % | Train loss |
|---|---|---|
| $\beta = 0.01$ | $74.93 \pm 0.11$ | $0.0091 \pm 0.0002$ |
| $\beta = 0.1$ | $75.76 \pm 0.09$ | $0.0093 \pm 0.0001$ |
| $\beta = 0.2$ | $75.89 \pm 0.08$ | $0.0098 \pm 0.0001$ |
| $\beta = 0.5$ | $76.21 \pm 0.17$ | $0.0115 \pm 0.0001$ |
| $\beta = 0.8$ | $77.26 \pm 0.12$ | $0.0119 \pm 0.0002$ |
| $\beta = 0.9$ | $77.57 \pm 0.09$ ● | $0.0078 \pm 0.0001$ ● |
| $\beta = 0.95$ | $76.57 \pm 0.32$ | $0.0127 \pm 0.0004$ |
| $\beta = 0.98$ | $69.79 \pm 0.86$ | $0.1648 \pm 0.0148$ |
| $\beta = 0.99$ | $55.62 \pm 5.32$ | $1.3609 \pm 0.2312$ |
| $\beta = 0.995$ | $62.54 \pm 3.49$ | $0.7271 \pm 0.0892$ |
| $\beta = 0.999$ | $68.48 \pm 3.25$ | $0.1772 \pm 0.0229$ |
| **Ours (2 options)** | $78.06 \pm 0.07$ ○ | $0.0080 \pm 0.0001$ ● |
| **Ours (5 options)** | $78.33 \pm 0.49$ ○ | $0.0073 \pm 0.0001$ ○ |

2010). For our optimal AdamW-type optimizers using the *two-option switch*, we use $\beta_1$ endpoints of 0.5 and 0.99 for all configurations. This covers the working range of $\beta_1$ values used in practice for each type of experiment. We also report the average performance across all datasets.

## D.2 EXTENDED EXPERIMENTAL RESULTS

This section provides additional experimental results for the main manuscript. For visual clarity, we use gold, silver, and bronze medals to denote the best, second-best, and third-best results, respectively, in all tables hereafter.

**ResNet-18 on CIFAR-100.** Figure 3 in the main manuscript demonstrates how optimizer hyperparameters affect the final training loss and validation accuracy, and how our optimal optimizers compare to baseline optimizers with fixed hyperparameters. Here, the complete results are provided in Tables 13 and 14. Figure 4 further compares the training curves of baseline optimizers and our optimal optimizers. For clarity, each baseline plot shows only the best run, and only baselines with robust momentum values are displayed. Specifically, we show results for $\beta \in [0.8, 0.99]$ for SGD+Momentum and $\beta_1 \in [0.8, 0.99]$ for Adam (Kingma & Ba, 2015). Although the abrupt learning rate decay of the scheduler introduces perturbations that are not considered in our theory, our implementation of the greedy optimal optimizer generally reduces the training loss rapidly and achieves better performance than the baselines. Moreover, the complete

Table 14: Full test results of Adam on CIFAR-100 with ResNet-18. mean $\pm$ std.

| Method | Test acc. % | Train loss |
|---|---|---|
| $\beta_1 = 0.1$ | $72.78 \pm 0.43$ | $0.0414 \pm 0.0037$ |
| $\beta_1 = 0.2$ | $72.65 \pm 0.18$ | $0.0396 \pm 0.0023$ |
| $\beta_1 = 0.5$ | $72.86 \pm 0.14$ ● | $0.0351 \pm 0.0019$ |
| $\beta_1 = 0.8$ | $73.20 \pm 0.21$ ● | $0.0324 \pm 0.0042$ ● |
| $\beta_1 = 0.9$ | $72.85 \pm 0.38$ | $0.0314 \pm 0.0044$ ● |
| $\beta_1 = 0.95$ | $72.78 \pm 0.38$ | $0.0347 \pm 0.0068$ |
| $\beta_1 = 0.98$ | $72.69 \pm 0.20$ | $0.0372 \pm 0.0038$ |
| $\beta_1 = 0.99$ | $72.45 \pm 0.20$ | $0.0333 \pm 0.0045$ |
| **Ours (2 options)** | $73.26 \pm 0.31$ ● | $0.0115 \pm 0.0010$ ● |

Table 15: Full test results of Gemma-2B trained with MetaMathQA-395K, validated on GSM8K. mean $\pm$ std.

| Method | GSM8K acc. (%) | Train loss |
|---|---|---|
| $\beta_1 = 0.5$ | $52.57 \pm 1.10$ ● | $0.2080 \pm 0.0004$ ● |
| $\beta_1 = 0.8$ | $52.31 \pm 1.00$ ● | $0.2080 \pm 0.0004$ ● |
| $\beta_1 = 0.9$ | $51.76 \pm 0.99$ | $0.2081 \pm 0.0004$ ● |
| $\beta_1 = 0.95$ | $51.12 \pm 0.77$ | $0.2085 \pm 0.0004$ |
| $\beta_1 = 0.98$ | $51.25 \pm 0.38$ | $0.2093 \pm 0.0005$ |
| $\beta_1 = 0.99$ | $50.97 \pm 0.68$ | $0.2103 \pm 0.0004$ |
| **Ours** | $52.77 \pm 0.93$ ● | $0.2084 \pm 0.0003$ ● |

results in Tables 13 and 14 show that by increasing the number of selectable options from two to five, our instantiation of the greedy optimal optimizers achieves significantly better performance than baseline optimizers with any fixed hyperparameters. This opens up new opportunities for research into the *dynamic hyperparameter tuning* framework, which is first enabled by our theory.

**Gemma-2B and Llama-3-8B on MetaMathQA-395K.** We extend the experimental results in Table 3 of the main manuscript by providing the full baseline results in Table 15. The results show that our optimal optimizer yields a training loss comparable to the best baseline optimizer, while achieving significantly better validation accuracy. Importantly, this achievement does not result from tedious manual hyperparameter tuning, but from our *dynamic hyperparameter tuning* framework enabled by our theory.

**Gemma-2B on Commonsense-170K.** An extended version of the results in Table 5 of the main manuscript is provided in Table 16. As in the previous experiments, our optimal optimizer achieves comparable and occasionally better performance than the best baseline optimizer with fixed hyperparameters. In the main manuscript, we provided an abbreviated version of this table that only includes the best baseline optimizer. For reference, the best baseline is $\beta_1 = 0.95$ for LoRA training and $\beta_1 = 0.99$ for full fine-tuning.

### D.3 RUNTIME OVERHEAD

In this final section of experimental results, we present the runtime overhead of our optimal optimizers compared to baseline optimizers with fixed hyperparameters, as shown in Table 17. For small-scale experiments such as ResNet-18 on CIFAR-100, the runtime overhead is about 5% of the training time. However, this overhead dilutes significantly as model and dataset sizes increase. For larger and more practical experiments like LLM training, we even observe a runtime speedup, likely due to the implementation efficiency of our code. That said, we generally expect a positive runtime overhead. Overall, these results demonstrate the practical usefulness of our framework.

However, we do not claim that this is the minimal possible runtime overhead. The argmax operation that repeatedly appears in our theoretical results can be implemented in various ways; in this work, we provided only the most naïve solution: selecting between multiple fixed optimizers with different hyperparameters. This overhead can certainly be further reduced by more sophisticated implementations. We leave this for future work.

Table 16: Full test results of Gemma-2B trained with CommonsenseQA-170K. mean ± std.

| Gemma-2B (LoRA) | BoolQ | PIQA | Social IQA | HellaSwag | Winogrande | OBQA | Avg |
|---|---|---|---|---|---|---|---|
| $\beta_1 = 0.5$ | 65.25 ± 0.28 | 78.73 ± 0.27 | 73.97 ± 0.50 ⬤ | 72.81 ± 1.54 | 70.96 ± 0.90 | 72.07 ± 0.34 | 71.65 ± 0.29 |
| $\beta_1 = 0.8$ | 65.42 ± 0.20 | 78.80 ± 0.71 | 73.51 ± 0.23 | 73.46 ± 1.21 ⬤ | 71.43 ± 0.28 | 72.20 ± 0.34 | 71.83 ± 0.20 ⬤ |
| $\beta_1 = 0.9$ | 65.31 ± 0.27 | 78.87 ± 0.67 ⬤ | 73.66 ± 0.37 ⬤ | 72.97 ± 1.47 | 71.40 ± 0.30 | 73.20 ± 0.65 ⬤ | 71.99 ± 0.24 ⬤ |
| $\beta_1 = 0.95$ | 65.69 ± 0.29 ⬤ | 78.93 ± 0.49 ⬤ | 73.61 ± 0.16 ⬤ | 74.07 ± 0.28 ⬤ | 71.61 ± 0.44 ⬤ | 72.67 ± 0.68 | 72.12 ± 0.04 ⬤ |
| $\beta_1 = 0.98$ | 65.65 ± 0.27 ⬤ | 78.82 ± 0.40 | 73.52 ± 0.34 | 68.47 ± 1.83 | 71.45 ± 0.21 | 71.67 ± 1.23 | 71.14 ± 0.23 |
| $\beta_1 = 0.99$ | 65.48 ± 0.43 ⬤ | 78.69 ± 0.56 | 73.13 ± 0.15 | 68.66 ± 0.95 | 72.01 ± 0.52 ⬤ | 73.00 ± 0.43 ⬤ | 71.36 ± 0.11 |
| **Ours** | 65.31 ± 0.04 | 79.00 ± 0.36 ⬤ | 73.58 ± 0.06 | 75.09 ± 1.02 ⬤ | 71.80 ± 0.39 ⬤ | 73.27 ± 1.15 ⬤ | 72.12 ± 0.21 ⬤ |

| Gemma-2B (Full FT) | BoolQ | PIQA | Social IQA | HellaSwag | Winogrande | OBQA | Avg |
|---|---|---|---|---|---|---|---|
| $\beta_1 = 0.5$ | 62.79 ± 0.27 ⬤ | 74.12 ± 0.26 ⬤ | 66.63 ± 0.33 ⬤ | 40.50 ± 1.15 ⬤ | 61.48 ± 0.32 ⬤ | 62.60 ± 1.02 ⬤ | 61.86 ± 0.16 ⬤ |
| $\beta_1 = 0.8$ | 62.50 ± 0.22 ⬤ | 72.62 ± 0.49 ⬤ | 64.02 ± 0.19 ⬤ | 40.11 ± 0.21 ⬤ | 54.38 ± 0.62 ⬤ | 57.20 ± 0.71 ⬤ | 59.06 ± 0.35 ⬤ |
| $\beta_1 = 0.9$ | 62.42 ± 0.24 | 72.05 ± 0.38 | 62.88 ± 0.54 | 39.45 ± 0.33 | 52.28 ± 0.26 | 55.47 ± 0.52 | 57.92 ± 0.17 |
| $\beta_1 = 0.95$ | 62.38 ± 0.20 | 71.60 ± 0.16 | 62.20 ± 0.16 | 39.14 ± 0.16 | 51.33 ± 0.48 | 54.13 ± 0.98 | 57.24 ± 0.13 |
| $\beta_1 = 0.98$ | 62.42 ± 0.18 | 70.84 ± 0.65 | 61.19 ± 0.24 | 38.10 ± 0.24 | 51.09 ± 0.15 | 53.07 ± 0.34 | 56.57 ± 0.17 |
| $\beta_1 = 0.99$ | 62.25 ± 0.01 | 70.82 ± 0.64 | 60.70 ± 0.25 | 37.41 ± 0.55 | 50.72 ± 0.43 | 52.87 ± 1.09 | 56.31 ± 0.11 |
| **Ours** | 63.29 ± 0.78 ⬤ | 75.70 ± 0.22 ⬤ | 68.41 ± 0.69 ⬤ | 42.47 ± 1.06 ⬤ | 62.46 ± 4.64 ⬤ | 64.40 ± 0.86 ⬤ | 63.36 ± 0.93 ⬤ |

Table 17: Runtime overhead of our optimal optimizers compared to the baseline optimizers with fixed hyperparameters and parameter counts for representative experiments.

| | ResNet-18 (full model) | Gemma-2B ($r = 32$ LoRA) | Gemma-2B (Full FT) | Llama-3-8B ($r = 32$ LoRA) | ViT-Base ($r = 32$ LoRA) | ViT-Large ($r = 8$ LoRA) |
|---|---|---|---|---|---|---|
| # Parameters Total ($10^6$) | 11.2 | 2,545 | 2,545 | 8,114 | 87.1 | 304 |
| # Parameters Trained ($10^6$) | 11.2 (100%) | 39.2 (1.54%) | 2,021 (79.40%) | 83.9 (1.03%) | 1.26 (1.44%) | 0.89 (0.29%) |
| Per-iteration Runtime | +4.2% | −9.4% | −17.3% | −7.7% | +0.79% | +0.30% |

# E    AUTOMATING VALIDATION-AWARE OPTIMIZER TUNING

So far, we have not specified which datasets are in use in optimizing the optimizer. This section addresses a more delicate question of how to *systematically* exploit validation sets for optimizer design. It is commonly considered bad practice to use validation sets directly in the optimization loop. Rather, they are typically used to generate subtle clues that indirectly guide engineers when making decisions about model architecture, optimizers, and associated hyperparameters. We can regard this manual tuning process as a "human-in-the-loop" optimization that fits the optimizer and hyperparameters to the validation set. In this sense, it is natural to automate this process by casting it into a mathematical optimization problem.

In this framework, we separate the gradient moments according to training and validation sets, and represent them as autocorrelation and symmetrized cross-correlation, respectively.

$$R_{\text{tr}}[k] \coloneqq \mathbb{E}[g_{\text{tr}}[n]\, g_{\text{tr}}[n-k]^\top],\ C[k] \coloneqq \mathbb{E}[g_{\text{val}}[n]\, g_{\text{tr}}[n-k]^\top],\ R_{\text{val}}[k] \coloneqq \frac{1}{2}(C[k] + C[k]^\top), \quad (26)$$

where $k \geq 0$. Note that the autocorrelation $R_{\text{tr}}[k]$ is the same as the gradient moments $R[k]$ we have used in our discussion throughout the main text. The *instantaneous validation power* is then the inner product (detailed derivation of this result is in Appendix G):

$$P_{\text{val}}(Q; n) \coloneqq \mathbb{E}\big[g_{\text{val}}[n]^\top \dot{\theta}_{\text{tr}}[n]\big] = \mathbb{E}\left[g_{\text{val}}[n]^\top \sum_{k=0}^{\infty} Q[k]\, g_{\text{tr}}[n-k]\right] = \sum_{k=0}^{\infty} \text{Tr}\big(Q[k]^\top R_{\text{val}}[k]\big) = \langle Q, R_{\text{val}}\rangle_{\mathcal{H}}.$$
(27)

The corresponding optimization problem is the recast of our original optimization problem P3 of Section 3 in terms of maximizing the instantaneous *validation* loss drop as

$$\underset{Q \in \mathcal{Q}}{\text{maximize}}\ -\dot{\mathcal{L}}_{\text{val}}\ =\ \sum_n \mathbb{E}[g_{\text{val}}[n]^\top \dot{\theta}_{\text{tr}}[n]]\ =\ \langle Q, R_{\text{val}}\rangle_{\mathcal{H}}\ =\ P_{\text{val}}(Q). \quad \text{(P4)}$$

where $\dot{\theta}_{\text{tr}}[n] = (q * g_{\text{tr}})[n]$ is the parameter velocity guided solely by the training set, just like how we typically do in machine learning. Problem P4 is mathematically equivalent to problem P3 of Section 3 in the main text but with the cross-moment $R_{\text{val}}$. This formulation turns the manual procedure of tuning optimizers from inspection of validation losses into a mathematical optimization problem that we can solve algorithmically. This approach may or may not conflict with traditional practice, potentially requiring an additional subdivision of the dataset beyond the typical training/validation split. Reaching consensus on this is beyond the scope of this work. We instead focus on the theoretical and empirical demonstration of validation-aware optimizer optimization.

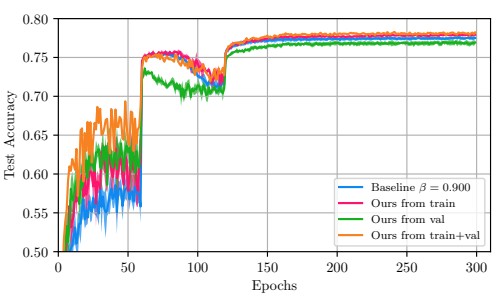

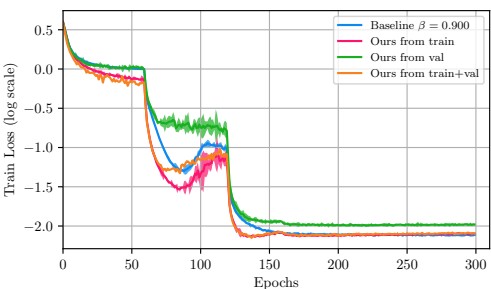

(a) Test accuracy of SGD+M optimizers.

(b) Training curve of SGD+M optimizers.

Figure 5: Demonstration of effectiveness of validation-aware design of gradient-based optimizers. The validation-*aware* optimizers achieve the highest test accuracy among all optimizers. The SGD+M optimizer is trained on the CIFAR-100 dataset (Krizhevsky, 2009) with ResNet-18 (He et al., 2016).

The next proposition shows that the validation-awareness gives better validation loss drop speed than the training-only optimizer.

**Proposition 10** (Validation optimality in power). *Let $Q_{\text{val}}^{\star} \in \arg\max_{Q \in \mathcal{Q}} \langle Q, R_{\text{val}} \rangle_{\mathcal{H}}$ and $Q_{\text{tr}}^{\star} \in \arg\max_{Q \in \mathcal{Q}} \langle Q, R_{\text{tr}} \rangle_{\mathcal{H}}$. Then $\langle Q_{\text{val}}^{\star}, R_{\text{val}} \rangle_{\mathcal{H}} \geq \langle Q_{\text{tr}}^{\star}, R_{\text{val}} \rangle_{\mathcal{H}}$.*

*Proof.* Since $Q_{\text{tr}}^{\star} \in \mathcal{Q}$, we have $\langle Q_{\text{tr}}^{\star}, R_{\text{val}} \rangle_{\mathcal{H}} \leq \max_{Q \in \mathcal{Q}} \langle Q, R_{\text{val}} \rangle_{\mathcal{H}} = \langle Q_{\text{val}}^{\star}, R_{\text{val}} \rangle_{\mathcal{H}}$. ☐

Extending our instantiation of optimal optimizers in Section 3 to the validation-aware setting requires only minimal modifications: we calculate the argmax operand $J(\beta, R)$ using the validation cross-moment $R_{\text{val}}$ instead of the training moment $R_{\text{tr}}$. The rest of the algorithm remains the same. Experiments in Figure 5, conducted under the same setting as Section 3, demonstrate the effectiveness of validation-aware optimizers. Table 18 quantitatively summarizes the results. Although validation-

Table 18: Demonstration of SGD+Momentum validation-aware training. Best baseline at $\beta = 0.9$. mean ± std. of 10 runs.

| Method | Test acc. % |
|--------|-------------|
| Best baseline ($\beta = 0.9$) | 77.57 ± 0.09 🔴 |
| **Ours: val only** | 77.10 ± 0.29 |
| **Ours: train only** | 78.06 ± 0.07 ⚪ |
| **Ours: train+val** | 78.30 ± 0.12 🟡 |

*only* optimizers perform worse than training-only optimizers, the validation-*aware* optimizers, which use *both* training and validation sets to inquire the optimal optimizer and its hyperparameters, achieve better test accuracy compared to the training-only optimizers. We strongly emphasize that, during the experiments, we do not use the validation set gradients to *update the model parameters*; they are only used to *select the optimizer hyperparameters*, following the traditional practice of manual engineering. What is the difference? If we were to deviate from this—for instance, by accidentally or deliberately leaking information from the validation set gradients into the optimizer input—the validation accuracy would immediately saturate to near 100%, much like how the training accuracy saturates in the standard training-only setting. Therefore, the poorer results of the validation-only optimizer are an indicator that no such leakage occurs. During the experiments, we zero out the gradients of the validation set before and after changing the optimizer hyperparameters to avoid any leakage.

Although, from Proposition 10 and from our experiment, we know that validation-aware optimal optimizer gives instantaneously favorable choice over the training-only optimizer, a deeper analysis is required to understand and measure the exact effect of validation-awareness. We leave this as future work and resort to empirical evaluation for now. This concludes our theoretical and empirical demonstration of automatic, validation-aware design of gradient-based optimizers.

# F    REVERSE ENGINEERING COMMON OPTIMIZERS

In the main manuscript, we have provided the optimal form of two of the most widely used optimizers: SGD with momentum and Adam (Kingma & Ba, 2015), and postponed the extension to other optimizers to this appendix, leaving only Table 7 for the full list of optimizers in the main text.

This section completes the reverse engineering of various optimizers under our greedy paradigm. The derivation follows the same structure as the main manuscript, by extensively using the facts delivered in Theorems 1 and 3, and Lemma 4. This reverse engineering not only allows us to find out hidden design principles of these optimizers, but also have these optimizers registered in a unified framework, suggesting a systematic way to derive their *optimal hyperparameters*.

## F.1 SUMMARY OF THE RESULTS

Our results encompass the following optimizers: SGD, Adam/AdamW (Kingma & Ba, 2015; Loshchilov & Hutter, 2019), natural gradient descent (NGD) (Amari, 1998), Gauss-Newton, K-FAC (Martens & Grosse, 2015), Shampoo (Gupta et al., 2018), and Muon (Liu et al., 2025). In the following, we will present the results by first giving the compact summary tables, and then present each optimizer in detail, with at least one corollary that mathematically registers each optimizer as a special case of our framework. The proofs are very similar to those of Corollaries 5 and 6 for SGD with momentum and Adam, respectively, in the main text.

In the previous section, we have derived various types of optimizers from our convex optimization framework. We can now register various optimizers under a single unified table, as shown in Table 7. Each optimizer corresponds to a specific choice of moment matrix $M$, budget constraint $\mathcal{Q}$, and resulting equalizer $Q$. "Param restrict" rows are feasible points in the convex programs that can either be kept and fitted to target moments, or replaced with full closed-form solutions.

## F.2 GRADIENT DESCENT

Gradient descent is a special case of Corollary 5. We present the result for the completeness of the framework. Since there is no momentum hyperparameter associated with gradient descent, the theorem only shows the connection between the budget (the trust region size) and the learning rate. In other words, the budget $B$ can be thought of as a theory-friendly notation for the learning rate $\eta$. In practical scenarios, we usually want to increase this value as much as possible to achieve the best performance and speed. Comparing this with Corollary 5, we get an insight of how the momentum hyperparameter is introduced to the stateless optimal optimizer, as well as how the backbone structure of the proofs for the optimal optimizers in this paper is constructed.

**Corollary 11** (Instantaneously optimal GD). *Consider the general family of* Frobenius trust regions $\mathcal{Q}_{\mathrm{F}}(B)$ *and a cone* $\mathcal{C}_{0\mathrm{p}}$ *of memoryless isotropic optimizers, i.e., a scaled identity matrix:*

$$\mathcal{Q}_{\mathrm{F}}(B) := \{Q : \|Q\|_{\mathcal{H}} \leq \sqrt{B}\}, \quad \mathcal{C}_{0\mathrm{p}} := \{Q[n] = \eta I \delta[n] : \eta \geq 0\}. \tag{28}$$

*Given current gradient $g[n]$, the optimal solution of problem P3 under the trust region $\mathcal{Q}_{\mathrm{F}}(B) \cap \mathcal{C}_{0\mathrm{p}}$ is a simple* gradient descent *optimizer with* optimal hyperparameter:

$$\eta^{\star} = \sqrt{\frac{B}{d}}, \tag{29}$$

*where $d$ is the dimension of the parameter space.*

*Proof of Corollary 11.* We work in the impulse-space as defined in Section 3, i.e., a Hilbert space $(\mathcal{H}, \langle \cdot, \cdot \rangle_{\mathcal{H}})$ of causal LTI filters with matrix impulse response $\{q_k\}_{k \geq 0}$ with a Hilbert norm $\|Q\|_{\mathcal{H}}^2 = \sum_{k=0}^{\infty} \mathrm{Tr}(q_k^{\top} q_k)$, where $k$ is the time index.

The memoryless isotropic family forms a feasible submanifold within the trust region constraint $\mathcal{Q}_{\mathrm{F}}(B) = \{Q : \|Q\|_{\mathcal{H}} \leq \sqrt{B}\}$ and the causality cone $\mathcal{C}$.

By Lemma 4, the optimal learning rate corresponds to the projection of the unconstrained optimizer $Q^{\star}$ onto this submanifold, which reduces to maximizing the inner product $\langle Q_{\mathrm{SGD}}, R \rangle_{\mathcal{H}}$ subject to the trust region constraint.

*Step 1 — Fit the cone to the trust region.* The impulse response is $q_0 = \eta I$ and $q_k = 0$ for $k > 0$. By definition,

$$\|Q_{\mathrm{SGD}}\|_{\mathcal{H}}^2 = \mathrm{Tr}((\eta I)^{\top}(\eta I)) = \eta^2 d. \tag{30}$$

The trust region constraint $\|Q_{\text{SGD}}\|_{\mathcal{H}} \leq \sqrt{B}$ imposes

$$\eta \leq \sqrt{\frac{B}{d}}. \tag{31}$$

*Step 2 — Get the learning power objective.* For the instantaneous gradient moment $R[0] = gg^\top$ and $R[n] = 0$ for $n > 0$, the inner product is

$$P_{\text{SGD}} = \langle Q_{\text{SGD}}, R \rangle_{\mathcal{H}} = \text{Tr}((\eta I)^\top (gg^\top)) = \eta \|g\|_2^2. \tag{32}$$

*Step 3 — Derive the optimal hyperparameters.* The objective is linear in $\eta$ while the constraint is quadratic, so the maximizer saturates the trust region $\|Q_{\text{SGD}}\|_{\mathcal{H}} = \sqrt{B}$. This gives the corresponding learning rate:

$$\eta^\star = \sqrt{\frac{B}{d}}. \tag{33}$$

Plugging this to the cone $\mathcal{C}_{0\text{p}}$, we get the optimal optimizer $Q^\star = \eta^\star I$, completing the proof. $\qquad\square$

### F.3   PRECONDITIONED GRADIENT DESCENT

Preconditioned gradient descent extends standard gradient descent with a fixed positive definite matrix $P \succ 0$. This restricts the direction of update to the particular direction specified by $P$. The matrix $P$ can also be viewed as the principal direction of an ellipsoid that circumscribes the (hard) trust region constraint. Standard gradient descent is a special case when $P = I$. The theorem shows that the resulting optimizer is a scaled version of the positive definite matrix $P$.

Further, we can extend this result to various different preconditioners $P$, as shown in the sections after this, e.g., Hessian $P = H^{-1}$ for Newton's method in Section F.4 and Fisher information matrix $P = F^{-1}$ for natural gradient descent in Section F.5. Stateless optimizers can also be derived from this proof by setting the cone as $\mathcal{C}_{0\text{p}}(P) = \{Q[n] = \eta P \delta[n] : \eta \geq 0\}$.

**Corollary 12** (Instantaneously optimal preconditioned gradient descent)**.** *Consider the general family of elliptic trust regions $\mathcal{Q}(B, P)$ and a cone $\mathcal{C}_{1\text{p}}(P)$ of 1-pole optimizers with a fixed positive definite matrix $P \succ 0$:*

$$\mathcal{Q}(B, P) := \left\{ Q : \|Q\|_{\mathcal{H}, P^{-1}} = \sum_{k=0}^{\infty} \text{Tr}(q_k^\top P^{-1} q_k) \leq \sqrt{B} \right\}, \tag{34}$$

$$\mathcal{C}_{1\text{p}}(P) := \{ Q_{\eta, \beta, P} = \eta (1 - \beta) \beta^n P : \eta \geq 0, 0 < \beta < 1 \}. \tag{35}$$

*Given gradients $g[n]$, define the standard momentum $m_\beta$ and $P$-weighted momentum $u_\beta$ as follows:*

$$m_\beta[n] := \sum_{k=0}^{\infty} \beta^k g[n-k], \quad u_\beta[n] := P m_\beta[n]. \tag{36}$$

*at time $n$, respectively. The optimal solution of problem P3 under the trust region $\mathcal{Q}(B, P) \cap \mathcal{C}_{1\text{p}}(P)$ is a* preconditioned gradient descent *optimizer with* optimal hyperparameters*:*

$$\beta^\star[n] = \arg\max_{\beta \in (0,1)} \sqrt{1 - \beta^2} \, \mathbb{E}[g[n]^\top u_\beta[n]], \quad \eta^\star = \left( \frac{B(1+\beta^\star)}{(1-\beta^\star)\text{Tr}(P)} \right)^{1/2}, \tag{37}$$

*where $d$ is the dimension of the parameter space. Therefore, we update parameters by:*

$$\Delta\theta = -\eta^\star u_{\beta^\star} = -\eta^\star P m_{\beta^\star}. \tag{38}$$

*Proof of Corollary 12.* We work in the impulse-space as defined in Section 3, i.e., a Hilbert space $(\mathcal{H}, \langle \cdot, \cdot \rangle_{\mathcal{H}})$ of causal LTI filters with matrix impulse response $\{q_k\}_{k \geq 0}$ with elliptic $P^{-1}$-weighted norm $\|Q\|_{\mathcal{H}, P^{-1}}^2 = \sum_{k=0}^{\infty} \text{Tr}(q_k^\top P^{-1} q_k)$.

The 1-pole $P$-weighted family forms a feasible submanifold within the elliptic trust region constraint $\mathcal{Q}(B, P)$ and the causality cone $\mathcal{C}$.

By Lemma 4, the optimal preconditioned gradient descent parameters correspond to the projection of the unconstrained optimizer $Q^\star$ onto this submanifold, which reduces to maximizing the inner product $\langle Q_{\text{PGD}}, R \rangle_{\mathcal{H}}$ subject to the elliptic trust region constraint.

*Step 1 — Norm of the 1-pole P-weighted optimizer.* The impulse response is $q_k = \eta(1-\beta)\beta^k P$. By definition,

$$\|Q_{\text{PGD}}\|^2_{\mathcal{H},P^{-1}} = \sum_{k=0}^{\infty} \text{Tr}((\eta(1-\beta)\beta^k P)^\top P^{-1} (\eta(1-\beta)\beta^k P)) \tag{39}$$

$$= \eta^2(1-\beta)^2 \sum_{k=0}^{\infty} \beta^{2k} \text{Tr}(P) = \eta^2 \frac{(1-\beta)^2}{1-\beta^2} \text{Tr}(P). \tag{40}$$

The elliptic trust region constraint $\|Q_{\text{PGD}}\|_{\mathcal{H},P^{-1}} \leq \sqrt{B}$ imposes

$$\eta \leq \sqrt{B} \left( \frac{(1-\beta)^2}{1-\beta^2} \text{Tr}(P) \right)^{-1/2}. \tag{41}$$

*Step 2 — Alignment with the moment operator.* The inner product with the gradient moment $R[k] = \mathbb{E}[g[n]g[n-k]^\top]$ is

$$\langle Q_{\text{PGD}}, R \rangle_{\mathcal{H}} = \sum_{k=0}^{\infty} \text{Tr}((\eta(1-\beta)\beta^k P)^\top R[k]) \tag{42}$$

$$= \eta(1-\beta) \sum_{k=0}^{\infty} \beta^k \text{Tr}(P^\top R[k]) = \eta(1-\beta) \sum_{k=0}^{\infty} \beta^k T_k, \tag{43}$$

where $T_k := \text{Tr}(P^\top R[k])$.

*Step 3 — Reduce to 1-D search; saturate trust region.* For fixed $\beta$, the objective is linear in $\eta$ while the constraint is quadratic, so the maximizer saturates the elliptic trust region. The trust region-normalized gain is

$$J(\beta) := \frac{\langle Q_{\text{PGD}}, R \rangle_{\mathcal{H}}}{\|Q_{\text{PGD}}\|_{\mathcal{H},P^{-1}}} = \frac{\sqrt{1-\beta^2}}{\sqrt{\text{Tr}(P)}} \sum_{k=0}^{\infty} \beta^k T_k. \tag{44}$$

*Step 4 — Solving for streaming gradients.* From $R_k[n] = \mathbb{E}[g[n]g[n-k]^\top]$ and $T_k[n] = \text{Tr}(P^\top R_k[n])$, we have

$$J(\beta; n) = \frac{\sqrt{1-\beta^2}}{\sqrt{\text{Tr}(P)}} \sum_{k=0}^{\infty} \beta^k \text{Tr}(P^\top \mathbb{E}[g[n]g[n-k]^\top]) \tag{45}$$

$$= \frac{\sqrt{1-\beta^2}}{\sqrt{\text{Tr}(P)}} \mathbb{E}[g[n]^\top P \sum_{k=0}^{\infty} \beta^k g[n-k]] \tag{46}$$

$$= \frac{\sqrt{1-\beta^2}}{\sqrt{\text{Tr}(P)}} \mathbb{E}[g[n]^\top P m_\beta[n]], \tag{47}$$

where $m_\beta[n] = \sum_{k=0}^{\infty} \beta^k g[n-k]$ is the standard momentum, which can be calculated sequentially as:

$$m_\beta[n] = g[n] + \beta m_\beta[n-1]. \tag{48}$$

Therefore, $\beta^\star[n] = \arg\max_{0<\beta<1} J(\beta; n)$ and $\eta^\star$ saturates the trust region constraint. $\qquad\square$

Keen readers will notice that the above proof is similar to the proof of optimal Adam in Corollary 6. Only the preconditioner $P$ is replaced by the sum of inverse per-parameter costs.

### F.4 NEWTON'S METHOD

As we have discussed in the last section, we can derive the Newton's method with momentum variable by changing the preconditioner $P$ to the inverse of the Hessian matrix $H^{-1}$ in the result of Corollary 12, assuming that the Hessian matrix $H \succ 0$ is positive definite and changing slowly compared to the gradient updates. We can also arbitrarily control the time-dependent memory components by changing the projection cone.

**Corollary 13** (Instantaneously optimal Newton's method (slow curvature change)). *Consider the general family of* curvature-aware trust regions $\mathcal{Q}(B, H)$ *and a cone* $\mathcal{C}_{1\mathrm{p}}(H)$ *of 1-pole Hessian-scaled optimizers:*

$$\mathcal{Q}(B, H) := \{Q : \|Q\|_{\mathcal{H}, H} \leq \sqrt{B}\}, \tag{49}$$

$$\mathcal{C}_{1\mathrm{p}}(H) := \{Q_{\eta, \beta, H} = \eta(1 - \beta)\beta^n H^{-1} : \eta \geq 0, 0 < \beta < 1\}. \tag{50}$$

*where $H \succ 0$ is the Hessian matrix. Given gradients $g[n]$, define $m_\beta[n] := \sum_{k=0}^{\infty} \beta^k g[n - k]$ as the standard momentum at time $n$ with momentum parameter $\beta$. The optimal solution of problem P3 under the trust region $\mathcal{Q}(B, H) \cap \mathcal{C}_{1\mathrm{p}}(H)$ is* Gaussian-Newton *with optimal momentum:*

$$\beta^\star[n] = \arg\max_{\beta \in (0,1)} \sqrt{1 - \beta^2} \, \mathbb{E}[g[n]^\top H^{-1} m_\beta[n]], \quad \eta^\star = \left(\frac{B(1 + \beta^\star)}{(1 - \beta^\star) \operatorname{Tr}(H^{-1})}\right)^{1/2}, \tag{51}$$

*where the parameter update is $\Delta\theta = -\eta^\star H^{-1} m_{\beta^\star}$.*

*Proof of Corollary 13.* Setting $P = H^{-1}$ in the result of Corollary 12, we have the result. □

### F.5 NATURAL GRADIENT DESCENT

As we have discussed in the previous section, we can derive the natural gradient descent with momentum variable by changing the preconditioner $P$ to the inverse of the Fisher information matrix $F^{-1}$ in the result of Corollary 12, assuming that the Fisher information matrix $F \succ 0$ is positive definite and changing slowly compared to the gradient updates. We can also arbitrarily control the time-dependent memory components by changing the projection cone.

**Corollary 14** (Instantaneously optimal natural gradient descent (slow curvature change)). *Consider the general family of* Fisher information trust regions $\mathcal{Q}(B, F)$ *and a cone* $\mathcal{C}_{1\mathrm{p}}(F)$ *of 1-pole Fisher-scaled optimizers:*

$$\mathcal{Q}(B, F) := \{Q : \|Q\|_{\mathcal{H}, F} \leq \sqrt{B}\}, \tag{52}$$

$$\mathcal{C}_{1\mathrm{p}}(F) := \{Q_{\eta, \beta, F} = \eta(1 - \beta)\beta^n F^{-1} : \eta \geq 0, 0 < \beta < 1\}. \tag{53}$$

*Given gradients $g[n]$, define $m_\beta[n] := \sum_{k=0}^{\infty} \beta^k g[n - k]$ as the standard momentum at time $n$ with momentum parameter $\beta$. The optimal solution of problem P3 under the trust region $\mathcal{Q}(B, F) \cap \mathcal{C}_{1\mathrm{p}}(F)$ is* natural gradient descent *with optimal momentum:*

$$\beta^\star[n] = \arg\max_{\beta \in (0,1)} \sqrt{1 - \beta^2} \, \mathbb{E}[g[n]^\top F^{-1} m_\beta[n]], \quad \eta^\star = \left(\frac{B(1 + \beta^\star)}{(1 - \beta^\star) \operatorname{Tr}(F^{-1})}\right)^{1/2}, \tag{54}$$

*where the parameter update is $\Delta\theta = -\eta^\star F^{-1} m_{\beta^\star}$.*

*Proof of Corollary 14.* Setting $P = F^{-1}$ in the result of Corollary 12, we have the result. □

#### F.5.1 K-FAC

K-FAC (Martens & Grosse, 2015) is a block-diagonal Fisher-scaled optimizer that approximates the natural gradient descent by approximating the Fisher information matrix as a block-diagonal matrix. We assume that the Fisher information matrices are positive definite and changing slowly compared to the gradient updates, so we can approximate its inverse as a constant. Like the full NGD, we can derive K-FAC using our framework, extending the result of Corollary 12 to the block-diagonal case.

**Corollary 15** (Instantaneously optimal K-FAC). *Consider the general family of* block-diagonal *Fisher trust regions* $\mathcal{Q}_{block}(B, \{F_\ell\})$ *and a cone* $\mathcal{C}_{1p}(\{F_\ell\})$ *of 1-pole block-diagonal Fisher-scaled optimizers:*

$$\mathcal{Q}_{block}(B, \{F_\ell\}) := \left\{ Q : \|Q\|_{\mathcal{H},block} = \sum_{k=0}^\infty \sum_\ell \operatorname{Tr}(Q_\ell[k]^\top F_\ell Q_\ell[k]) \leq B \right\}, \tag{55}$$

$$\mathcal{C}_{1p}(\{F_\ell\}) := \{ Q_{\eta,\beta,\{F_\ell\}} = \eta(1-\beta)\beta^n \operatorname{bdiag}(F_\ell^{-1}) : \eta \geq 0, 0 < \beta < 1 \}, \tag{56}$$

*where* bdiag *is the block-diagonal operator that forms a block-diagonal matrix from the set of blocks* $\{F_\ell^{-1}\}$. *Given gradients* $g[n]$, *define* $m_\beta[n] := \sum_{k=0}^\infty \beta^k g[n-k]$ *as the standard momentum at time* $n$ *with momentum parameter* $\beta$. *The optimal solution of problem P3 under the trust region* $\mathcal{Q}_{block}(B, \{F_\ell\}) \cap \mathcal{C}_{1p}(\{F_\ell\})$ *is* K-FAC *with optimal momentum:*

$$\beta^\star[n] = \arg \max_{\beta \in (0,1)} \sqrt{1-\beta^2} \, \mathbb{E}[g[n]^\top \operatorname{bdiag}(F_\ell^{-1}) \, m_\beta[n]], \quad \eta^\star = \left( \frac{B(1+\beta^\star)}{(1-\beta^\star) \sum_\ell \operatorname{Tr}(F_\ell^{-1})} \right)^{1/2}, \tag{57}$$

*where the parameter update is* $\Delta\theta = -\eta^\star \operatorname{bdiag}(F_\ell^{-1}) \, m_{\beta^\star}$.

*Proof of Corollary 15.* Setting $P = \operatorname{bdiag}(F_\ell^{-1})$ in the preconditioned gradient descent result in Corollary 12, we have the result. $\qquad\square$

### F.6 SHAMPOO

Shampoo (Gupta et al., 2018) is a Kronecker-factored optimizer that approximates arbitrary color gradient descent (with or without momentum) by approximating the second moment matrix $P$ as a Kronecker product of *mode-wise* second moment matrices. Like all above, we can derive Shampoo using our framework, assuming that the second moment matrices are positive definite and changing slowly compared to the gradient updates.

**Corollary 16** (Instantaneously optimal Shampoo). *Consider weight tensors* $\theta \in \mathbb{R}^{d_1 \times d_2 \times \cdots \times d_k}$ *with total dimension* $d = \prod_i d_i$ *and define mode-wise second moment matrices* $G_i \in \mathbb{R}^{d_i \times d_i}$ *for each mode* $i$. *Consider the general family of* Kronecker-factored *trust regions* $\mathcal{Q}_{Kron}(B, \{G_i\})$ *and a cone* $\mathcal{C}_{1p}(\{G_i\})$ *of 1-pole Kronecker-scaled optimizers:*

$$\mathcal{Q}_{Kron}(B, \{G_i\}) := \left\{ Q : \|Q\|_{\mathcal{H},Kron} = \sum_{k=0}^\infty \operatorname{Tr}\left( Q[k]^\top \left( \bigotimes_i G_i \right) Q[k] \right) \leq B \right\}, \tag{58}$$

$$\mathcal{C}_{1p}(\{G_i\}) := \left\{ Q_{\eta,\beta,\{G_i\}} = \eta(1-\beta)\beta^n \bigotimes_i G_i^{-1/2} : \eta \geq 0, 0 < \beta < 1 \right\}. \tag{59}$$

*Given gradients* $g[n]$, *define* $m_\beta[n] := \sum_{k=0}^\infty \beta^k g[n-k]$ *as the standard momentum at time* $n$ *with momentum parameter* $\beta$. *The optimal solution of problem P3 under the trust region* $\mathcal{Q}_{Kron}(B, \{G_i\}) \cap \mathcal{C}_{1p}(\{G_i\})$ *is* Shampoo *with optimal momentum:*

$$\beta^\star[n] = \arg \max_{\beta \in (0,1)} \sqrt{1-\beta^2} \, \mathbb{E}\left[ g[n]^\top \bigotimes_i G_i^{-1/2} \, m_\beta[n] \right], \quad \eta^\star = \left( \frac{B(1+\beta^\star)}{(1-\beta^\star)d} \right)^{1/2}, \tag{60}$$

*where the parameter update is* $\Delta\theta = -\eta^\star \bigotimes_i G_i^{-1/2} \, m_{\beta^\star}$.

*Proof of Corollary 16.* Setting $P = \bigotimes_i G_i^{-1/2}$ in the result of Corollary 12, we have the result. In addition, we have the impulse response $q_k = \eta(1-\beta)\beta^k \bigotimes_i G_i^{-1/2}$. By definition,

$$\|Q_{\text{Shampoo}}\|_{\mathcal{H},\text{Kron}}^2 = \sum_{k=0}^{\infty} \text{Tr}\left(\left(\eta(1-\beta)\beta^k \bigotimes_i G_i^{-1/2}\right)^{\top} \left(\bigotimes_i G_i\right)\left(\eta(1-\beta)\beta^k \bigotimes_i G_i^{-1/2}\right)\right) \tag{61}$$

$$= \eta^2(1-\beta)^2 \sum_{k=0}^{\infty} \beta^{2k} \text{Tr}\left(\bigotimes_i (G_i^{-1/2} G_i G_i^{-1/2})\right) \tag{62}$$

$$= \eta^2(1-\beta)^2 \sum_{k=0}^{\infty} \beta^{2k} \text{Tr}\left(\bigotimes_i I_{d_i}\right) \tag{63}$$

$$= \eta^2 \frac{(1-\beta)^2}{1-\beta^2} d, \tag{64}$$

where $d = \prod_i d_i$ is the total dimension. The Kronecker-factored trust region constraint $\|Q_{\text{Shampoo}}\|_{\mathcal{H},\text{Kron}} \le \sqrt{B}$ imposes

$$\eta \le \sqrt{B}\left(\frac{(1-\beta)^2}{1-\beta^2} d\right)^{-1/2}. \tag{65}$$

$\square$

## F.7 ADAPTIVE-MOMENT FAMILY

The adaptive-moment family includes AdaGrad (Duchi et al., 2011; Agarwal et al., 2019), RMSProp (Tieleman & Hinton, 2012), and Adam (Kingma & Ba, 2015). We have already derived Adam in Corollary 6 in the main text. This section completes the derivation of its family, which share the same proof structure. We assume that the second moment matrices change slower than the gradients, which is typical in practice.

### F.7.1 ADAGRAD

Regarding AdaGrad (Duchi et al., 2011; Agarwal et al., 2019), we have the expensive full-matrix version and the relatively cheap diagonal version.

**Corollary 17** (Instantaneously optimal full-matrix AdaGrad with momentum)**.** *Consider the general family of full-matrix trust regions* $\mathcal{Q}(B,G)$ *and a cone* $\mathcal{C}_{1\text{p}}(G)$ *of full-matrix 1-pole optimizers with second-moment matrix* $G$:

$$\mathcal{Q}(B,G) := \{Q : \text{Tr}(G^{1/2} \sum_{k \ge 0} Q[k]^{\top} Q[k] G^{1/2}) \le B\}, \tag{66}$$

$$\mathcal{C}_{1\text{p}}(G) := \{Q_{\eta,\beta}[k] = \eta(1-\beta)\beta^k G^{-1/2} : \eta \ge 0, 0 < \beta < 1\}. \tag{67}$$

*Given gradients* $g[t]$, *maintain the cumulative second-moment matrix*

$$G[t] := \epsilon I + \sum_{s \le t} g[s]g[s]^{\top} \succ 0, \tag{68}$$

*with regularization* $\epsilon > 0$. *Define* $m_\beta[n] := \sum_{k=0}^{\infty} \beta^k g[n-k]$ *as the standard momentum at time* $n$ *with momentum parameter* $\beta$. *The optimal solution of problem P3 under the trust region* $\mathcal{Q}(B,G) \cap \mathcal{C}_{1\text{p}}(G)$ *is* full-matrix AdaGrad with optimal momentum *(classical full-matrix AdaGrad corresponds to* $\beta = 0$*) with* optimal momentum*:*

$$\beta^\star[n] = \arg\max_{\beta \in (0,1)} \sqrt{1-\beta^2}\, \mathbb{E}[g[n]^{\top} G^{-1/2} m_\beta[n]], \quad \eta^\star = \left(\frac{B\,(1+\beta^\star)}{(1-\beta^\star)\,\text{Tr}(G^{-1})}\right)^{1/2}, \tag{69}$$

*where the parameter update is* $\Delta\theta = -\eta^\star G^{-1/2} m_{\beta^\star}$.

*Proof of Corollary 17.* Due to the commutativity of multiplication and the linearity of the trace operator, we have:

$$\text{Tr}\left(G^{1/2}\sum_{k\geq 0}Q[k]^\top Q[k]G^{1/2}\right) = \sum_{k\geq 0}\text{Tr}(Q[k]^\top GQ[k]) = \|Q\|_{\mathcal{H},G}. \tag{70}$$

By plugging in $P = G$ in the result of Corollary 12, we have the exact same structure. This gives the desired results. □

**Corollary 18** (Instantaneously optimal diagonal AdaGrad with momentum)**.** *Consider the general family of diagonal trust regions $\mathcal{Q}_D(B, c)$ and a cone $\mathcal{C}_{1p}(c)$ of diagonal 1-pole optimizers with coordinate-wise costs $c$:*

$$\mathcal{Q}_D(B, c) := \{\text{diag}(q_{j,k}) : \sum_j c_j \sum_{k\geq 0}|q_{j,k}|^2 \leq B\}, \tag{71}$$

$$\mathcal{C}_{1p}(c) := \{Q_{\eta,\beta}[k] = \eta(1-\beta)\beta^k\,\text{diag}(1/c_j) : \eta \geq 0, 0 < \beta < 1\}. \tag{72}$$

*Given gradients $g[t]$, maintain the cumulative second-moment $v_j[t] := \epsilon + \sum_{s\leq t}g_j[s]^2 > 0$ with regularization $\epsilon > 0$, and define the coordinate-wise costs $c_j := v_j[t]^{1/2}$. Define $u_{\beta,diag}[n] := \text{diag}(1/c_j)\sum_{k=0}^\infty \beta^k g[n-k]$ as the diagonal-scaled update at time $n$ with momentum parameter $\beta$. The optimal solution of problem P3 under the trust region $\mathcal{Q}_D(B, c) \cap \mathcal{C}_{1p}(c)$ is AdaGrad with optimal momentum (classical AdaGrad corresponds to $\beta = 0$) with optimal momentum:*

$$\beta^\star[n] = \arg\max_{\beta\in(0,1)}a(\beta)\,\mathbb{E}[g[n]^\top u_{\beta,diag}[n]], \quad \eta^\star = \sqrt{B}\cdot a(\beta^\star), \tag{73}$$

*where $a(\beta) := \sqrt{(1+\beta)/((1-\beta)\sum_j 1/c_j)}$ is the normalization factor and the optimizer is $Q^\star = \eta^\star(1-\beta^\star)\sum_{k=0}^\infty \beta^{\star k}\,\text{diag}(1/c_j)$.*

*Proof of Corollary 18.* We work in the impulse-space as defined in Section 3, i.e., a diagonal Hilbert space $(\mathcal{H}_D, \langle\cdot,\cdot\rangle_{\mathcal{H}_D})$ of diagonal causal LTI filters with weighted norm $\|Q\|_{\mathcal{H}_D}^2 = \sum_j c_j \sum_{k=0}^\infty |q_{j,k}|^2$ where $c_j = v_j^{1/2}$ are the coordinate-wise costs.

The 1-pole diagonal family forms a feasible submanifold within the diagonal weighted trust region constraint $\mathcal{Q}_D(B, c)$ and the causality cone $\mathcal{C}$.

By Lemma 4, the optimal AdaGrad parameters correspond to the projection of the unconstrained optimizer $Q^\star$ onto this submanifold, which reduces to maximizing the inner product $\langle Q_{\text{AdaGrad}}, R\rangle_{\mathcal{H}_D}$ subject to the diagonal weighted trust region constraint.

*Step 1 — Norm of the 1-pole diagonal optimizer.* The impulse response is $q_{j,k} = \eta(1-\beta)\beta^k/c_j$. By definition,

$$\|Q_{\text{AdaGrad}}\|_{\mathcal{H}_D}^2 = \sum_j c_j \sum_{k=0}^\infty (\eta(1-\beta)\beta^k/c_j)^2 \tag{74}$$

$$= \eta^2\frac{(1-\beta)^2}{1-\beta^2}\sum_j\frac{1}{c_j} \tag{75}$$

$$= \eta^2\frac{(1-\beta)^2}{1+\beta}W, \tag{76}$$

where $W := \sum_j 1/c_j$ is the normalization factor.

*Step 2 — Alignment with the moment operator.* The inner product with the diagonal gradient moment $R[k] = \text{diag}(\mathbb{E}[g_j[n]g_j[n-k]])$ is

$$\langle Q_{\text{AdaGrad}}, R\rangle_{\mathcal{H}_D} = \eta(1-\beta)\sum_{k=0}^\infty \beta^k\sum_j\frac{\mathbb{E}[g_j[n]g_j[n-k]]}{c_j} \tag{77}$$

$$= \eta(1-\beta)\sum_{k=0}^\infty \beta^k T_k, \tag{78}$$

where $T_k := \sum_j \mathbb{E}[g_j[n]g_j[n-k]]/c_j$.

*Step 3 — Reduce to 1-D search; saturate trust region.* For fixed $\beta$, the objective is linear in $\eta$ while the constraint is quadratic, so the maximizer saturates the trust region. The trust region-normalized gain is

$$J(\beta) := \frac{\langle Q_{\text{AdaGrad}}, R \rangle_{\mathcal{H}_D}}{\|Q_{\text{AdaGrad}}\|_{\mathcal{H}_D}} = \sqrt{\frac{1+\beta}{(1-\beta)W}} \sum_{k=0}^{\infty} \beta^k T_k. \tag{79}$$

*Step 4 — Solving for streaming gradients.* From $R_k[n] = \text{diag}(\mathbb{E}[g_j[n]g_j[n-k]])$ and $T_k[n] = \sum_j \mathbb{E}[g_j[n]g_j[n-k]]/c_j$, we have

$$J(\beta; n) = \sqrt{\frac{1+\beta}{(1-\beta)W}} \sum_{k=0}^{\infty} \beta^k \sum_j \frac{\mathbb{E}[g_j[n]g_j[n-k]]}{c_j} \tag{80}$$

$$= \sqrt{\frac{1+\beta}{(1-\beta)W}} \mathbb{E}\left[ \sum_j \frac{g_j[n]}{c_j} \sum_{k=0}^{\infty} \beta^k g_j[n-k] \right] \tag{81}$$

$$= \sqrt{\frac{1+\beta}{(1-\beta)W}} \mathbb{E}[g[n]^\top u_{\beta,\text{diag}}[n]], \tag{82}$$

where $u_{\beta,\text{diag}}[n] = \text{diag}(1/c_j) \sum_{k=0}^{\infty} \beta^k g[n-k]$ is the diagonal-scaled update.

Therefore, $\beta^\star[n] = \arg\max_{0<\beta<1} J(\beta; n)$ and $\eta^\star$ saturates the trust region constraint. $\qquad\square$

### F.7.2 RMSProp

The RMSProp optimizer (Tieleman & Hinton, 2012) can also be registered into our framework as a diagonal optimizer family, similarly to AdaGrad in Corollary 18. This is interesting because RMSProp does not use an explicit first-order moment, unlike many of the optimizers we have studied.

**Corollary 19** (Instantaneously optimal RMSProp-style scaling)**.** *Consider the general family of diagonal trust regions $\mathcal{Q}_D(B, c)$ and a cone $\mathcal{C}_{\text{RMSProp}}(c)$ of instantaneous diagonal optimizers with coordinate-wise costs $c$:*

$$\mathcal{Q}_D(B, c) := \{\text{diag}(q_j) : \sum_j c_j |q_j|^2 \le B\}, \quad \mathcal{C}_{\text{RMSProp}}(c) := \{Q_\eta = \eta \, \text{diag}(1/c_j) : \eta \ge 0\}. \tag{83}$$

*Given gradients $g[n]$, maintain the second-moment EMA*

$$v_j[n] := \beta v_j[n-1] + (1-\beta)g_j[n]^2 > 0, \qquad \beta \in (0, 1), \tag{84}$$

*and define the coordinate-wise costs $c_j(\beta; n) := v_j[n]^{1/2}$. The optimal solution of problem P3 under the trust region $\mathcal{Q}_D(B, c) \cap \mathcal{C}_{\text{RMSProp}}(c)$ is an* RMSProp-style diagonal scaling *with instantaneously optimal hyperparameters:*

$$\beta^\star[n] \in \arg\max_{\beta \in (0,1)} \frac{\sum_j c_j(\beta; n)}{\sqrt{\sum_j 1/c_j(\beta; n)}}, \qquad \eta^\star[n] = \sqrt{\frac{B}{\sum_j 1/c_j(\beta^\star[n]; n)}}, \tag{85}$$

*and the corresponding optimizer is $Q^\star[n] = \eta^\star[n] \, \text{diag}(1/c_j(\beta^\star[n]; n))$, which yields the RMSProp-style update*

$$\Delta\theta_j[n] = -\eta^\star[n] \frac{g_j[n]}{c_j(\beta^\star[n]; n)} = -\eta^\star[n] \frac{g_j[n]}{v_j[n]^{1/2}}. \tag{86}$$

*Proof of Corollary 19.* We work in the parameter space as defined in Section 3, in a diagonal space of instantaneous filters $Q = \text{diag}(q_j)$ endowed with the weighted norm

$$\|Q\|_{\mathcal{H}_D}^2 := \sum_j c_j(\beta; n) |q_j|^2, \qquad c_j(\beta; n) = v_j[n]^{1/2}. \tag{87}$$

The RMSProp diagonal family $\mathcal{C}_{\text{RMSProp}}(c)$ forms a feasible one-dimensional submanifold within the diagonal weighted trust region constraint $\mathcal{Q}_{\text{D}}(B, c)$. At time $n$, the learning power under $Q$ is

$$P(Q; n) := \mathbb{E}\big[g[n]^\top Q\, g[n]\big] = \sum_j q_j\, \mathbb{E}[g_j[n]^2] \approx \sum_j q_j\, v_j(\beta; n), \tag{88}$$

where we approximate the second moments $\mathbb{E}[g_j[n]^2]$ by the EMA $v_j(\beta; n)$.

*Step 1 — Norm of the instantaneous diagonal optimizer.* Within $\mathcal{C}_{\text{RMSProp}}(c)$, the diagonal is

$$q_j = \frac{\eta}{c_j(\beta; n)}. \tag{89}$$

By definition,

$$\|Q_{\text{RMSProp}}\|^2_{\mathcal{H}_D} = \sum_j c_j(\beta; n) \left(\frac{\eta}{c_j(\beta; n)}\right)^2 = \eta^2 \sum_j \frac{1}{c_j(\beta; n)}. \tag{90}$$

The diagonal weighted trust region constraint $\|Q_{\text{RMSProp}}\|_{\mathcal{H}_D} \leq \sqrt{B}$ gives

$$\eta \leq \sqrt{\frac{B}{\sum_j 1/c_j(\beta; n)}}. \tag{91}$$

For fixed $\beta$, the optimal $\eta$ saturates the trust region.

*Step 2 — Alignment with the (second-moment) gradient statistics.* The expected instantaneous power under $Q_\eta$ is

$$P(Q_\eta; n) \approx \sum_j q_j\, v_j(\beta; n) = \eta \sum_j \frac{v_j(\beta; n)}{c_j(\beta; n)} = \eta \sum_j c_j(\beta; n), \tag{92}$$

since $c_j(\beta; n) = v_j(\beta; n)^{1/2}$. Plugging in $\eta^\star[n; \beta]$ gives

$$P^\star(\beta; n) \approx \sqrt{\frac{B}{\sum_j 1/c_j(\beta; n)}} \sum_j c_j(\beta; n). \tag{93}$$

*Step 3 — Reduce to 1-D search in $\beta$.* Up to the constant factor $\sqrt{B}$, the budget-normalized gain is

$$J(\beta; n) := \frac{P^\star(\beta; n)}{\sqrt{B}} = \frac{\sum_j c_j(\beta; n)}{\sqrt{\sum_j 1/c_j(\beta; n)}}. \tag{94}$$

Therefore an instantaneously optimal $\beta^\star[n]$ satisfies

$$\beta^\star[n] \in \arg\max_{\beta \in (0,1)} J(\beta; n), \tag{95}$$

and the corresponding learning rate is

$$\eta^\star[n] = \sqrt{\frac{B}{\sum_j 1/c_j(\beta^\star[n]; n)}}. \tag{96}$$

This completes the proof. □

### F.8 MUON

The Muon optimizer (Liu et al., 2025) is an approximated preconditioned SGD with momentum, which is recently proposed for training large language models, designed to replace the former widely used alternative: AdamW (Loshchilov & Hutter, 2019). It combines momentum with an orthogonalization step that approximates the orthogonal polar factor of the momentum matrix, effectively performing steepest descent under a spectral-norm trust region.

For a matrix-shaped parameter block $\Theta_t \in \mathbb{R}^{p \times q}$, Muon performs the following operations:

$$G_t = \nabla_\Theta \mathcal{L}_t(\Theta_{t-1}), \tag{97}$$

$$B_t = \mu B_{t-1} + G_t = \sum_{k=0}^{\infty} \mu^k G_{t-k}, \tag{98}$$

$$O_t = \text{NewtonSchulz5}(B_t) \approx \text{Ortho}(B_t), \tag{99}$$

$$\Theta_t = \Theta_{t-1} - \eta O_t, \tag{100}$$

where $\text{Ortho}(B_t)$ denotes the orthogonal polar factor of $B_t$, and $\text{NewtonSchulz5}(B_t)$ is a fast analytic approximation.

**Corollary 20** (Instantaneously optimal Muon). *Consider a matrix-shaped parameter block $\Theta_n \in \mathbb{R}^{p \times q}$ and the spectral-norm trust region*

$$\mathcal{U}_\gamma = \{\Delta\Theta \in \mathbb{R}^{p \times q} : \|\Delta\Theta\|_{op} \leq \gamma\}. \tag{101}$$

*Given gradients $G_n$, define the matrix momentum $B_\mu[n] := \sum_{k=0}^{\infty} \mu^k G_{n-k}$ and its orthogonalized version $O_\mu[n] := \text{Ortho}(B_\mu[n])$.*

*At time $n$, Muon with momentum parameter $\mu \in (0,1)$ and learning rate $\eta = \gamma$ performs the update*

$$\Delta\Theta_n(\mu) = -\eta O_\mu[n], \quad O_\mu[n] = \text{Ortho}\left(\sum_{k=0}^{\infty} \mu^k G_{n-k}\right). \tag{102}$$

*The expected instantaneous loss drop is*

$$-\mathbb{E}[\Delta\mathcal{L}_n] \approx \eta J_{\text{Muon}}(\mu; n), \tag{103}$$

*where*

$$J_{\text{Muon}}(\mu; n) := \mathbb{E}\left[\left\langle G_n, \text{Ortho}\left(\sum_{k=0}^{\infty} \mu^k G_{n-k}\right)\right\rangle\right]. \tag{104}$$

*Since $\|\text{Ortho}(B_\mu[n])\|_{op} = 1$ for all $\mu$, the trust region is independent of $\mu$, and the instantaneously optimal Muon momentum at step $n$ is*

$$\mu^\star[n] \in \arg\max_{\mu \in (0,1)} \mathbb{E}\left[\langle G_n, \text{Ortho}\left(\sum_{k=0}^{\infty} \mu^k G_{n-k}\right)\rangle\right]. \tag{105}$$

*In practice, we replace $\text{Ortho}(\cdot)$ by $\text{NewtonSchulz5}(\cdot)$.*

*Proof of Corollary 20.* We work with matrix-valued parameters and use the Frobenius inner product $\langle A, B \rangle = \text{Tr}(A^\top B)$.

*Step 1 — Properties of the orthogonal polar factor.* For any nonzero matrix $X \in \mathbb{R}^{p \times q}$ with SVD $X = U\Sigma V^\top$, the orthogonal polar factor is $\text{Ortho}(X) = UV^\top$. Key properties:

- Semi-orthogonality: $\|\text{Ortho}(X)\|_{op} = 1$ and $\|\text{Ortho}(X)\|_F^2 = \text{rank}(X)$.

- Scale invariance: $\text{Ortho}(\alpha X) = \text{Ortho}(X)$ for any $\alpha > 0$.

- Spectral-nuclear duality: For the spectral-norm unit ball $\mathcal{U}_1 = \{Y : \|Y\|_{op} \leq 1\}$,

$$\sup_{Y \in \mathcal{U}_1} \langle X, Y \rangle = \|X\|_*, \quad \arg\sup_{Y \in \mathcal{U}_1} \langle X, Y \rangle = \{\text{Ortho}(X)\}, \tag{106}$$

where $\|\cdot\|_*$ is the nuclear norm.

*Step 2 — Stateless optimal update for fixed momentum.* Fix time $n$ and momentum parameter $\mu$. Consider the momentum matrix $B_\mu[n] = \sum_{k=0}^{\infty} \mu^k G_{n-k}$. To maximize alignment with $B_\mu[n]$ under the spectral-norm trust region:

$$\Delta\Theta_n^\star(\mu) \in \arg\max_{\Delta\Theta_n \in \mathcal{U}_\gamma} \langle B_\mu[n], \Delta\Theta_n \rangle. \tag{107}$$

By spectral-nuclear duality, the unique maximizer is

$$\Delta\Theta_n^\star(\mu) = \gamma \operatorname{Ortho}(B_\mu[n]). \tag{108}$$

For gradient descent, we move against the momentum: $\Delta\Theta_n(\mu) = -\eta\, O_\mu[n]$ with $O_\mu[n] = \operatorname{Ortho}(B_\mu[n])$ and $\eta = \gamma$.

*Step 3 — Trust region saturation and independence from $\mu$.* Since $\|\operatorname{Ortho}(B_\mu[n])\|_{\mathrm{op}} = 1$ for all $\mu$ and all realizations of $B_\mu[n]$, we have

$$\|\Delta\Theta_n(\mu)\|_{\mathrm{op}} = \eta\|\operatorname{Ortho}(B_\mu[n])\|_{\mathrm{op}} = \eta. \tag{109}$$

With $\eta = \gamma$, the spectral-norm trust region is always saturated, independent of $\mu$. Unlike SGD+Momentum and Adam, where the trust region norm depends on momentum parameters, here orthogonalization pins the operator norm to 1, so $\eta$ controls step size and $\mu$ only affects direction.

*Step 4 — Instantaneous optimality.* With $\eta$ fixed to saturate the trust region, maximizing instantaneous loss drop is equivalent to maximizing the normalized alignment:

$$J_{\mathrm{Muon}}(\mu; n) = \mathbb{E}[\langle G_n, O_\mu[n]\rangle] = \mathbb{E}\left[\left\langle G_n, \operatorname{Ortho}\left(\sum_{k=0}^\infty \mu^k G_{n-k}\right)\right\rangle\right]. \tag{110}$$

The expected instantaneous loss drop is $-\mathbb{E}[\Delta\mathcal{L}_n] \approx \eta J_{\mathrm{Muon}}(\mu; n)$, so the instantaneously optimal momentum is

$$\mu^\star[n] \in \arg\max_{0 < \mu < 1} J_{\mathrm{Muon}}(\mu; n). \tag{111}$$

*Step 5 — First-order optimality condition.* If $J_{\mathrm{Muon}}(\mu; n)$ is differentiable and the maximizer lies in the interior, any optimal $\mu^\star[n]$ satisfies:

$$\left.\frac{d}{d\mu} J_{\mathrm{Muon}}(\mu; n)\right|_{\mu=\mu^\star[n]} = 0. \tag{112}$$

Using the chain rule:

$$\frac{d}{d\mu} J_{\mathrm{Muon}}(\mu; n) = \mathbb{E}\left[G_n : D\operatorname{Ortho}(B_\mu[n])\left[\frac{\partial B_\mu[n]}{\partial\mu}\right]\right], \tag{113}$$

where $D\operatorname{Ortho}(B)[\cdot]$ is the Fréchet derivative of the orthogonalization map and $\frac{\partial B_\mu[n]}{\partial\mu} = \sum_{k=1}^\infty k\mu^{k-1} G_{n-k}$.

*Step 6 — Practical implementation with NewtonSchulz5.* In practice, Muon uses NewtonSchulz5 as a fast analytic approximation of the orthogonal polar factor. The practical optimality condition becomes:

$$\mu^\star[n] \approx \arg\max_{0 < \mu < 1} \mathbb{E}\left[\left\langle G_n, \operatorname{NewtonSchulz5}\left(\sum_{k=0}^\infty \mu^k G_{n-k}\right)\right\rangle\right]. \tag{114}$$

This completes the proof. $\qquad\square$

# G   MORE MATHEMATICAL RESULTS

This section provides more detailed mathematical foundations of the main text, which was omitted for brevity. Appendix G.1 gives a detailed derivation of equation 9, showing that the learning power of a dynamic optimizer is also represented by some inner product between the optimizer operator and the gradient moment. This has the same structure as in the stateless case. Appendix G.3 lifts the results of four general families of optimizers to the dynamic setting. This completes the connections between Section 2 and Section 3.

## G.1   DERIVATION OF DYNAMIC PROBLEM P3 (SUPPLEMENT TO SECTION 3)

Here we provide a detailed derivation of equation 9 that was abbreviated in the main text. We assume the gradient sequence $\{g[n]\}$ is zero-mean, wide-sense stationary (WSS) with finite second

moments. Hence the lag-$k$ moment $R[k] = \mathbb{E}[g[n]g[n-k]^\top]$ depends only on $k$. Then $P(Q) = \langle Q, R \rangle_{\mathcal{H}}$. We start from the convolution definition of the dynamic optimizer:

$$\dot{\theta}[n] = \sum_{k=0}^{\infty} Q[k]\, g[n-k]. \tag{115}$$

The we can calculate the instantaneous power as follows:

$$P(Q; n) = \mathbb{E}\big[g[n]^\top \dot{\theta}[n]\big] \tag{116}$$

$$= \mathbb{E}\Big[g[n]^\top \sum_{k=0}^{\infty} Q[k]\, g[n-k]\Big] \tag{117}$$

$$= \sum_{k=0}^{\infty} \mathbb{E}\big[g[n]^\top Q[k]\, g[n-k]\big] \quad \text{(linearity of } \mathbb{E}) \tag{118}$$

$$= \sum_{k=0}^{\infty} \mathrm{Tr}\big(Q[k]\, \mathbb{E}[g[n-k]\, g[n]^\top]\big) \tag{119}$$

$$= \sum_{k=0}^{\infty} \mathrm{Tr}\big(Q[k]^T\, \mathbb{E}[g[n]\, g[n-k]^\top]\big) \quad \text{(trace transpose)} \tag{120}$$

$$= \sum_{k=0}^{\infty} \mathrm{Tr}\big(Q[k]^T\, R[k]\big) \tag{121}$$

$$= \langle Q, R \rangle_{\mathcal{H}}. \tag{122}$$

The equations come from the following facts: (i) linearity of expectation, (ii) Fubini/Tonelli theorem to swap summation and expectation (valid for finite-energy filters and WSS gradients), and (iii) the scalar-trace identity $a^\top B c = \mathrm{Tr}(Bca^\top)$. This completes the derivation of equation 9.

## G.2 DERIVATION OF INSTANTANEOUS VALIDATION POWER (SUPPLEMENT TO APPENDIX E)

Similar to above derivation, we can also derive the instantaneous validation power introduced in equation 27 of Appendix E as follows:

$$P_{\mathrm{val}}(Q; n) = \mathbb{E}\big[g_{\mathrm{val}}[n]^\top \dot{\theta}[n]\big] \tag{123}$$

$$= \mathbb{E}\Big[g_{\mathrm{val}}[n]^\top \sum_{k=0}^{\infty} Q[k]\, g_{\mathrm{tr}}[n-k]\Big] \tag{124}$$

$$= \sum_{k=0}^{\infty} \mathbb{E}\big[g_{\mathrm{val}}[n]^\top Q[k]\, g_{\mathrm{tr}}[n-k]\big] \tag{125}$$

$$= \sum_{k=0}^{\infty} \mathrm{Tr}\big(Q[k]\, \mathbb{E}[g_{\mathrm{tr}}[n-k]\, g_{\mathrm{val}}[n]^\top]\big) \tag{126}$$

$$= \sum_{k=0}^{\infty} \mathrm{Tr}\big(Q[k]^T\, \mathbb{E}[g_{\mathrm{val}}[n]\, g_{\mathrm{tr}}[n-k]^\top]^T\big) \tag{127}$$

$$= \sum_{k=0}^{\infty} \mathrm{Tr}\big(Q[k]^T\, C[k]^T\big), \tag{128}$$

where $C[k] := \mathbb{E}[g_{\mathrm{val}}[n]\, g_{\mathrm{tr}}[n-k]^\top]$. For the symmetric cross-moment $R_{\mathrm{val}}[k] := \frac{1}{2}\big(C[k] + C[k]^\top\big)$, when using Hermitian PSD filters with symmetric $Q[k]$, we have

$$\mathrm{Tr}(Q[k]^T C[k]^T) = \mathrm{Tr}(Q[k]\, \tfrac{1}{2}(C[k] + C[k]^\top)) = \mathrm{Tr}(Q[k]^T R_{\mathrm{val}}[k]), \tag{129}$$

since for symmetric $A$, $\mathrm{Tr}(A\,\frac{1}{2}(B - B^\top)) = 0$. Therefore, $P_{\mathrm{val}}(Q; n) = \langle Q, R_{\mathrm{val}} \rangle_{\mathcal{H}}$. Under WSS, both powers become independent of $n$, yielding $P_{\mathrm{tr}}(Q) = \langle Q, R_{\mathrm{tr}} \rangle_{\mathcal{H}}$ and $P_{\mathrm{val}}(Q) = \langle Q, R_{\mathrm{val}} \rangle_{\mathcal{H}}$. Likewise, the key assumptions used are: linearity of expectation, Fubini/Tonelli theorem to swap summation and expectation (valid for finite-energy filters and WSS gradients), and the scalar-trace identity $a^\top B c = \mathrm{Tr}(Bca^\top)$.

### G.3 FOUR FAMILIES OF OPTIMAL DYNAMIC OPTIMIZERS (SUPPLEMENT TO SECTION 3)

Here we lift the key results of Section 2 to the dynamic setting of Section 3 for completeness. Solving the optimization problem with constraint $\mathcal{Q}$ *determines* the optimal dynamic optimizer $Q^\star$, and endows the optimizer with different characteristics and algorithmic behaviors. Recall that the moment operator $R[n]$ is a Hermitian PSD matrix for each time $n$ and delay $k$ which is defined as:

$$R[n; k] = \mathbb{E}[g[n]\, g[n-k]^\top], \tag{130}$$

for $k = 0, 1, 2, \ldots$ and $n = 0, 1, 2, \ldots$ Consider the following four types of dynamic trust regions:

- *Frobenius ball type* $\mathcal{Q}_{\mathrm{F}}(B) = \{Q : \|Q\|_{\mathcal{H}}^2 = \sum_{k=0}^{\infty} \mathrm{Tr}(Q[k]^\top Q[k]) \leq B\}$ is the simplest and the largest family that does not favor any particular direction in the parameter space, but requires larger memory to store its hyperparameters.

- *Spectral type* $\mathcal{Q}_{\mathrm{S}}(\tau, \lambda) = \{Q : \mathrm{Tr}(Q[k]) \leq \tau[k],\, 0 \preceq Q[k] \preceq \lambda[k]I\, \forall k\}$ is a trust region that upper limits the (1) per-direction spectrum for safety and the (2) trace for total update budget, simultaneously at each time delay $k$.

- *Lyapunov type* $\mathcal{Q}_{\mathrm{L}}(B) = \{Q : \mathrm{Tr}(Q[k]^\top R[k]Q[k]) \leq B[k]\, \forall k\}$ utilizes the lag-covariance sequence itself as the metric, leading to a natural dynamic Lyapunov-like stability condition.

- *Diagonal type* $\mathcal{Q}_{\mathrm{D}}(B, c) = \{Q : Q[k] = \mathrm{diag}(q_j[k]) \succeq 0,\, \sum_j c_j[k]q_j[k]^2 \leq B\, \forall k\}$ represents element-wise optimizers, a memory-efficient family that are commonly used in large-scale machine learning.

Instantiating the construction from Theorem 3 on each of these families, we obtain the closed-form optimal dynamic optimizer $Q^\star$ and the corresponding optimal learning power $P^*(R)$.

**Corollary 21** (Closed-form solutions for dynamic trust regions)**.** *Omit the time index $n$ for brevity. Let the moment $R[k]$ has the eigendecomposition $R[k] = U[k]\,\mathrm{diag}(\sigma_1[k] \geq \cdots \geq \sigma_d[k])U[k]^*$ for each time delay $k$. The closed-form optimal solutions are:*

*(i)* (Frobenius ball): $Q_{\mathrm{F}}^\star = \sqrt{B}R/\|R\|_{\mathcal{H}}$ *(if $R \neq 0$; otherwise any feasible $Q$ is optimal). This gives $P_{\mathrm{F}}^\star(R) = \sqrt{B}\|R\|_{\mathcal{H}}$.*

*(ii)* (Spectral): $Q_{\mathrm{S}}^\star[k]$ *shares the same eigenstructure as $R[k]$ but with eigenvalues determined by water-filling, i.e.,*

$$Q_{\mathrm{S}}^\star[k] = U[k]\,\mathrm{diag}(q_i^\star[k])U[k]^*, \tag{131}$$

*where the eigenvalues are sorted in nonincreasing order of $\sigma_i[k]$ and given by:*

$$q_i^\star[k] = \begin{cases} \lambda[k] & \text{if } i \leq \kappa[k], \\ \tau[k] - \kappa[k]\lambda[k] & \text{if } i = \kappa[k] + 1 \text{ and } \tau[k] < d\lambda[k], \\ 0 & \text{otherwise}, \end{cases} \tag{132}$$

*with $\kappa[k] = \min(\lfloor \tau[k]/\lambda[k] \rfloor, d)$. This gives*

$$P_{\mathrm{S}}^\star(R) = \sum_{k=0}^{\infty} \left[ \lambda[k] \sum_{i \leq \kappa[k]} \sigma_i[k] + (\tau[k] - \kappa[k]\lambda[k])\sigma_{\kappa[k]+1}[k] \right]. \tag{133}$$

*(iii)* (Lyapunov): *For each delay $k$, $Q_{\mathrm{L}}^\star[k] = \alpha[k]\Pi_{R[k]}$, where $\Pi_{R[k]}$ is the orthogonal projection onto the support of $R[k]$, and $\alpha[k] = \sqrt{B[k]/\sum_{i:\sigma_i[k]>0} \sigma_i[k]}$. This gives $P_{\mathrm{L}}^\star(R) = \sum_{k=0}^{\infty} \sqrt{B[k]\sum_i \sigma_i[k]}$.*

*(iv)* (Diagonal): *For each delay $k$, $[Q_{\mathrm{D}}^\star[k]]_{jj} = \alpha[k]\sigma_j[k]/c_j[k]$ where $\alpha[k] = \sqrt{B/\sum_\ell \sigma_\ell[k]^2/c_\ell[k]}$, and $U[k] \equiv I$. This gives $P_{\mathrm{D}}^\star(R) = \sum_{k=0}^{\infty} \sqrt{B\sum_j \sigma_j[k]^2/c_j[k]}$.*

*Proof.* We apply Theorem 3 to each dynamic feasible set.

*(i) Frobenius ball* $\mathcal{Q}_{\mathrm{F}}(B) = \{Q : \|Q\|_{\mathcal{H}} \leq \sqrt{B}\}$. The Lagrangian is $L(Q, \lambda) = \langle Q, R \rangle_{\mathcal{H}} - \lambda(\|Q\|_{\mathcal{H}}^2 - B)$. Taking the gradient with respect to $Q$ and setting to zero gives

$$\nabla_Q L = R - 2\lambda Q = 0 \quad \Rightarrow \quad Q = \frac{R}{2\lambda}. \tag{134}$$

The constraint $\|Q\|_{\mathcal{H}} = \sqrt{B}$ gives $\|R/(2\lambda)\|_{\mathcal{H}} = \sqrt{B}$, so $2\lambda = \|R\|_{\mathcal{H}}/\sqrt{B}$. Hence

$$Q_{\mathrm{F}}^\star = \sqrt{B}\,\frac{R}{\|R\|_{\mathcal{H}}}, \quad P_{\mathrm{F}}^\star(R) = \langle Q_{\mathrm{F}}^\star, R\rangle_{\mathcal{H}} = \sqrt{B}\,\|R\|_{\mathcal{H}}. \tag{135}$$

*(ii) Spectral* $\mathcal{Q}_{\mathrm{S}}(\tau, \lambda) = \{Q : \mathrm{Tr}(Q[k]) \leq \tau[k], 0 \preceq Q[k] \preceq \lambda[k]I\,\forall k\}$. The problem decouples over delays $k$. For each $k$, by von Neumann's trace inequality, the maximizer has the form $Q[k] = U[k]\,\mathrm{diag}(q_i[k])U[k]^*$ where the eigenvalues $q_i[k]$ solve the linear program:

$$\max_{0 \leq q_i[k] \leq \lambda[k]} \sum_i q_i[k]\sigma_i[k] \quad \text{s.t.} \quad \sum_i q_i[k] \leq \tau[k]. \tag{136}$$

The optimal solution allocates maximum weight $\lambda[k]$ to the largest eigenvalues $\sigma_i[k]$ until the trace budget $\tau[k]$ is exhausted, giving the stated water-filling formula.

*(iii) Lyapunov* $\mathcal{Q}_{\mathrm{L}}(B) = \{Q : \mathrm{Tr}(Q[k]^\top R[k]Q[k]) \leq B[k]\,\forall k\}$. The problem decouples over delays $k$. For each $k$, the Lagrangian is

$$L_k(Q[k], \mu[k]) = \mathrm{Tr}(Q[k]^\top R[k]) - \mu[k](\mathrm{Tr}(Q[k]^\top R[k]Q[k]) - B[k]). \tag{137}$$

The first-order condition gives:

$$R[k] - 2\mu[k]R[k]Q[k] = 0 \quad \Rightarrow \quad R[k](I - 2\mu[k]Q[k]) = 0. \tag{138}$$

This implies $Q[k] = \frac{1}{2\mu[k]}I$ on the support of $R[k]$, i.e., $Q[k] = \alpha[k]\,\Pi_{R[k]}$ where $\alpha[k] = \frac{1}{2\mu[k]}$. Using the constraint:

$$\mathrm{Tr}(Q[k]^\top R[k]Q[k]) = \alpha[k]^2\,\mathrm{Tr}(R[k]) = \alpha[k]^2 \sum_{i:\sigma_i[k]>0} \sigma_i[k] = B[k] \tag{139}$$

Therefore, $\alpha[k] = \sqrt{B[k]/\sum_{i:\sigma_i[k]>0}\sigma_i[k]}$, giving the stated result.

*(iv) Diagonal* $\mathcal{Q}_{\mathrm{D}}(B, c) = \{Q : Q[k] = \mathrm{diag}(q_j[k]) \succeq 0, \sum_j c_j[k]q_j[k]^2 \leq B\,\forall k\}$. The problem decouples over delays $k$. For each $k$, assuming $R[k]$ is diagonal with $R[k] = \mathrm{diag}(\sigma_j[k])$, we solve:

$$\max_{q_j[k] \geq 0} \sum_j \sigma_j[k]q_j[k] \quad \text{s.t.} \quad \sum_j c_j[k]q_j[k]^2 \leq B. \tag{140}$$

By Cauchy-Schwarz, the maximizer satisfies $q_j^\star[k] \propto \sigma_j[k]/c_j[k]$. Normalizing by the constraint gives the stated result. $\qquad\square$

The proof is similar to the proof of Corollary 2 of the main manuscript, which is provided in Appendix H. These analytic solutions reveal how the characteristics of different types of optimal dynamic optimizers $Q^\star$ are induced by controlling the feasible set $\mathcal{Q}$. Interpretation of the solutions is the same as the interpretation of the corresponding stateless solutions in Section 2.

## H    PROOFS OMITTED FROM THE MAIN TEXT

This section does all the proofs that has been omitted in the main text. The proofs are organized in the same order as the theorems appear in the main manuscript.

### H.1    PROOF OF THEOREM 1

*Proof of Theorem 1.* We establish each claim in turn.

*(i) Existence & sublinearity:* Since $\mathcal{Q}$ is compact by assumption (a nonempty, compact, convex subset of $\mathbb{S}_+^d$), and the inner product $(Q, \Sigma) \mapsto \langle Q, \Sigma\rangle = \mathrm{Tr}(Q^\top\Sigma) = \mathrm{Tr}(Q\Sigma)$ is continuous, the maximum is attained by the Weierstrass extreme value theorem. The optimal power $P^\star(\Sigma) = \sup_{Q\in\mathcal{Q}}\mathrm{Tr}(Q\Sigma)$ is a supremum of linear functionals in $\Sigma$, hence sublinear (convex and positively homogeneous). Finiteness follows from compactness of $\mathcal{Q}$.

*(ii) Conjugacy identities:* We establish the three identities in equation 5.

*(1) Optimal power = conjugate of indicator.* By the definition of convex conjugate,

$$(\delta_{\mathcal{Q}})^*(\Sigma) = \sup_{Q \in \mathbb{S}^d} \{\langle Q, \Sigma \rangle - \delta_{\mathcal{Q}}(Q)\} = \sup_{Q \in \mathcal{Q}} \langle Q, \Sigma \rangle = P^\star(\Sigma). \tag{141}$$

Thus $P^\star = (\delta_{\mathcal{Q}})^*$.

*(2) Optimal power = gauge of polar.* By the definition of polar, $\Sigma \in \mathcal{Q}^\circ$ if and only if $\sup_{Q \in \mathcal{Q}}\langle Q, \Sigma \rangle \le 1$, i.e., $P^\star(\Sigma) \le 1$. Therefore

$$\gamma_{\mathcal{Q}^\circ}(\Sigma) = \inf\{\lambda > 0 : \Sigma \in \lambda \mathcal{Q}^\circ\} = \inf\{\lambda > 0 : P^\star(\Sigma) \le \lambda\} = P^\star(\Sigma). \tag{142}$$

Thus $P^\star = \gamma_{\mathcal{Q}^\circ}$.

*(3) Conjugate of gauge = indicator of polar.* We establish $(\gamma_{\mathcal{Q}})^* = \delta_{\mathcal{Q}^\circ}$. Consider two cases:

- If $\Sigma \in \mathcal{Q}^\circ$, then for all $Q$,

$$\langle Q, \Sigma \rangle \le \gamma_{\mathcal{Q}}(Q) \cdot \sup_{Q' \in \mathcal{Q}} \langle Q', \Sigma \rangle \le \gamma_{\mathcal{Q}}(Q), \tag{143}$$

  since $\sup_{Q' \in \mathcal{Q}}\langle Q', \Sigma \rangle \le 1$ by definition of polar. Hence $\langle Q, \Sigma \rangle - \gamma_{\mathcal{Q}}(Q) \le 0$ for all $Q$, with equality at $Q = 0$. Taking the supremum gives $(\gamma_{\mathcal{Q}})^*(\Sigma) = 0 = \delta_{\mathcal{Q}^\circ}(\Sigma)$.

- If $\Sigma \notin \mathcal{Q}^\circ$, there exists $Q_0 \in \mathcal{Q}$ with $\langle Q_0, \Sigma \rangle > 1$. For any $\alpha > 0$, we have $\gamma_{\mathcal{Q}}(\alpha Q_0) = \alpha \gamma_{\mathcal{Q}}(Q_0) = \alpha$ (since $Q_0 \in \mathcal{Q}$ and $\mathcal{Q}$ is bounded so $\gamma_{\mathcal{Q}}(Q_0) = 1$), and thus

$$\langle \alpha Q_0, \Sigma \rangle - \gamma_{\mathcal{Q}}(\alpha Q_0) = \alpha \langle Q_0, \Sigma \rangle - \alpha = \alpha(\langle Q_0, \Sigma \rangle - 1) \to +\infty \quad (\text{as } \alpha \to \infty). \tag{144}$$

  Hence $(\gamma_{\mathcal{Q}})^*(\Sigma) = +\infty = \delta_{\mathcal{Q}^\circ}(\Sigma)$.

Thus $(\gamma_{\mathcal{Q}})^* = \delta_{\mathcal{Q}^\circ}$.

By (1), (2), and (3), we have $P^\star = \gamma_{\mathcal{Q}^\circ} = (\delta_{\mathcal{Q}^\circ})^*$ and $(\gamma_{\mathcal{Q}})^* = \delta_{\mathcal{Q}^\circ}$.

*(iii) Construction:* Let $Q^\star \in \arg\max_{Q \in \mathcal{Q}} \operatorname{Tr}(Q\Sigma)$. For any $M \in \mathbb{S}^d$,

$$P^\star(M) = \max_{Q \in \mathcal{Q}} \operatorname{Tr}(QM) \ge \operatorname{Tr}(Q^\star M) = \operatorname{Tr}(Q^\star \Sigma) + \operatorname{Tr}(Q^\star(M - \Sigma)) = P^\star(\Sigma) + \operatorname{Tr}(Q^\star(M - \Sigma)), \tag{145}$$

which is the defining inequality for $Q^\star \in \partial P^\star(\Sigma)$. If the maximizer is unique, then $\partial P^\star(\Sigma) = \{Q^\star\}$, and thus $P^\star$ is differentiable at $\Sigma$ with $\nabla P^\star(\Sigma) = Q^\star$.

*(iv) Order preservation on $\mathbb{S}_+^d$):* If $\Sigma \succeq 0$, then for any $Q \in \mathcal{Q} \subseteq \mathbb{S}_+^d$, we have $\operatorname{Tr}(Q\Sigma) \ge 0$. Since $0 \in \mathcal{Q}$, the maximum over $Q \in \mathcal{Q}$ is $\ge 0$. If $\Sigma_1 \succeq \Sigma_2$, then $P^\star(\Sigma_1) \ge P^\star(\Sigma_2)$. Moreover, strict inequality holds if there exists $Q \in \mathcal{Q}$ with $\operatorname{Tr}\big(Q(\Sigma_1 - \Sigma_2)\big) > 0$.

*(v) Lipschitz continuity in symmetrized polar gauge:* We establish the one-sided bounds first. Since $P^\star = \gamma_{\mathcal{Q}^\circ}$ by the conjugacy identities, we have:

$$P^\star(\Sigma) - P^\star(\hat{\Sigma}) = \max_{Q \in \mathcal{Q}} \langle Q, \Sigma \rangle - \max_{Q \in \mathcal{Q}} \langle Q, \hat{\Sigma} \rangle \tag{146}$$

$$\le \max_{Q \in \mathcal{Q}} \langle Q, \Sigma - \hat{\Sigma} \rangle \tag{147}$$

$$= \gamma_{\mathcal{Q}^\circ}(\Sigma - \hat{\Sigma}). \tag{148}$$

Similarly, $P^\star(\hat{\Sigma}) - P^\star(\Sigma) \le \gamma_{\mathcal{Q}^\circ}(\hat{\Sigma} - \Sigma)$. Therefore,

$$|P^\star(\Sigma) - P^\star(\hat{\Sigma})| \le \max\{\gamma_{\mathcal{Q}^\circ}(\Sigma - \hat{\Sigma}), \gamma_{\mathcal{Q}^\circ}(\hat{\Sigma} - \Sigma)\} = \|\Sigma - \hat{\Sigma}\|_{\mathcal{Q}^\circ}^{\mathrm{sym}}. \tag{149}$$

This Lipschitz property is essential for robustness analysis. By using an estimated moment $\hat{\Sigma}$ instead of the true moment $\Sigma$, the error in optimal power can be bounded by $|P^\star(\Sigma) - P^\star(\hat{\Sigma})| \le \|\Sigma - \hat{\Sigma}\|_{\mathcal{Q}^\circ}^{\mathrm{sym}}$. This provides a principled way to assess estimation sensitivity. $\qquad\square$

## H.2 Proof of Corollary 2

*Proof of Corollary 2.* We apply Theorem 1 to each type of trust regions. Let $\Sigma = U \operatorname{diag}(\sigma_1 \geq \cdots \geq \sigma_d)U^\top$ be the eigendecomposition of the moment matrix.

*(i) Frobenius ball* $\mathcal{Q}_F(B) = \{Q \succeq 0 : \|Q\|_F \leq \sqrt{B}\}$. The Lagrangian is $L(Q, \lambda) = \operatorname{Tr}(Q\Sigma) - \lambda(\|Q\|_F^2 - B)$. Taking the gradient with respect to $Q$ and setting to zero gives

$$\nabla_Q L = \Sigma - 2\lambda Q = 0 \quad \Rightarrow \quad Q = \frac{\Sigma}{2\lambda}. \tag{150}$$

The constraint $\|Q\|_F = \sqrt{B}$ gives $\|\Sigma/(2\lambda)\|_F = \sqrt{B}$, so $2\lambda = \|\Sigma\|_F/\sqrt{B}$. Hence

$$Q_F^\star = \sqrt{B}\,\frac{\Sigma}{\|\Sigma\|_F}, \quad P_F^\star(\Sigma) = \operatorname{Tr}(Q_F^\star\Sigma) = \sqrt{B}\,\|\Sigma\|_F. \tag{151}$$

*(ii) Spectral* $\mathcal{Q}_S(\tau, \lambda) = \{Q \succeq 0 : \operatorname{Tr}(Q) \leq \tau,\ Q \preceq \lambda I\}$. By Neumann's inequality, the maximizer has the form $Q = U \operatorname{diag}(q_i)U^\top$ where the eigenvalues $q_i$ solve the water-filling problem:

$$\max_{q_i \geq 0} \sum_i q_i \sigma_i \quad \text{s.t.} \quad \sum_i q_i \leq \tau,\ q_i \leq \lambda. \tag{152}$$

The KKT conditions yield

$$q_i^\star = \begin{cases} \lambda & (i \leq k), \\ \tau - k\lambda & (i = k+1), \\ 0 & (i > k+1). \end{cases} \tag{153}$$

where $k = \lfloor \tau/\lambda \rfloor$. The optimal power is

$$P_S^\star(\Sigma) = \lambda \sum_{i \leq k} \sigma_i + (\tau - k\lambda)\sigma_{k+1}. \tag{154}$$

*(iii) Lyapunov* $\mathcal{Q}_L(B) = \{Q \succeq 0 : \operatorname{Tr}(Q^2\Sigma) \leq B\}$. The Lagrangian is $\mathcal{L}(Q, \mu) = \operatorname{Tr}(Q\Sigma) - \mu(\operatorname{Tr}(Q^2\Sigma) - B)$. The first-order condition gives

$$\nabla_Q L = \Sigma - 2\mu Q\Sigma = 0 \quad \Rightarrow \quad Q\Sigma = \frac{1}{2\mu}\Sigma. \tag{155}$$

This implies $Q = \frac{1}{2\mu}I$ on the support of $\Sigma$, i.e., $Q = \alpha\,\Pi_\Sigma$ where $\Pi_\Sigma$ is the orthogonal projection onto $\operatorname{supp}(\Sigma)$ and $\alpha = \frac{1}{2\mu}$. Using the constraint $\operatorname{Tr}(Q^2\Sigma) = B$:

$$\operatorname{Tr}(\alpha^2\Pi_\Sigma^2\Sigma) = \alpha^2 \operatorname{Tr}(\Pi_\Sigma\Sigma) = \alpha^2 \sum_{i:\sigma_i>0} \sigma_i = B. \tag{156}$$

Therefore $\alpha = \sqrt{B}\,(\sum_{i:\sigma_i>0} \sigma_i)^{-1/2}$, giving:

$$Q_L^\star = \alpha\,\Pi_\Sigma, \quad P_L^\star(\Sigma) = \operatorname{Tr}(Q_L^\star\Sigma) = \alpha \sum_i \sigma_i = \sqrt{B}\,(\sum_i \sigma_i)^{1/2}. \tag{157}$$

*(iv) Diagonal* $\mathcal{Q}_D(B, c) = \{Q = \operatorname{diag}(q_j) \succeq 0 : \sum_j c_j q_j^2 \leq B\}$. The problem decouples coordinate-wise:

$$\max_{q_j \geq 0} \sum_j q_j \Sigma_{jj} \quad \text{s.t.} \quad \sum_j c_j q_j^2 \leq B. \tag{158}$$

By Cauchy-Schwarz, the maximizer satisfies $q_j^\star \propto \Sigma_{jj}/c_j$. Normalizing by the constraint:

$$q_j^\star = \sqrt{\frac{B}{\sum_k \Sigma_{kk}^2/c_k}} \cdot \frac{\Sigma_{jj}}{c_j}, \quad P_D^\star(\Sigma) = \sqrt{B \sum_j \frac{\Sigma_{jj}^2}{c_j}}. \tag{159}$$

Letting $\sigma_j = \Sigma_{jj}$, we have the result. $\qquad\square$

## H.3 Proof of Theorem 3

*Proof of Theorem 3.* We establish each claim in turn. The proof is largely similar to Theorem 1 due to the similarity of the problem settings, except for the type of inner product being used (Frobeinus vs. Hibert). Here, we give the full proof for completeness.

*(i) Existence & sublinearity:* Since $\mathcal{Q}$ is weakly compact in the Hilbert space $\mathcal{H}$ (closed and bounded sets in reflexive spaces are weakly compact), and the inner product $(Q, R) \mapsto \langle Q, R \rangle_{\mathcal{H}}$ is continuous in the weak topology, the maximum is attained by the Weierstrass extreme value theorem. The optimal power $P^\star(R) = \sup_{Q \in \mathcal{Q}} \langle Q, R \rangle_{\mathcal{H}}$ is a supremum of linear functionals in $R$, hence sublinear (convex and positively homogeneous). Finiteness follows from compactness of $\mathcal{Q}$.

*(ii) Conjugacy identities:* We establish the three identities in the dynamic setting.

*(1) Optimal power = conjugate of indicator.* By the definition of convex conjugate in $\mathcal{H}$,

$$(\delta_{\mathcal{Q}})^*(R) = \sup_{Q \in \mathcal{H}} \{\langle Q, R \rangle_{\mathcal{H}} - \delta_{\mathcal{Q}}(Q)\} = \sup_{Q \in \mathcal{Q}} \langle Q, R \rangle_{\mathcal{H}} = P^\star(R). \tag{160}$$

Thus $P^\star = (\delta_{\mathcal{Q}})^*$.

*(2) Optimal power = gauge of polar.* By the definition of polar in $\mathcal{H}$, $R \in \mathcal{Q}^\circ$ if and only if $\sup_{Q \in \mathcal{Q}} \langle Q, R \rangle_{\mathcal{H}} \leq 1$, i.e., $P^\star(R) \leq 1$. Therefore

$$\gamma_{\mathcal{Q}^\circ}(R) = \inf\{\lambda > 0 : R \in \lambda \mathcal{Q}^\circ\} = \inf\{\lambda > 0 : P^\star(R) \leq \lambda\} = P^\star(R). \tag{161}$$

Thus $P^\star = \gamma_{\mathcal{Q}^\circ}$.

*(3) Conjugate of gauge = indicator of polar.* We establish $(\gamma_{\mathcal{Q}})^* = \delta_{\mathcal{Q}^\circ}$. Consider two cases:

- If $R \in \mathcal{Q}^\circ$, then for all $Q$,

$$\langle Q, R \rangle_{\mathcal{H}} \leq \gamma_{\mathcal{Q}}(Q) \cdot \sup_{Q' \in \mathcal{Q}} \langle Q', R \rangle_{\mathcal{H}} \leq \gamma_{\mathcal{Q}}(Q), \tag{162}$$

  since $\sup_{Q' \in \mathcal{Q}} \langle Q', R \rangle_{\mathcal{H}} \leq 1$ by definition of polar. Hence $\langle Q, R \rangle_{\mathcal{H}} - \gamma_{\mathcal{Q}}(Q) \leq 0$ for all $Q$, with equality at $Q = 0$. Taking the supremum gives $(\gamma_{\mathcal{Q}})^*(R) = 0 = \delta_{\mathcal{Q}^\circ}(R)$.

- If $R \notin \mathcal{Q}^\circ$, there exists $Q_0 \in \mathcal{Q}$ with $\langle Q_0, R \rangle_{\mathcal{H}} > 1$. For any $\alpha > 0$, we have $\gamma_{\mathcal{Q}}(\alpha Q_0) = \alpha \gamma_{\mathcal{Q}}(Q_0) = \alpha$ (since $Q_0 \in \mathcal{Q}$ and $\mathcal{Q}$ is bounded so $\gamma_{\mathcal{Q}}(Q_0) = 1$), and thus

$$\langle \alpha Q_0, R \rangle_{\mathcal{H}} - \gamma_{\mathcal{Q}}(\alpha Q_0) = \alpha \langle Q_0, R \rangle_{\mathcal{H}} - \alpha = \alpha(\langle Q_0, R \rangle_{\mathcal{H}} - 1) \to +\infty \quad (\text{as } \alpha \to \infty). \tag{163}$$

  Hence $(\gamma_{\mathcal{Q}})^*(R) = +\infty = \delta_{\mathcal{Q}^\circ}(R)$.

Thus $(\gamma_{\mathcal{Q}})^* = \delta_{\mathcal{Q}^\circ}$.

By (1), (2), and (3), we have $P^\star = \gamma_{\mathcal{Q}^\circ} = (\delta_{\mathcal{Q}^\circ})^*$ and $(\gamma_{\mathcal{Q}})^* = \delta_{\mathcal{Q}^\circ}$.

*(iii) Construction:* Let $Q^\star \in \arg\max_{Q \in \mathcal{Q}} \langle Q, R \rangle_{\mathcal{H}}$. For any $M \in \mathcal{H}$,

$$P^\star(M) = \max_{Q \in \mathcal{Q}} \langle Q, M \rangle_{\mathcal{H}} \geq \langle Q^\star, M \rangle_{\mathcal{H}} = \langle Q^\star, R \rangle_{\mathcal{H}} + \langle Q^\star, M - R \rangle_{\mathcal{H}} = P^\star(R) + \langle Q^\star, M - R \rangle_{\mathcal{H}}, \tag{164}$$

which is the defining inequality for $Q^\star \in \partial P^\star(R)$. If the maximizer is unique, then $\partial P^\star(R) = \{Q^\star\}$, and thus $P^\star$ is differentiable at $R$ with $\nabla P^\star(R) = Q^\star$.

*(iv) Order preservation on $\mathcal{H}_+$:* If $R \in \mathcal{H}_+$, then for any $Q \in \mathcal{Q} \subseteq \mathcal{H}_+$, we have $\langle Q, R \rangle_{\mathcal{H}} \geq 0$ (each term $\mathrm{Tr}(H_k^\top R_k) \geq 0$). Since $0 \in \mathcal{Q}$, the maximum over $Q \in \mathcal{Q}$ is $\geq 0$. If $R_1 - R_2 \in \mathcal{H}_+ \setminus \{0\}$, then there exists $Q \in \mathcal{Q}$ with $\langle Q, R_1 - R_2 \rangle_{\mathcal{H}} > 0$, hence $P^\star(R_1) > P^\star(R_2)$.

*(v) Lipschitz continuity in symmetrized polar gauge:* We establish the one-sided bounds first. Since $P^\star = \gamma_{\mathcal{Q}^\circ}$ by the conjugacy identities, we have:

$$P^\star(R) - P^\star(\hat{R}) = \max_{Q \in \mathcal{Q}} \langle Q, R \rangle_{\mathcal{H}} - \max_{Q \in \mathcal{Q}} \langle Q, \hat{R} \rangle_{\mathcal{H}} \tag{165}$$

$$\leq \max_{Q \in \mathcal{Q}} \langle Q, R - \hat{R} \rangle_{\mathcal{H}} \tag{166}$$

$$= \gamma_{\mathcal{Q}^\circ}(R - \hat{R}). \tag{167}$$

Similarly, $P^\star(\hat{R}) - P^\star(R) \le \gamma_{\mathcal{Q}^\circ}(\hat{R} - R)$. Therefore,

$$|P^\star(R) - P^\star(\hat{R})| \le \max\{\gamma_{\mathcal{Q}^\circ}(R - \hat{R}), \gamma_{\mathcal{Q}^\circ}(\hat{R} - R)\} = \|R - \hat{R}\|_{\mathcal{Q}^\circ}^{\mathrm{sym}}. \tag{168}$$

Therefore, by using an estimated moment $\hat{R}$ instead of the true moment $R$, the error in optimal power can be bounded by $|P^\star(R) - P^\star(\hat{R})| \le \|R - \hat{R}\|_{\mathcal{Q}^\circ}^{\mathrm{sym}}$. $\qquad\square$

### H.4 PROOF OF LEMMA 4

*Proof of Lemma 4.* The proof follows from standard convex optimization theory (Boyd & Vandenberghe, 2004), specifically the KKT conditions for linear maximization and the characterization of metric projections.

Note that $\mathcal{H}$ is a real Hilbert space, $\mathcal{C} \subset \mathcal{H}$ is a cone (closed under positive scaling), and $\mathcal{Q} \subset \mathcal{H}$ is a closed convex set with $0 \in \mathcal{Q}$. We also denote the normal cone of $\mathcal{Q} \cap \mathcal{C}$ at $Q$ as $N_{\mathcal{Q} \cap \mathcal{C}}(Q) := \{M \in \mathcal{H} : \langle M, Q' - Q\rangle_{\mathcal{H}} \le 0 \ \forall Q' \in \mathcal{Q} \cap \mathcal{C}\}$ and projection onto $\mathcal{C}$ as $\Pi_{\mathcal{C}}(Q) := \arg\min_{M \in \mathcal{C}} \|M - Q\|_{\mathcal{H}}$. We use two facts:

- *(KKT for linear maximization)* $x^\star \in \arg\max_{y \in \mathcal{C}} \langle y, M\rangle_{\mathcal{H}} \iff M \in N_{\mathcal{C}}(x^\star)$.

- *(Metric projection)* For $y \in \mathcal{H}$, $x^\star = \Pi_{\mathcal{C}}(y) \iff y - x^\star \in N_{\mathcal{Q} \cap \mathcal{C}}(x^\star)$.

*(ii) $\Rightarrow$ (i):* Suppose there exists $M \in \mathcal{Q}_{\mathcal{C}}^\star(R) \subseteq \mathcal{Q} \cap \mathcal{C}$ such that $\{R, Q^\star - M\} \subset N_{\mathcal{Q} \cap \mathcal{C}}(M)$. From $Q^\star - M \in N_{\mathcal{Q} \cap \mathcal{C}}(M)$, the metric projection characterization gives $\Pi_{\mathcal{C}}(Q^\star) = M$. From $R \in N_{\mathcal{Q} \cap \mathcal{C}}(Q_{\mathcal{C}}^\star)$, the KKT condition for linear maximization gives $M \in \arg\max_{Q \in \mathcal{Q} \cap \mathcal{C}} \langle Q, R\rangle_{\mathcal{H}} = \mathcal{Q}_{\mathcal{C}}^\star(R)$. Therefore, $\Pi_{\mathcal{C}}(Q^\star) = M \in \mathcal{Q}_{\mathcal{C}}^\star(R)$.

*(i) $\Rightarrow$ (ii):* Suppose $M := \Pi_{\mathcal{C}}(Q^\star) \in \mathcal{Q}_{\mathcal{C}}^\star(R)$. By the metric projection characterization, $Q^\star - M \in N_{\mathcal{Q} \cap \mathcal{C}}(M)$. Since $M \in \mathcal{Q}_{\mathcal{C}}^\star(R) = \arg\max_{Q \in \mathcal{Q} \cap \mathcal{C}} \langle Q, R\rangle_{\mathcal{H}}$, the KKT condition for linear maximization gives $R \in N_{\mathcal{Q} \cap \mathcal{C}}(M)$. Thus, $\{R, Q^\star - M\} \subset N_{\mathcal{Q} \cap \mathcal{C}}(M)$. Let $Q_{\mathcal{C}}^\star := M$.

For the final statement, if $N_{\mathcal{Q} \cap \mathcal{C}}(Q_{\mathcal{C}}^\star) = \{\lambda M : \lambda \ge 0\}$ is a ray, then both $R$ and $Q^\star - Q_{\mathcal{C}}^\star$ must be non-negative multiples of the same direction $M$ for the normal-cone alignment condition to hold. $\qquad\square$

### H.5 PROOF OF COROLLARY 5

*Proof of Corollary 5.* We work in the impulse-space as defined in Section 3, i.e., a Hilbert space $(\mathcal{H}, \langle \cdot, \cdot\rangle_{\mathcal{H}})$ of causal LTI filters with matrix impulse response $\{q_k\}_{k \ge 0}$ with a Hilbert norm $\|Q\|_{\mathcal{H}}^2 = \sum_{k=0}^\infty \mathrm{Tr}(q_k^\top q_k)$, where $k$ is the time index.

The 1-pole momentum family forms a feasible submanifold within the trust region constraint $\mathcal{Q} = \{Q : \|Q\|_{\mathcal{H}} \le \sqrt{B}\}$ and the causality cone $\mathcal{C}$. By Lemma 4, the optimal momentum parameters correspond to the projection of the unconstrained optimizer $Q^\star$ onto this submanifold, which reduces to maximizing the inner product $\langle Q_{\mathrm{SGD+M}}, R\rangle_{\mathcal{H}}$ subject to the trust region constraint.

*Step 1 — Norm of the 1-pole optimizer.* The impulse response is $q_k = \eta P(1 - \beta)\beta^k$. By definition,

$$\|Q_{\mathrm{SGD+M}}\|_{\mathcal{H}}^2 = \sum_{k \ge 0} \mathrm{Tr}(q_k^\top q_k) = \sum_{k \ge 0} \mathrm{Tr}\left([\eta P(1 - \beta)\beta^k]^\top [\eta P(1 - \beta)\beta^k]\right) \tag{169}$$

$$= \eta^2(1 - \beta)^2 \sum_{k \ge 0} \beta^{2k} \mathrm{Tr}(P^\top P) = \eta^2(1 - \beta)^2 \frac{1}{1 - \beta^2} \mathrm{Tr}(P^\top P). \tag{170}$$

The trust region constraint $\|Q_{\mathrm{SGD+M}}\|_{\mathcal{H}} \le \sqrt{B}$ imposes

$$\eta \le \sqrt{B}\left(\mathrm{Tr}(P^\top P)\frac{(1 - \beta)^2}{1 - \beta^2}\right)^{-1/2}. \tag{171}$$

*Step 2 — Alignment with the moment operator.* The inner product with $R$ is

$$\langle Q_{\text{SGD+M}}, R\rangle_{\mathcal{H}} = \sum_{k\geq 0} \text{Tr}(q_k^\top R_k) = \eta(1-\beta)\sum_{k\geq 0} \beta^k \,\text{Tr}(P^\top R_k) \tag{172}$$

$$= \eta(1-\beta)\sum_{k\geq 0} \beta^k T_k, \tag{173}$$

where $T_k := \text{Tr}(P^\top R_k)$.

*Step 3 — Reduce to 1-D search; saturate trust region.* For fixed $\beta$, the inner product is linear in $\eta$ while the constraint is quadratic, so the maximizer saturates the trust region. The trust region-normalized gain is

$$J(\beta) := \frac{\langle Q_{\text{SGD+M}}, R\rangle_{\mathcal{H}}}{\|Q_{\text{SGD+M}}\|_{\mathcal{H}}} = \frac{\sqrt{1-\beta^2}}{\sqrt{\text{Tr}(P^\top P)}}\sum_{k\geq 0}\beta^k T_k. \tag{174}$$

Hence $\beta^\star = \arg\max_{0<\beta<1} J(\beta)$ and $\eta^\star$ saturates the trust region constraint.

*Step 4 — Specific case of parameter-independent SGD+Momentum.* For the specific case of parameter-independent SGD+Momentum where $P = I$, and therefore the impulse response is $q_k = \eta(1-\beta)\beta^k I$, we have $T_k = \text{Tr}(R_k)$, and the gain function is

$$J(\beta) = \frac{\sqrt{1-\beta^2}}{\sqrt{\text{Tr}(I^\top I)}}\sum_{k\geq 0}\beta^k \,\text{Tr}(R_k) = \frac{\sqrt{1-\beta^2}}{\sqrt{d}}\sum_{k\geq 0}\beta^k \,\text{Tr}(R_k), \tag{175}$$

where $d$ is the dimension of the parameter space, which is a constant. Therefore, we have the optimal momentum $\beta^\star$ and corresponding learning rate $\eta^\star$ as

$$\beta^\star = \arg\max_{0<\beta<1}\sqrt{1-\beta^2}\sum_{n=0}^\infty T[n]\beta^n, \qquad \eta^\star = \frac{\sqrt{B\left(1-\beta^{\star\,2}\right)}}{(1-\beta^\star)\sqrt{d}}. \tag{176}$$

*Step 5 — Solving J for streaming gradients.* Let $m[k] = g[k] + \beta g[k-1] + \beta^2 g[k-2] + \cdots$ be the unnormalized momentum at time $k$. This can be obtained from a sequential filtering process:

$$m[k] = g[k] + \beta m[k-1], \quad m[0] = 0, \tag{177}$$

which is exactly the same as how typical autograd frameworks implement momentum. Then, from $R_k[n] = \mathbb{E}[g[n]\,g[n-k]^\top]$ at specific time $n$ and time-interval $k$, we have $T_k[n] = \text{Tr}(R_k[n])$ and

$$J(\beta; n) = \sqrt{1-\beta^2}\sum_{k=0}^\infty T_k[n]\beta^k \tag{178}$$

$$= \sqrt{1-\beta^2}\sum_{k=0}^\infty \text{Tr}(\mathbb{E}[g[n]g[n-k]^\top])\beta^k \tag{179}$$

$$= \sqrt{1-\beta^2}\sum_{k=0}^\infty \text{Tr}(\mathbb{E}[g[n](\beta^k g[n-k]^\top)]) \tag{180}$$

$$= \sqrt{1-\beta^2}\,\text{Tr}\left(\mathbb{E}\left[g[n]\sum_{k=0}^\infty \beta^k g[n-k]^\top\right]\right) \tag{181}$$

$$= \sqrt{1-\beta^2}\,\text{Tr}(\mathbb{E}[g[n]\,m[n]^\top]) \tag{182}$$

$$= \sqrt{1-\beta^2}\,\mathbb{E}[\text{Tr}(g[n]\,m[n]^\top)] \tag{183}$$

$$= \sqrt{1-\beta^2}\,\mathbb{E}[g[n]^\top m[n]]. \tag{184}$$

This is due to the linearity of expectation, trace, summation, and inner product.

Therefore, we can rewrite the optimal momentum as

$$\beta^\star[n] = \arg\max_{0<\beta<1} J(\beta; n) = \arg\max_{0<\beta<1}\sqrt{1-\beta^2}\,\mathbb{E}[g[n]^\top m_\beta[n]], \tag{185}$$

where $m_\beta[n] = \sum_{k=0}^\infty \beta^k g[n-k]$ is the unnormalized momentum at time $n$ with momentum parameter $\beta$. This completes the proof. Note that, in theory, the expectation is taken over the entire possible gradient sequence $g[n]$, which should be approximated in the real-world application. $\square$

### H.6 PROOF OF COROLLARY 6

*Proof of Corollary 6.* We work in the impulse-space as defined in Section 3, i.e., a Hilbert space $(\mathcal{H}, \langle \cdot, \cdot \rangle_{\mathcal{H}})$ of causal LTI filters with matrix impulse response $\{q_k\}_{k \geq 0}$ with a Hilbert norm $\|Q\|_{\mathcal{H}}^2 = \sum_{k=0}^{\infty} \mathrm{Tr}(q_k^\top q_k)$, where $k$ is the time index.

The diagonal 1-pole Adam family forms a feasible submanifold within the diagonal trust region constraint $\mathcal{Q}_D$ and the causality cone $\mathcal{C}$. By Lemma 4, the optimal Adam parameters correspond to the projection of the unconstrained optimizer $Q^\star$ onto this submanifold, which reduces to maximizing the inner product $\langle Q_{\mathrm{Adam}}, R \rangle_{\mathcal{H}}$ subject to the diagonal trust region constraint.

*Step 1 — Norm of the diagonal optimizer.* First, for each parameter coordinate $j$, we have the running second moment $v_{\beta_2}$ and the coordinate-wise cost $c_j(\beta_2; n)$:

$$v_{\beta_2,j}[n] = (1 - \beta_2) \sum_{k=0}^{\infty} \beta_2^k g_j^2[n-k], \quad c_j(\beta_2; n) = v_{\beta_2,j}^{1/2}[n]. \tag{186}$$

From the definition of the Adam-family cone $\mathcal{C}_{1p}(c)$, the per-coordinate impulse response is:

$$q_{j,k} = \eta(1 - \beta_1)\beta_1^k / c_j. \tag{187}$$

Therefore, by the definition of the Hilbert norm, we have the norm of the diagonal optimizer:

$$\|Q_{\mathrm{Adam}}\|_{\mathcal{H}}^2 = \sum_{j=1}^{P} c_j \sum_{k \geq 0} |q_{j,k}|^2 = \eta^2 \frac{(1 - \beta_1)^2}{1 - \beta_1^2} \sum_{j=1}^{P} \frac{1}{c_j(\beta_2; n)} = \eta^2 \frac{(1 - \beta_1)^2}{1 - \beta_1^2} W(\beta_2; n), \tag{188}$$

where $W(\beta_2; n) := \sum_j 1/c_j(\beta_2; n)$ is the normalization factor.

*Step 2 — Alignment with the moment operator.* The inner product with $R_k[n] = \mathrm{diag}(r_{j,k}[n]) = \mathrm{diag}(\mathbb{E}[g_j[n]g_j[n-k]])$ is

$$\langle Q_{\mathrm{Adam}}, R \rangle_{\mathcal{H}} = \sum_{k \geq 0} \mathrm{Tr}(q_k^\top R_k) = \sum_{k \geq 0} \sum_{j=1}^{P} q_{j,k}\, r_{j,k} \tag{189}$$

$$= \eta(1 - \beta_1) \sum_{k \geq 0} \beta_1^k \sum_{j=1}^{P} \frac{r_{j,k}[n]}{c_j(\beta_2; n)} \tag{190}$$

$$= \eta(1 - \beta_1) \sum_{k \geq 0} \beta_1^k T_k(\beta_2; n), \tag{191}$$

where $T_k(\beta_2; n) := \sum_j r_{j,k}[n]/c_j(\beta_2; n)$ is the weighted trace of the moment operator.

*Step 3 — Reduce to 1-D search; saturate trust region.* For fixed $(\beta_1, \beta_2)$, the inner product is linear in $\eta$ while the constraint is quadratic, so the maximizer saturates the trust region. The trust region-normalized gain is

$$J(\beta_1, \beta_2) := \frac{\langle Q_{\mathrm{Adam}}, R \rangle_{\mathcal{H}}}{\|Q_{\mathrm{Adam}}\|_{\mathcal{H}}} = \frac{\sqrt{1 - \beta_1^2}}{\sqrt{W(\beta_2)}} \sum_{k \geq 0} \beta_1^k T_k(\beta_2). \tag{192}$$

Therefore, we have the optimal Adam hyperparameters as

$$(\beta_1^\star, \beta_2^\star) = \arg \max_{0 < (\beta_1, \beta_2) < 1} J(\beta_1, \beta_2), \tag{193}$$

and the corresponding learning rate is

$$\eta^\star = \sqrt{B}\, a(\beta_1^\star, \beta_2^\star), \tag{194}$$

where $a(\beta_1, \beta_2) := \sqrt{(1 - \beta_1^2)/W(\beta_2)}/(1 - \beta_1)$ is the normalization factor. This is from equation 188 above and the diagonal trust region definition:

$$\mathcal{Q}_D(B, c) := \{\mathrm{diag}(q_j) \succeq 0 : \sum_j c_j \sum_{k \geq 0} |q_j[k]|^2 \leq B\}, \tag{195}$$

*Step 4 — Solving J for streaming gradients.* The diagonal moment is $R_k[n] = \mathrm{diag}(\mathbb{E}[g_j[n]g_j[n-k]])$ at specific time $n$ and time-interval $k$. Therefore, we have $r_j[k;n] = \mathbb{E}[g_j[n]g_j[n-k]]$ and

$$T_k(\beta_2; n) = \sum_{j=1}^{P} \frac{r_{j,k}[n]}{c_j(\beta_2; n)} = \sum_{j=1}^{P} \mathbb{E}\left[\frac{g_j[n]g_j[n-k]}{c_j(\beta_2; n)}\right]. \tag{196}$$

$$J(\beta_1, \beta_2; n) = \frac{\sqrt{1-\beta_1^2}}{\sqrt{W(\beta_2)}} \sum_{k=0}^{\infty} \beta_1^k T_k(\beta_2; n) \tag{197}$$

$$= \frac{\sqrt{1-\beta_1^2}}{\sqrt{W(\beta_2)}} \sum_{k=0}^{\infty} \beta_1^k \sum_{j=1}^{P} \mathbb{E}\left[\frac{g_j[n]g_j[n-k]}{c_j(\beta_2; n)}\right] \tag{198}$$

$$= \frac{\sqrt{1-\beta_1^2}}{\sqrt{W(\beta_2)}} \sum_{j=1}^{P} \mathbb{E}\left[\frac{g_j[n]}{c_j(\beta_2; n)} \sum_{k=0}^{\infty} \beta_1^k g_j[n-k]\right] \tag{199}$$

$$= \frac{\sqrt{1-\beta_1^2}}{(1-\beta_1)\sqrt{W(\beta_2)}} \sum_{j=1}^{P} \mathbb{E}\left[g_j[n] \frac{m_{\beta_1,j}[n]}{c_j(\beta_2; n)}\right] \tag{200}$$

$$= \frac{\sqrt{1-\beta_1^2}}{(1-\beta_1)\sqrt{W(\beta_2)}} \mathbb{E}\left[\sum_{j=1}^{P} g_j[n] \frac{m_{\beta_1,j}[n]}{v_{\beta_2,j}^{1/2}[n]}\right] \tag{201}$$

$$= \frac{\sqrt{1-\beta_1^2}}{(1-\beta_1)\sqrt{W(\beta_2)}} \mathbb{E}[g[n]^\top u_{\beta_1,\beta_2}[n]], \tag{202}$$

where

$$m_{\beta_1,j}[k] = (1-\beta_1)\sum_{i=0}^{\infty} \beta_1^i g_j[k-i] = (1-\beta_1)m_{\beta_1,j}[k-1] + \beta_1 g_j[k], \tag{203}$$

$$v_{\beta_2,j}[k] = (1-\beta_2)\sum_{i=0}^{\infty} \beta_2^i g_j^2[k-i] = (1-\beta_2)v_{\beta_2,j}[k-1] + \beta_2 g_j^2[k], \tag{204}$$

are the unnormalized momentum and second moment for coordinate $j$ at time $k$, and

$$u_{\beta_1,\beta_2}[n] = \frac{m_{\beta_1,j}[n]}{v_{\beta_2,j}^{1/2}[n]} \tag{205}$$

is the unnormalized velocity at time $n$ of the original Adam optimizer. The equation above is due to the linearity of expectation, summation, and inner product. Therefore, we can rewrite the optimal momentum as

$$(\beta_1^\star[n], \beta_2^\star[n]) = \arg\max_{0<\beta_1<1, 0<\beta_2<1} \frac{\sqrt{1-\beta_1^2}}{(1-\beta_1)\sqrt{W(\beta_2; n)}} \mathbb{E}[g[n]^\top u_{\beta_1,\beta_2}[n]]. \tag{206}$$

This completes the proof. Note that, in theory, the expectation is taken over the entire possible gradient sequence $g[n]$, which should be approximated in the real-world application. □

### H.7 PROOF OF PROPOSITION 7

*Proof of Proposition 7.* We prove this by showing that commutativity of the optimizer $Q$ and the parameter Gram matrix $J^\top J$ forces the endpoint map to collapse into the canonical pseudoinverse operator.

*Step 1 — Equilibrium condition.* At equilibrium $\theta = \theta^\infty$, we have $\dot{\theta} = -QJ^\top(J\theta^\infty - y) = 0$. Since $Q \succeq 0$, this gives us $J^\top(J\theta^\infty - y) \in \ker(Q)$. Therefore, we get the endpoint map:

$$\theta^\infty = QJ^\top(JQJ^\top)^{-1}y, \tag{207}$$

where we interpret $(\cdot)^{-1}$ as the Moore-Penrose pseudoinverse when singular.

*Step 2 — Compact SVD of the Gram matrix.* Assume data space has higher dimension than parameter space, i.e., $n > d$. Take the compact SVD $J = U\Sigma V^\top$ with rank $r \leq \min(n, d)$, where

$$\Sigma = [\Sigma_1 \quad 0], \quad \Sigma_1 = \text{diag}(\sigma_1, \ldots, \sigma_r), \quad \sigma_i > 0, \tag{208}$$

with $U \in \mathbb{R}^{n \times n}$ and $V \in \mathbb{R}^{d \times d}$ orthogonal, and $\Sigma_1$ is a diagonal matrix with strictly positive entries. Then the parameter Gram matrix can be written as

$$J^\top J = V \begin{bmatrix} \Sigma_1^2 & 0 \\ 0 & 0 \end{bmatrix} V^\top. \tag{209}$$

The commutativity hypothesis $QJ^\top J = J^\top JQ$ is equivalent to saying that $Q$ and $J^\top J$ share the same eigenspace decomposition. That is, in the $V$-basis, $Q$ has the same block-diagonal structure as $J^\top J$:

$$Q = V \begin{bmatrix} Q_r & 0 \\ 0 & Q_0 \end{bmatrix} V^\top, \quad [Q_r, \Sigma_1^2] = 0. \tag{210}$$

Since $\Sigma_1$ is diagonal with strictly positive entries, commuting with $\Sigma_1^2$ implies commuting with $\Sigma_1$ itself: $[Q_r, \Sigma_1] = 0$.

*Step 3 — Compute $JQJ^\top$.*

$$JQJ^\top = U\Sigma V^\top QV\Sigma^\top U^\top \tag{211}$$

$$= U\Sigma \begin{bmatrix} Q_r & 0 \\ 0 & Q_0 \end{bmatrix} \Sigma^\top U^\top \tag{212}$$

$$= U(\Sigma_1 Q_r \Sigma_1)U^\top. \tag{213}$$

Since $\Sigma_1$ and $Q_r$ are invertible on the $r$-block:

$$(\Sigma_1 Q_r \Sigma_1)^{-1} = \Sigma_1^{-1} Q_r^{-1} \Sigma_1^{-1}. \tag{214}$$

*Step 4 — Compute the endpoint map.*

$$QJ^\top(JQJ^\top)^{-1} = V \begin{bmatrix} Q_r & 0 \\ 0 & Q_0 \end{bmatrix} V^\top \cdot V\Sigma^\top U^\top \cdot U(\Sigma_1 Q_r \Sigma_1)^{-1} U^\top \tag{215}$$

$$= V \begin{bmatrix} Q_r & 0 \\ 0 & Q_0 \end{bmatrix} \Sigma^\top (\Sigma_1^{-1} Q_r^{-1} \Sigma_1^{-1}) U^\top \tag{216}$$

$$= V \begin{bmatrix} Q_r \Sigma_1 \\ 0 \end{bmatrix} \Sigma_1^{-1} Q_r^{-1} \Sigma_1^{-1} U^\top \tag{217}$$

$$= V \begin{bmatrix} \Sigma_1^{-1} \\ 0 \end{bmatrix} U^\top. \tag{218}$$

*Step 5 — Compare with canonical pseudoinverse.*

$$J^\top(JJ^\top)^{-1} = V\Sigma^\top U^\top \cdot U(\Sigma_1^2)^{-1} U^\top \tag{219}$$

$$= V \begin{bmatrix} \Sigma_1 \\ 0 \end{bmatrix} \Sigma_1^{-2} U^\top \tag{220}$$

$$= V \begin{bmatrix} \Sigma_1^{-1} \\ 0 \end{bmatrix} U^\top. \tag{221}$$

Therefore, $QJ^\top(JQJ^\top)^{-1} = J^\top(JJ^\top)^{-1} = J^\dagger$, and the convergence endpoint is $\theta^\infty = J^\dagger y = \theta^\star$, the minimum norm solution. We can do the exact same proof for the case where the parameter space has higher dimension than data space, i.e., $d > n$, starting from Step 2. This is omitted here for brevity. $\qquad \square$

## H.8 PROOF OF LEMMA 8

For the proof of Lemma 8, we use the following lemma that formalizes the concept of *alignment* between the optimal optimizer and the gradient statistics as the sharing of common eigenstructures. The results are trivial from Theorem 1 but we state them for completeness.

**Lemma 22** (Greedy optimizers are gradient-aligned). *Let $R \succeq 0$ be a PSD moment matrix with eigendecomposition $R = U \operatorname{diag}(\sigma_1 \geq \cdots \geq \sigma_d)U^\top$. For each of the four families of trust regions defined in Section 2, the corresponding optimal optimizers $Q^\star$ that maximize $\operatorname{Tr}(QR)$ are aligned with the gradient statistics $R$ in the following manner:*

*(i) (Frobenius ball): $Q_F^\star = \sqrt{B}\, R/\|R\|_F$ shares eigenvectors with $R$ and has eigenvalues proportional to those of $R$.*

*(ii) (Spectral): $Q_S^\star = U \operatorname{diag}(q_i^\star)U^\top$ where $q_i^\star = \lambda$ for $i \leq k$, $q_{k+1}^\star = \tau - k\lambda$, and $q_i^\star = 0$ for $i > k+1$ with $k = \lfloor \tau/\lambda \rfloor$. The optimizer concentrates on the top eigenspace of $R$.*

*(iii) (Lyapunov): $Q_L^\star = \alpha \Pi_R$ where $\Pi_R$ is the orthogonal projection onto the support of $R$, and $\alpha = \sqrt{B}\,(\sum_{i:\sigma_i>0}\sigma_i)^{-1/2}$. The optimizer aligns with the support of the gradient moment.*

*(iv) (Diagonal): $[Q_D^\star]_{jj} \propto \sigma_j^2/c_j$ where $\sigma_j$ are the diagonal elements of $R$, and $U = I$. The optimizer weights are proportional to the squared gradient variances.*

*In all cases, the optimal optimizer $Q^\star$ shares the same eigenvector structure as the gradient moment $R$, which we casually call that the optimizer is aligned with the gradient moment.*

*Proof of Lemma 22.* We prove each case by applying the optimality conditions found in Theorem 1.

*Case (i): Frobenius ball.* The two matrices $Q_F^\star$ and $R$ are related by a scalar multiple $\sqrt{B}/\|R\|_F$.

*Case (ii): Spectral.* The two matrices $Q_S^\star$ and $R$ share the same eigenvector structure $U$.

*Case (iii): Lyapunov.* The two matrices $Q_L^\star$ and $\Pi_R$ are related by a scalar multiple $\alpha = \sqrt{B}(\sum_{i:\sigma_i>0}\sigma_i)^{-1/2}$.

*Case (iv): Diagonal.* The two matrices $Q_D^\star$ and $R$ are diagonal matrices and therefore share the same eigenvector structure $I$.

In all cases, the optimal $Q^\star$ *shares the same eigenvector structure* as the gradient moment $R$, which we casually call that the optimizer is *aligned* with the gradient moment. $\square$

Now we prove the commutativity of the optimal optimizer and the gradient moment in Lemma 8 of the main text.

*Proof of Lemma 8.* We prove commutativity for each family from Lemma 22.

*Case (i): Frobenius ball.* From Lemma 22, $Q_F^\star = \sqrt{B}\, R/\|R\|_F$. Since $Q_F^\star$ is a scalar multiple of $R$, they trivially commute:

$$Q_F^\star R = \frac{\sqrt{B}}{\|R\|_F}R^2 = R \cdot \frac{\sqrt{B}}{\|R\|_F}R = RQ_F^\star. \tag{222}$$

*Case (ii): Spectral.* Let $R = U \operatorname{diag}(\sigma_i)U^\top$ and $Q_S^\star = U \operatorname{diag}(q_i^\star)U^\top$. Since both matrices share the same eigenvector matrix $U$, they commute:

$$Q_S^\star R = U \operatorname{diag}(q_i^\star)U^\top U \operatorname{diag}(\sigma_i)U^\top = U \operatorname{diag}(q_i^\star \sigma_i)U^\top \tag{223}$$

$$= U \operatorname{diag}(\sigma_i q_i^\star)U^\top = U \operatorname{diag}(\sigma_i)U^\top U \operatorname{diag}(q_i^\star)U^\top = RQ_S^\star. \tag{224}$$

*Case (iii): Lyapunov.* We have $Q_L^\star = \alpha \Pi_R$ where $\Pi_R$ is the orthogonal projection onto the support of $R$. Since $\Pi_R$ commutes with any matrix that has the same null space (which includes $R$), we have:

$$Q_L^\star R = \alpha \Pi_R R = \alpha R \Pi_R = RQ_L^\star. \tag{225}$$

The second equality follows because $\Pi_R R = R$ (projection onto support) and $R\Pi_R = R$ (since $R$ maps into its own support).

*Case (iv): Diagonal.* Both $Q_D^\star = \operatorname{diag}(q_j^\star)$ and $R = \operatorname{diag}(\sigma_j)$ are diagonal matrices. Diagonal matrices always commute:

$$Q_D^\star R = \operatorname{diag}(q_j^\star \sigma_j) = \operatorname{diag}(\sigma_j q_j^\star) = RQ_D^\star. \tag{226}$$

In all four cases, the optimal optimizer $Q^\star$ *commutes* with the gradient moment matrix $R$. $\square$

## H.9   Proof of Theorem 9

*Proof of Theorem 9.* We establish the convergence endpoint through the minimum-norm characterization in $\mathcal{H}_{K_{Q^\star}}$.

*Step 1 — Function-space dynamics.* With squared loss $\mathcal{L}(\theta) = \frac{1}{2}\|f(X;\theta) - y\|^2$, the preconditioned parameter flow is

$$\dot{\theta}_t = -Q^\star \nabla_\theta \mathcal{L}(\theta_t) = -Q^\star J^\top (f_t(X) - y). \tag{227}$$

Under the NTK window (holding $J$ fixed), the function-space dynamics become

$$\dot{f}_t(\cdot) = -J(\cdot)\dot{\theta}_t = -J(\cdot)Q^\star J^\top (f_t(X) - y) = -K_{Q^\star}(\cdot, X)(f_t(X) - y). \tag{228}$$

where $K_{Q^\star}(\cdot, X) = J(\cdot)Q^\star J^\top$ is the optimizer-augmented tangent kernel.

*Step 2 — Convergence on training inputs.* Let $u_t = f_t(X) \in \mathbb{R}^n$. Then $\dot{u}_t = -K_{Q^\star}(u_t - y)$ where $K_{Q^\star} = JQ^\star J^\top$.

This linear ODE has solution $u_t = y - e^{-K_{Q^\star}t}(y - u_0)$.

Taking the spectral decomposition $K_{Q^\star} = U\Lambda U^\top$ with $\Lambda = \text{diag}(\lambda_i \geq 0)$, we have

$$e^{-K_{Q^\star}t} = U\text{diag}(e^{-\lambda_i t})U^\top \xrightarrow{t\to\infty} U\text{diag}(\mathbf{1}_{\{\lambda_i=0\}})U^\top = P_{\ker K_{Q^\star}}.$$

$P_{\ker K_{Q^\star}}$ is the projection onto the kernel of $K_{Q^\star}$.

Therefore, $u_\infty = K_{Q^\star}K_{Q^\star}^\dagger y + P_{\ker K_{Q^\star}}u_0$.

If $u_0 \in \ker K_{Q^\star}$ (e.g., small initialization), then $u_\infty = K_{Q^\star}K_{Q^\star}^\dagger y$.

*Step 3 — Minimum-norm characterization in $\mathcal{H}_{K_{Q^\star}}$.* By the Representer Theorem, any $f \in \mathcal{H}_{K_{Q^\star}}$ has the form

$$f(\cdot) = K_{Q^\star}(\cdot, X)\alpha, \qquad \|f\|^2_{\mathcal{H}_{K_{Q^\star}}} = \alpha^\top K_{Q^\star}\alpha,$$

and satisfies $f(X) = K_{Q^\star}\alpha$.

Among all interpolants $f$ with $f(X) = y \in \text{range}(K_{Q^\star})$, the minimum RKHS norm is obtained by minimizing $\alpha^\top K_{Q^\star}\alpha$ subject to $K_{Q^\star}\alpha = y$. The minimal-norm coefficient is $\alpha^\star = K_{Q^\star}^\dagger y$, yielding

$$f^\infty(\cdot) = K_{Q^\star}(\cdot, X)K_{Q^\star}^\dagger y, \qquad \|f^\infty\|^2_{\mathcal{H}_{K_{Q^\star}}} = y^\top K_{Q^\star}^\dagger y,$$

which coincides with the limit found in Step 2.

*Step 4 — Regularized case.* With ridge regularization $\lambda > 0$, the unique minimizer of $\frac{1}{2}\|f(X) - y\|^2 + \frac{\lambda}{2}\|f\|^2_{\mathcal{H}_{K_{Q^\star}}}$ is

$$f_\lambda^\infty(\cdot) = K_{Q^\star}(\cdot, X)(K_{Q^\star} + \lambda I)^{-1}y,$$

which also equals the steady state of gradient flow with weight decay $\lambda$.

*Step 5 — Commutation refinement for optimal optimizers.* For the Frobenius, spectral, data-metric, and diagonal families from Corollary 2, the optimal optimizer $Q^\star$ satisfies a key commutativity property with the Gram matrix $G = J^\top J$:

- Frobenius: $Q_F^\star \propto \Sigma = J^\top \Sigma_s J$, so $[Q_F^\star, G] = 0$

- Spectral: $Q_S^\star = U\,\text{diag}(q_i^\star)U^\top$ where $U$ diagonalizes $G$

- Lyapunov: $Q_L^\star = \alpha\,\Pi_\Sigma$ where $\Pi_\Sigma$ projects onto the support of $\Sigma = J^\top \Sigma_s J$

- Diagonal: $Q_D^\star$ is diagonal when $G$ is diagonal

This commutativity $[Q^\star, G] = 0$ implies $[K_{Q^\star}, K_I] = 0$ where $K_I = JJ^\top$ is the standard NTK. Hence $K_{Q^\star}$ and $K_I$ are simultaneously diagonalizable, and the kernel quantities separate mode-wise, enabling the water-filling closed forms under the trust region. □

