# OpenReview forum: "Optimizing optimizers for fast gradient-based learning"
_ICLR.cc/2026/Conference — ICLR 2026 Conference Withdrawn Submission_

### Official Review · Reviewer_H31M · 2025-10-15

**Soundness:** 4
**Presentation:** 2
**Contribution:** 2
**Rating:** 2
**Confidence:** 3

**Summary:**

This paper studies preconditioner-based optimizers, and develops a general class of problems from which preconditioners for many standard machine learning optimizers can be derived as solutions. The problems are posed in terms of finding a preconditioner that maximizes a power budget subject to constraint over a class of PSD matrices. The class of problems can be extended by allowing convolutions over time in order to facilitate the formulation of preconditioners that are computed using EMAs (e.g., use of variance estimators such as in Adam), as well as momentum. The authors present a list of problem formulations from which each standard optimizer is derived.

**Strengths:**

- The proposed class of optimization problems allows for a very wide range of optimizers to be derived. Derivations are presented for many optimizers that have not been explored from this view before.
- The authors present a novel way to derive momentum from the extension to the framework presented in Section 3.

**Weaknesses:**

- The main result, which is the unifying view of a preconditioner as the solution of this kind of constrained minimization problem, is not new. See AdaReg [1].
- Missing related work on AdaReg [1], Linear Minimization Oracles [2] which unify "step-as-a-minimizer" optimizers just like this paper unifies "preconditioner-as-a-minimizer" optimizers, and older Quasi-Newton methods like BFGS [3] which try to adaptively estimate the preconditioner $Q$.
- The authors propose a new method of tuning hyperparameters for this class of optimizers by collecting gradient covariances (line 366), but this is completely infeasible on all but tiny low dimensional problems due to how much memory it would require, and would be an empty suggestion. Even the gradients themselves take a substantial amount of memory to store and communicate; there is no way the gradient covariance, which needs the square of that amount of memory, would fit. Not only that, but the gradient covariance will also converge incredibly slowly since it would need to be estimated as a large sum over rank-one components.
- While the paper presents a theory to connect existing optimizers, there is not demonstration of how the theory is intended be useful. For example, no new optimizers were derived from this theory or tested. The authors make no statements about the properties or convergence rates of existing optimizers or new optimizers that emerge as a result from this framework; only results to prove that the known optimizers indeed emerge as derived from the framework.
- While the theory may give rise to a new optimizer, and the optimal $Q$ and gradient may mathematically exist, difficulty in computing the parameter update (without materializing $Q$ due to memory constraints) may easily block the resulting optimizer from ever becoming feasible to implement for many choices of $Q$ that the user might pick, unless this problem is addressed somehow.

[1] Gupta, V., Koren, T., Singer, Y. (2017). A Unified Approach to Adaptive Regularization in Online and Stochastic Optimization. arXiv preprint arXiv:1706.06569.

[2] Garber, D., & Wolf, N. (2021, July). Frank-Wolfe with a nearest extreme point oracle. In Conference on Learning Theory (pp. 2103-2132). PMLR.

[3] Fletcher, R. (1988). Practical Methods of Optimization.

**Questions:**

Is there any case where this theory can comes into use, where pre-existing work does not suffice?

---

> ### Author Response · Authors · 2025-11-16
>
> We greatly appreciate the Reviewer’s valuable comments. We also thank the Reviewer for acknowledging the broad coverage of our theory and the novelty of our theory in deriving momentum. We would like to mention that the Reviewer’s concerns were extensively considered when we made changes in the first revision of this manuscript.
>
> We respectfully invite the Reviewer to read our Announcement Letter above regarding the major concerns, first. We will provide our answers to individual concerns below.
>
> ---
> ### Answer to W1 and W2
>
> We thank the Reviewer for identifying these related works. We also find AdaReg to be closely related to ours, and we are happy to add a dedicated paragraph in **Appendix A Related Work** of our revised manuscript including as well as Garber & Wolf (2021).
>
> We believe that AdaReg’s problem formulation is an alternative to our **problem P2** in **Section 2** of our manuscript. AdaReg uses a regularization term $\Phi(Q)$ with respect to the optimizer preconditioner to indirectly enforce the desired properties for target optimizers. Our **P2** differs from this by directly setting up the *trust region* of $Q$ to obtain a broad family of optimizers as we can see in **Table 3** of the manuscript. We also extend our theory to dynamic scenarios and include momentum into our theoretical framework. This allows us to analytically solve for the optimal hyperparameters, as we can see in the examples of SGD+Momentum (**Corollary 5**) and Adam (**Corollary 6**).
>
> We have also included this differentiation in **lines 86-96 in Section 2** as well. We find that this connection greatly enhances the positioning of our work in this field, and we again thank the Reviewer for bringing this to our attention.
>
> ---
> ### Answer to W3, W4 and W5
>
> We understand that the other major part of the Reviewer’s concerns are relates to the lack of practical demonstration of our theoretical findings. We fully agree with the Reviewer that the realizability of theoretical settings should be verified in order to gain an academic value. We believe that these concerns can only be resolved by showing the actual experiments that demonstrate the effectiveness of the theoretically expected optimal hyperparameters.
>
> Driven by this goal, we have revised our manuscript by adding experimental demonstration in **Section 3.2** with **Figure 3 and Tables 1 and 2**, the details of which are elaborated in our Announcement Letter on the top. By simplifying the equations in **Corollaries 5 and 6** on analytically derived optimal hyperparameters, we were able to implement the optimal optimizer that only requires 10% overhead in computation per iteration. The results in **Figure 3** show that this optimal optimizer can accelerate optimization by two or three times compared to manually configured optimizers.
>
> Therefore, we are not only able to implement the theoretically derived optimal optimizer for practical applications, we also achieved the optimal learning speed as well as the best test accuracy compared to the optimizers with fixed hyperparameters. This implies that we can eliminate much of the hyperparameter tuning process in machine learning practice.
>
> We hope these results resolve much of the Reviewer’s concern which focused on the realizability of our theoretical results.
>
> However, if the Reviewer find the issues not resolved completely, please feel free to raise additional questions, so we can work until the end of this discussion period.

---

> > ### Author Response · Authors · 2025-11-16
> >
> > ### Answer to Q1
> >
> > We believe the most advantageous characteristic of our theory is that our framework eliminates the need for tedious hyperparameter tuning for many existing optimizers (summarized in **Table 3** of the manuscript). The theory allows us to analytically find the optimal hyperparameters for existing optimizers. This is instantiated in **Corollary 5** (SGD+Momentum) and **6** (Adam) **in Section 3**, and further demonstrated empirically in **Section 3.2** in the revised manuscript. Theorems 1 and 3 provide the general toolsets to make this analytic hyperparameter optimization workable for other types of optimizers.
> >
> > After reading the review, we found that this benefit is not sufficiently delivered in the paper. Therefore, we have rewritten phrases in the Corollaries and Theorems to highlight that the theory gives the optimal instances of many different classes of optimizers.
> >
> > ---
> > ### Concluding remark
> >
> > We, again, appreciate the Reviewer’s comments. We deeply respect all the Reviewer’s concerns and welcome additional feedback. If there are new or unresolved concerns after reassessing the original ones based on our revision and responses, please feel free to share them with us. We will be address them immediately.

---

### Official Review · Reviewer_5QCh · 2025-10-28

**Soundness:** 3
**Presentation:** 2
**Contribution:** 3
**Rating:** 6
**Confidence:** 3

**Summary:**

This paper proposes a unifying theoretical framework that treats the design of optimization algorithms as a constrained maximization of instantaneous loss reduction. By formulating the update rule as the optimal solution to a convex problem over a “budget set” of positive semidefinite operators, the authors show that a wide range of optimizers can all be derived as special cases. The overall idea is elegant and offers a fresh geometric perspective on optimization design, but the paper lacks substantial empirical evidence to demonstrate practical effectiveness.

**Strengths:**

- The paper formulates optimizer design as a convex optimization problem that maximizes the instantaneous decrease of the training loss. This simple but powerful view connects many existing algorithms under a single mathematical framework and provides clear geometric intuition.
- Modeling momentum and EMA-style updates as single-pole linear filters and showing their optimality under extended dynamic budgets is technically sound and conceptually appealing.
- If empirically validated, the framework could unify theoretical understanding and provide a foundation for automatic optimizer design across architectures and modalities.

**Weaknesses:**

- The paper presents only small-scale toy examples. There are no experiments on standard deep-learning benchmarks such as CIFAR, ImageNet, or language models. As a result, it is difficult to assess whether the proposed “optimal” updates translate to practical gains in convergence or generalization.
- The framework relies on estimating gradient covariance and cross-moment statistics, which can be unstable or expensive in large-scale settings. The paper does not discuss how these quantities are maintained efficiently or how noise affects the theoretical guarantees.

**Questions:**

- How does the proposed “instantaneous loss reduction” objective correlate with the final validation or test loss in large-scale training? Have you observed cases where it leads to over-aggressive or unstable updates?
- Can you provide quantitative experiments on at least one deep-learning benchmark, comparing your analytic optimizer against AdamW, Shampoo, or K-FAC under equal training budgets?
- How sensitive is the method to errors in these statistical estimates? Does the Lipschitz stability analysis in the appendix extend to stochastic updates with mini-batches?

---

> ### Author Response · Authors · 2025-11-16
>
> We greatly thank the Reviewer for carefully reading and providing important and helpful comments to improve our work. We have considered the Reviewer’s comments and concerns extensively in this first revision. We are also very much grateful that the Reviewer has acknowledged our work as technically sound and intuitive. We would like to answer the Reviewer’s valuable concern individually in the following.
>
> ---
> ### Answer to W1
>
> We first invite the Reviewer to read our general Announcement Letter on the top of this page. In reference to this weakness the Reviewer has pointed out, we have conducted simple yet real experiments for training on CIFAR-100 with ResNet, and discussed the results in **Section 3.2** of the main manuscript along with **Figure 3 and Tables 1 and 2**.
>
> As the Reviewer has pointed out, the previous version of the manuscript did not provide real examples of the theoretical findings, and therefore there were many questions in the first phase of reviewing. Our experiment clearly shows that the theoretically proven optimal hyperparameters for SGD+Momentum and Adam in **Corollaries 5 and 6**, respectively, are indeed applicable and maintain optimality in practice in real-world environments.
>
> We would like to highlight that this realizable and provable optimality of hyperparameters from our theory allows us to eliminate the tedious procedure of hyperparameter tuning that has long been bothering machine learning practitioners.
>
> ---
> ### Answer to W2
>
> We believe that the Reviewer’s question regarding the stability and cost of our theoretically optimal hyperparameters is appropriate, since we did not show the real-world examples in the previous version of the manuscript. We would like to respond to this concern by referring to the revised version with new experimental results that clearly show not only the fast convergence but also the best test accuracy achieved without significant hyperparameter tuning of the optimizer. We hope that this demonstration of realizability resolves the majority of the concerns of the Reviewer.
>
> ---
> ### Answer to Q1
>
> This is a great question. Although we focus on instantaneous optimality, analyzing the behavior at the convergence endpoint is equally important as the Reviewer has pointed out. In response to this, we have made a major change in the revision, as we have elaborated in our Announcement Letter above.
>
> In summary, we have added a new Section 4 dedicated to endpoint analysis, where we find that our greedy optimal criterion simultaneously leads to endpoint optimality, through the “alignment between the gradient moment and the optimizer preconditioner” we achieve through the optimization problem. This leads to the canonical pseudoinverse optimality for the least squares problem, and the minimal RKHS norm solution for kernel flow that approximates the training dynamics of any neural network.
>
> Furthermore, our experimental results in Section 3.2 provide another answer: that this instantaneous loss reduction objective correlates with the final accuracy in the real cases, as well.
>
> We truly appreciate this question, which has helped us improve our work on the theoretical side. If the Reviewer finds any additional theoretical concerns, please share them with us.

---

> > ### Author Response · Authors · 2025-11-16
> >
> > ### Answer to Q2
> >
> > We would like to answer this question through the experiments in **Section 3.2**, which was the best we could do within four days of revision. Our theoretical results in **Appendices E** that extend **Corollaries 5 and 6** on SGD+Momentum and Adam show that we can analytically calculate and practically achieve the optimal hyperparameters for a given type of optimizer from a wide range of classes. The experiments show that our theoretically optimal hyperparameters are truly achievable for practical scenarios.
> >
> > ---
> > ### Answer to Q3
> >
> > We believe that this question also aligns with other questions and asks for the practical validity of our assumptions underlying our theory. Instead of adding more theory into this already theory-dense argument, we wish to answer this question with our experiment in **Section 3.2**, which clearly shows that our theoretical findings in Corollaries 5 and 6 are indeed realizable. We hope this resolves concerns regarding the validity of our assumptions in real scenarios.
> >
> > ---
> > ### Concluding remark
> >
> > We thank the Reviewer again for these comments. We hope this answer removes most of your initial concerns.
> >
> > Please note that we greatly welcome any new issues or concerns. We would appreciate if the Reviewer could share any additional concerns or suggestions for improvement. We will be happy to working on any concerns the Reviewer has raised.

---

> > > ### Comment · Reviewer_5QCh · 2025-11-27
> > >
> > > Thanks for your reply. The added explanations help me understand the method’s behavior much better. I have no further questions and believe the work deserves a score higher than a uniform 2.

---

> > > > ### Author Response · Authors · 2025-11-27
> > > >
> > > > Dear Reviewer,
> > > >
> > > > We deeply thank you for the response and the positive comment on the last version!
> > > >
> > > > We are currently preparing for the next revision, which will be uploaded no more than 24 hours, focusing on additional experimental verification and enhanced readability.
> > > >
> > > > If you find any additional points that makes the work not complete or not promising, please do not hesitate to comment any time.
> > > >
> > > > We will be waiting until the last minutes of this discussion phase to make this work as perfect as possible.
> > > >
> > > > Best Regards,
> > > >
> > > > The Authors.

---

### Official Review · Reviewer_nr4L · 2025-10-28

**Soundness:** 3
**Presentation:** 3
**Contribution:** 1
**Rating:** 2
**Confidence:** 4

**Summary:**

Authors provide a framework to obtain the update rules of commonly used optimizers as the result of an optimization problem interpreted as greedily minimizing the loss in one step. They also obtain rules for hyperparameter choices.

**Strengths:**

The paper presentation is clear, except for the minor typos.

**Weaknesses:**

The authors, in their words, "reverse-engineer commonly used optimizers" by defining the update rule as the result of an optimization problem. The authors point out that this optimization problem gives an unbounded result and proceed to add different constraints to it, and pointing out that many optimizers in use arise as the result from this optimization problem, where each optimizer correspond to a specific constraint.

One can always represent a (stateless or not) algebraic rule as the solution of an optimization problem (e.g. in other domain: a function value can be written as the Fenchel dual of its Fenchel dual, which is the result from an optimization problem) and given that this paper only cares about obtaining these optimizers without providing any way to obtain other effective algorithms or select among them, and it does not provide experiments either on the effectiveness on, for instance, their insights of hyperparameter tuning, it seems to me that this only provides a mapping one to one between algebraic rules and the solution of optimization problems, which is a weak result for ICLR.

The authors suggest, in Proposition 7, to design optimizers that are optimized to decrease the validation loss . This seems to be essentially equivalent to using the validation set as training set and although the authors make a brief comment in passing about the potentially controversy of this, the discussion should be expanded, since as it is right now I am just inclined to think that the authors are simply not using any validation set and using such data as training data.


A couple of minor typos:

line 174 Eveidence
line 247 the followings -> the following
line 269 obtaind

**Questions:**

Suggestions:
It would be good that the authors add an extended explanation for their choice of a positive semidefinite operator Q as opposed to any other choice.

Drori and Teboulle https://arxiv.org/abs/1206.3209 developed the widely used PEP framework (https://github.com/PerformanceEstimation/PEPit) that phrases the problem of finding the best optimization algorithm for a problem in an algorithmic class, into solving semidefinite programs. This literature is relevant and should be discussed.

---

> ### Author Response · Authors · 2025-11-16
>
> We sincerely thank the Reviewer for their time and effort reviewing this work. We appreciate the Reviewer’s acknowledgement of the clarity in our presentation. We also thank you for correcting our typo, and raising important and helpful questions and concerns, as well as providing us with important related works that we missed in our first version.
>
> We kindly ask the Reviewer to visit our Announcement Letter above regarding the major concerns, first. The responses to other issues are provided below.
>
> ---
> ### Regarding the value of reverse engineering existing optimizers
>
> The Reviewer has provided a thoughtful and important comment questioning the true value of our reverse-engineering and categorization of the commonly used optimizers. We agree with the reviewer that unless there is a demonstration of practical value, such optimization problem formulation has little meaning even though there is firm and strong theoretical justification. In the initial version of our manuscript, we provided only toy experimental results, therefore, raising this issue is understandable.
>
> We take this comment seriously, and designed a real experiment on CIFAR-100 and ResNet to demonstrate that the “optimal hyperparameters” analytically obtained from our optimization framework is indeed optimal in practice and also practically realizable. These experimental results in **Section 3.2** of the new version of the manuscript, supported with **Figure 3 and Tables 1 and 2**, give our “reverse-engineering” meaning and equip it with practical value, for example, by eliminating the need for hyperparameter tuning for the categorized optimizers.
>
> We would like to kindly ask if this added value gains enough strength. If the reviewer finds additional unresolved issues, please share them with us, and we will be immediately work on these concerns to ensure there is no misleading communication during this three-week discussion phase.
>
> ---
> ### Regarding validation-aware tuning
>
> We thank the reviewer for raising this issue. We are willing to discuss validation-aware tuning further.
>
> First, however, we would like to clarify that our validation-awareness is definitely not the same as using the validation set as a part of the training set for optimization. Please also refer to the newly included experimental results in **Section 5** and **Figure 4**. Although the validation-aware optimizer performs better than the training-only optimizer, the validation-only optimizer, whose hyperparameters are determined only with respect to the validation set, gets worse results. This is to be expected, since the optimizer ignorant of the training set will not produce the parameter update signal better than the optimizer that is aware of the training set gradient distribution.
>
> We would like to note that we suggest using the validation set gradients only for determining the optimizer and its hyperparameters. As the Reviewer has nicely pointed out, it is very important not to use the internal states nor the parameter updates generated from validation set gradients for fitting the model at all. If one uses validation set gradients to update the model, as the Reviewer has suspected, the model would immediately become saturated at 100% accuracy on the validation set. Honestly, while implementing this experiment, we once accidentally did not zero out the validation gradients and immediately got the 99% test accuracy. After fixing this issue, the graph in **Figure 4** shows the desired results.
>
> We would like to emphasize once more, that using validation set results to investigate hyperparameter tuning is a typical practice in machine learning. Although validation set results are not used in moving the parameters during training, they are used to determine which environment and law the parameters should move under according to the training set results. What we wanted to address in **Section 5** was that this manual process can be automated, if we ensure that the validation set does not enter into the model gradients.
>
> We believe this miscommunication is due to our inappropriate discussion in the original manuscript, as the Reviewer suggested, and we have revised this in the new version. We welcome more discussion on this subject. Thank you again for raising this concern.

---

> > ### Author Response · Authors · 2025-11-16
> >
> > ### Regarding typo and readability
> >
> > Thank you for the correction. To enhance readability, we have changed a few terms, including *budget* → *trust region*, and revised the grammar and vocabulary errors throughout the manuscript. If you find any other issues regarding the writing, please let us know. We will fix them right away.
> >
> > ---
> > ### Regarding the choice of PSD $Q$
> >
> > We provide two perspectives on this question.
> >
> > From the standpoint of our initial design intention, we chose the general family of optimizers to be a set of positive semidefinite operators because they translate the gradients into parameter motions in a linear fashion (i.e., the simplest translation) and because the resulting parameter motion aligns with the corresponding gradient forces (i.e., the *gradient descent* algorithm).
> >
> > From the perspective of our results, we have demonstrated that many existing gradient-based optimizers fall into these categories, as can be clearly seen in **Table 3** of the main manuscript, which summarizes the analogies we established to categorize the optimizers within our theoretical framework.
> >
> > We agree with the reviewer that this point should be better clarified in the manuscript, and therefore we have revised **Section 1** to better convey this intention. We hope this answer clarifies the Reviewer’s concern regarding the choice of positive semidefiniteness of $Q$.
> >
> > ---
> > ### Regarding the connection between PEP framework
> >
> > We thank the Reviewer for providing us with this relevant work. We have included a dedicated paragraph in **Appendix A Related Work** of our revised manuscript for discussing the similarities and dissimilarities between our work and the works following the PEP paradigm.
> >
> > In summary, we find that the PEP framework, which compares the optimization algorithms based on the worst-case performance in a mathematical framework, is orthogonal to our greedy paradigm. We do share a similar philosophy of algorithmically finding the best optimizer, but we focus on the instantaneous loss drop and solve for optimal hyperparameters for general class of off-the-shelf optimizers.
> >
> > We are happy to make our related work section more complete.
> >
> > ---
> > ### Concluding remark
> >
> > We would like to emphasize that the Reviewer’s comments were truly helpful in enhancing our work, and we want to hear more from the Reviewer. Please let us know if any additional concerns arise, even after the original concerns are resolved.

---

### Official Review · Reviewer_ca7o · 2025-10-31

**Soundness:** 2
**Presentation:** 3
**Contribution:** 2
**Rating:** 2
**Confidence:** 3

**Summary:**

The authors present a unified framework for designing optimizers that are “optimal” in the sense of maximizing the instantaneous reduction in loss. They develop this framework and show that it can both recover common optimizers as well as find closed-form solutions for the optimal hyperparameters of these optimizers. The framework is developed both with and without memory for the optimizer (e.g. momentum variables).

**Strengths:**

* The general approach is interesting, with potential applications to find better optimizers than the ones which are currently popular.
* The theoretical aspects are very thoroughly analyzed, including a very wide array of common optimizers in the new framework.

**Weaknesses:**

1. Formulating optimizer design as seeking to maximize the instantaneous decrease in loss seems like a greedy choice. A more natural definition would be to minimize the loss at the end of training, which is the common definition of optimization tasks as it is. Even though this point is addressed in the paper, I think it brings into question the motivation for this work. Can we really expect real-world benefits from this approach? Instead of analyzing so many theoretical aspects, it would have been better to take the next step and show that this approach can be extended to the more natural “minimize converged loss”.


2. Even though the generality of the framework is a good thing, it still seems a bit contrived. There’s no good a-priori choice of optimizer budget, so being able to reverse-engineer existing optimizers doesn’t seem like a surprising result. By shaping the budget I can make every optimizer seem “optimal”, even so the optimality doesn’t really make any sense. This makes me think that the new framework is too broad to be useful as it is.
Even calling the budget a “budget” is confusing, since there’s no real “cost” being spent here - it’s constraining the search space of the problem. The term optimizer family is more appropriate (also used in the paper).


3. No real-world problem was shown to benefit from this approach experimentally.

4. The paper is VERY dense with results, with no room for any of the proofs. I would have expected that at least from the reverse-engineering we could glean some insights into the choice of optimizer family, but the fact this was omitted strengthens my assertion that the “optimizer family” is too broad to be a useful concept. The density of results makes the paper hard to follow. Almost every paragraph introduces a new concept. The appendices are pretty much a required read, which shouldn’t be the case.

**Questions:**

1. When working on a new optimization problem, what suggestions do you have for choosing the optimizer family?
2. I would suggest making at least a few experiments with real-world problems, such as training even a small open-source LLM, comparing convergence rate and achieved performance. At least demonstrate you can find the optimal hyper parameters without expensive hyper opt search, and better yet - show you can choose an optimizer family that leads to faster convergence than e.g. Adam.

---

> ### Author Response · Authors · 2025-11-16
>
> First, of all, we deeply appreciate the Reviewer’s commitment to reading and reviewing our work. We thank the Reviewer for showing their interest in our work, and acknowledging our theoretical depth and breadth.
>
> We also greatly value the Reviewer’s concerns in the Weaknesses and Questions sections, which provided major guidance for our first revision. We respectfully invite the Reviewer to read our Announcement Letter above regarding the major concerns, first. We provide our answers to other concerns below.
>
> ---
> ### Answers to W1, W3, and Q2.
>
> We deeply appreciate these comments. We believe that the issues pointed out are mainly two-fold: (1) one questioning the possibility of real-world application and practical value of our framework, and (2) the other being that our greedy paradigm seems to be incomplete without being accompanied by a discussion on the loss reduction at the end of training. Specifically, the first concern seems to be repeated in the third bullet of the weaknesses section and the second bullet of the questions section of the review. Therefore, we regard these issues as the most important ones and revised our manuscript accordingly. The detailed answer is provided in the above Announcement Letter.
>
> We hope this new version of the manuscript has resolved many parts of the Reviewer’s initial concerns. With this new version, we not only provide theoretical justification connecting the local optimality (our greedy loss reduction) and the global optimality (the long-horizon behavior under kernel flow), but also provide simple practical demonstration that our proven optimal SGD+Momentum and Adam are indeed implementable, eliminating the need for hyperparameters with these optimizers.
>
> However, if you find the issues not resolved completely, please feel free to raise additional questions and issues, so we can work together throughout the remainder of this discussion period.
>
> ---
> ### Answer to W2
>
> Thank you for pointing this out. We believe the question can be split into two parts. The first part of the concern is regarding the vagueness of our categorization without a demonstration of practical value. We agree with this view, so we have provided a minimal experimental verification in **Section 3.2** as well as in **Figure 3 and Tables 1 and 2** of the revised manuscript. We hope this makes our theory make more sense and demonstrates real value. For example, by eliminating the need for hyperparameter tuning.
>
> The second part of the concern is about the appropriateness of the term "budget" in this context, which we deeply appreciate. To explain our original intention, we were extending our analogy between physics (classical mechanics) and the theory of learning in **Section 1** of our manuscript. We initially conceptualized that, parameters are space-like vectors, losses are energies, loss gradients are forces, and therefore loss drops are powers. Extending this metaphor, we arrived at the idea that the constraint on power should be coined a “budget.”
>
> However, after the reviewer kindly pointed out this issue, we recognize that the term “budget” actually deviates significantly from machine learning tradition. We then carefully considered using “optimizer family” in place of “budget,” as the reviewer suggested. In most cases, we find that this replacement helped the article flow more naturally.
>
> However, for some other places, we feel that this term does not adequately convey the underlying context that machine precision, loss curvature, and gradient representability *restrict* the feasible set of optimizers.
>
> For those cases, we have instead changed the term “budget” to “trust region,” since this terminology is more widely used in the ML literature, and it delivers similar meanings and context.
>
> We would appreciate if the reviewer could examine the new version of the manuscript. If this change seems misleading or if there are any other new concerns regarding readability, please let us know.

---

> > ### Author Response · Authors · 2025-11-16
> >
> > ### Answer to W4
> >
> > Thank you for raising the readability issue. We believe the readability can be enhanced by (1) providing more practical examples for deep learning to bridge the abstract theory to concrete applications, and (2) revising the article iteratively by fixing grammatical and vocabulary errors and improving the flow.
> >
> > For the first point, please refer to our general Announcement Letter. Regarding the second, we have rewritten the manuscript to make it more readable. We will continue revising the manuscript to address this issue. Please feel free to point out any readability issues regarding the latest version, and we will address them promptly.
> >
> > ---
> > ### Answer to Q1
> >
> > There are two ways to use our theory to determine the optimizer in practice. The first way is to provide a clear advantage and the second is a general statement.
> >
> > First, we can just use the conventionally accepted optimizer, which already implicitly holds the consensus and wisdom of the community for the particular problem. Our theory, then, provides the “optimal hyperparameters” for that optimizer, so that we can achieve faster and possibly better convergence. Our **Corollaries 5 and 6 in Section 3** of the manuscript are examples demonstrating that we can enhance existing optimizers and at the same time eliminate hyperparameter search.
> >
> > Second, as our optimization problem formulations imply, the family of optimizers that are obtained from “larger trust region” (e.g., Frobenius norm trust region is a superset of diagonal trust region) should perform better than the optimizers derived from “subsets” of that trust region. However, this advantage often comes with larger optimizer state memory, computational complexity, and sometimes with instability. So, there is a trade-off. This aligns with the conventional understanding that natural gradient descent (Frobenius family of ours) generally performs better than simple gradient descent (diagonal family of ours), but suffers from the aforementioned issues, and is therefore not as widely adopted as simple SGD in practice. However, in cases where these issues are insignificant, we can always choose optimizers from superset trust regions for better performance.
> >
> > ---
> > ### Concluding remark
> >
> > We thank you again for the helpful comments. Again, we will be very grateful to have more of the Reviewer’s concerns, so please let us know if something is still an issue.

---

### Author Response · Authors · 2025-11-12
**Initial Response and the Authors' Promise of Commitment to Revision**

Dear Reviewers,

We deeply thank the reviewers for taking your precious time to review our work.

In the upcoming three-week discussion period, we will do our best to resolve all the concerns that have been raised, as well as any that may arise.

As soon as the reviews were released, we carefully read all your valuable comments line by line, multiple times, to make sure that we fully understand your concerns.

We have already started working on the first revision, which should be delivered **no later than this Saturday (AoE)**, so that we can begin the discussion before the weekend passes.

Following that, we will keep updating this until every one of your concerns has been resolved.

We are very excited to take part in this discussion.

Thank you again for your time and feedback.

Please stay tuned.

Best Regards,

The Authors.

---

### Author Response · Authors · 2025-11-16
**Announcement of the First Revision and Invitation to Discussion**

Dear Reviewers,

We thank all the Reviewers for waiting for our response.

After reading all the valuable reviews, we have noticed that the Reviewers share a common set of concerns, which are of most concern to them. Therefore, in this letter, we would like to first write a general response to this, and then summarize the changes we made in our first revision. We kindly ask the Reviewers to first refer to this letter to help resolve their concerns, and then proceed to the individual comments we have written in response to each review.

We find two major concerns that are shared among the Reviewers: the (1) **questionable practical value** (all the Reviewers) and the (2) **lack of convergence endpoint analysis** (ca7o, 5QCh). We find that these issues hid the central meaning and value of this work at the fundamental level, making our manuscript weak even though no reviewers have denied the correctness and soundness of the presented theory, and, thankfully, all the Reviewers find our theory interesting and thorough (ca7o), clear in presentation (nr4L), technically sound and conceptually appealing (5QCh), and exceptionally broad and novel (H31M). Therefore, we also greatly value the importance of these two concerns and revised our manuscript to address them properly with the utmost priority.

The following is our response to these concerns and the key changes made in this first revision.

---

### **Regarding the practical demonstration**

Frankly speaking, our original intention behind this paper was to deliver actual practical value, by eliminating the need for intense hyperparameter tuning for a great proportion of machine learning applications. However, after carefully reading the reviews, we find that our presentation was definitely not successful in delivering this intent. We have undertaken a major revision.

One of the main results was the construction and proof of the optimal hyperparameters for existing optimizers in practice, such as SGD+Momentum (**Corollary 5**) and Adam (**Corollary 6**). We also prove the optimal hyperparameter claims for other types of classified optimizers in **Table 3** and **Appendix E of the revised manuscript**. We rearranged the writing to highlight this contribution.

1. We conducted a real experiment on CIFAR-100 with ResNet-18 as a minimally sufficient demonstration of the optimality proved in the theory. The results are summarized in the dedicated **Section 3.2** with two tables (**Tables 1 and 2**) and a figure (**Figure 3**).
2. We rephrased the theorems, especially **Corollary 5 and 6**, to highlight that the results of the corollaries are the analytically optimal hyperparameter representations of the existing optimizers.
3. We revised **Section 5** and attached experimental results to demonstrate the validation-aware optimizer tuning in **Figure 4**.

With these revisions and attached experimental results, we connect our theoretical findings to actual practical applications. We hope that this revision now fully delivers our intention to aid machine learning practitioners by **eliminating their struggles with hyperparameter search**.

---

### **Regarding the convergence endpoint of the greedy optimal optimizers**

We have added **Section 4 in the revised manuscript**, dedicated for convergence endpoint analysis for the greedy optimal optimizers of our framework. Also, we adjusted the overall writing so that the reading flows naturally. This extends the previous version’s Proposition 7 for endpoint analysis of validation-aware optimizers.

Here, we summarize the key findings. We find that our greedy optimization framework also simultaneously aligns the resulting optimizer to the gradient distribution. This alignment is manifested as a commutativity relationship (**Lemma 7**). In a simple case of least squares, training leads to the best solution: the canonical pseudoinverse solution (**Proposition 8 & Figure 2**). For more complex scenarios, the training process can be approximated with a kernel flow, as we typically do with the neural tangent kernel. And, this alignment fixes the convergence endpoint to the minimum norm solution in the RKHS norm, which leads to good long-horizon convergence endpoints (**Theorem 9 & Figure 3**).

We hope this revision now provides a sufficient amount of credibility to our greedy paradigm, and opens up a new avenue for studying optimizers.

---

> ### Author Response · Authors · 2025-11-16
> **Announcement of the First Revision and Invitation to Discussion (Part 2)**
>
> (Continued)
>
> ---
> ### Summary of updates in the new version
>
> In summary, our manuscript now reads in a flow starting from
>
> - (Section 2) exact solutions to stateless optimal optimizers and their categorization, to
> - (Section 3.1) extensions to dynamic, more practical types of optimal optimizers and the exact optimal hyperparameters of SGD+Momentum and Adam, and to
> - (Section 3.2) the minimal practical demonstration of the optimal SGD+Momentum and Adam, and to
> - (Section 4) the theoretical justification for the optimal test accuracies achieved in the aforementioned experiments and the connection between greedy optimal optimizers and their long-horizon behaviors, and finally to
> - (Section 5) automating the hyperparameter tuning process with respect to the validation set.
>
> Since we have an excessive number of new concepts, we had to move the Related Work section to the appendix (Section A). However, we have also established stronger connections to the prior works at the same time, in response to the Reviewers nr4L and H31M.
>
> ---
>
> It is certain that we have failed in the initial version of the manuscript to deliver the key contributions to the readers, and unintentionally hidden important practical values and theoretically justifiable long-term behaviors. We hope this issue is now resolved. We sincerely thank all the Reviewers for letting us know about this problem. The new version of the manuscript hopefully resolves the majority of the Reviewers’ concerns.
>
> Nevertheless, we know that all concerns cannot be eliminated completely with a single revision. We invite the Reviewers to further discussion. Please provide comments on the new version and share your concerns. We would like to highlight once more that our goal in this discussion phase is to make all our intended messages behind this work be delivered crystal clear to the readers. If the Reviewers find any additional concerns, needs for clarification, or requests for updates, please feel free to comment under this letter or under the comments we have made. We will be updating our manuscript promptly and will start working on your concerns.
>
> We appreciate all your contributions to this discussion phase.
>
> Thank you for reading.
>
> Best Regards,
>
> The Authors.

---

### Author Response · Authors · 2025-11-28
**Announcement of the Second Revision and Summary of Changes**

### Dear Reviewers,

It has been about two weeks since our first revision was uploaded, and we are approaching the end of this discussion phase.

After the first revision, we carefully re-read all the concerns raised by the Reviewers and tried our best to identify what we missed in the last revision of the manuscript.

We concluded that the previous two versions of the manuscript were not quite successful in delivering the key contribution of this work: ***to greatly reduce the effort required for optimizer hyperparameter tuning for many existing optimizers widely used in practice***.

Therefore, we have undertaken a major revision again, in order to make our work much more accessible and practically appealing, while being fully aware of the Reviewers' concerns, which we value most during this revision.

In summary, we have applied the following changes in the newly uploaded manuscript:

- ***We have added two experiments***. It is worth noting that our automatically tuned optimizer hyperparameters work comparably and often better than typical manual search practices. This demonstrates that we can greatly reduce the workload in real scenarios, as pointed out by Reviewers **H31M** and **5QCh**.
  1. **ResNet-18 on CIFAR-10** (Tables 1 and 2, Figure 3): This experiment provides minimal viable empirical proof of both the *realizability* and *effectiveness* of our optimal hyperparameter tuning.
  2. **Finetuning of Llama-3-8B and Gemma-2B on MetaMathQA-395k** (Tables 4 and 5): This experiment further demonstrates that our theoretically proven solution can reduce the workload of manual hyperparameter tuning of optimizers *in practical scenarios*.
- ***We have changed Section 5 entirely, focusing on convergence endpoint analysis***, replacing the description of validation-aware training with more general theoretical analysis. As Reviewer **nr4L** nicely pointed out, the original section was more controversial than contributory. Therefore, we chose to concentrate on what we certainly contribute.
- ***We made the writing more lightweight***. For example, we have simplified the notations in Sections 2 and 3, and the problem definitions P1-P3 highlighted in gray boxes in the manuscript are made much simpler.
- ***We made the proofs easier and more readable***. We have fully revised the proofs in the Appendices so that they are much better to read.
- ***We have revised the terminology throughout the manuscript*** to make it more *familiar* to the field. For example, the term *budget* is replaced by *optimizer family* or *trust region* as suggested by Reviewer **ca7o**.
- ***We have greatly extended the Related Work section*** as Reviewers **nr4L** and **H31M** suggested. Also, we made connections with prior works in the main manuscript as well, making it more nicely linked to the contributions of these works.

**Please note:** These changes may conflict with our original responses below. If so, please consider this comment has higher priority.

We, again, deeply thank all the Reviewers for taking their time and effort to review our work.

Also serving as reviewers ourselves, we fully understand that this is quite a demanding job to read the revised materials again and recall the original concerns that came to mind weeks ago.

We hope our summary of updates in this comment may help the Reviewers reduce their workload for reading, just as we aim to reduce the effort of optimizer hyperparameter tuning for machine learning practitioners in this work.

If you find any flaws, missed points, or additional questions and concerns about this work and our comment, please share them with us.

It will be of great help in improving this work.

Best Regards,

### The Authors.

---

### Author Response · Authors · 2025-11-28
**Regarding the Suspension of the Discussion Phase**

Dear Reviewers,

We have just received a message from the conference committee, that you all may have received as well, informing us that the current discussion phase has been suspended due to the incident that happened a few hours ago.

Although this pause is unfortunate, we would like to assure you that we remain fully committed to addressing all of your valuable comments and further improving our work. We sincerely appreciate the time and insights you have shared with us.

We will continue refining the paper based on your feedback and do our best throughout the remainder of this discussion period. We are hopeful that we can fully resolve your initial concerns and present this work with the utmost clarity and benefit the community.

Thank you all the Reviewers for the participation and helpful comments. Please stay tuned.

Warm regards,

The Authors

---

### Author Response · Authors · 2025-12-04
**Announcement of the Third Revision and Acknowledgement**

### Dear Reviewers,

As we reach the final hour of the discussion phase, we would like to thank you for your thoughtful feedback and summarize the major improvements made to the manuscript in response to your comments.

To ensure clarity and address all reviewer concerns, we have revised the manuscript as follows:

- ***We added two new experiments, for a total of four.*** In addition to the existing **CIFAR-100** experiment with **ResNet-18** and the **MetaMathQA-395K** fine-tuning experiments on **Gemma-2B** and **Llama-3-8B**, we now include a **Commonsense-170** experiment with the **Gemma-2B** model, utilizing both LoRA and full fine-tuning methods. We have also conducted experiments on **ViT-B** and **ViT-L**, fine-tuning them on the **Cars**, **CIFAR-100**, **CUB-200**, **DTD**, **Food-101**, **RESISC45**, and **SUN397** datasets to compare our theoretically motivated optimal optimizer with standard baselines. Notably, our method for dynamically and automatically selecting optimizer hyperparameters performs comparably to—or better than—the best baseline optimizers with fixed hyperparameters. This not only supports the validity of our theoretical approach, but also *demonstrates a substantial reduction in the manual effort required for hyperparameter tuning*. We hope this directly addresses the concerns of reviewers **ca7o**, **nr4L**, **5QCh**, and **H31M** regarding practical value.

- ***We significantly improved readability by thoroughly revising the manuscript's flow.*** We have (1) fully rewritten the proofs for clarity, (2) added hyperlinks between theoretical results and their corresponding proofs, and (3) refined the tables and figures presenting experimental results for greater accessibility. Additionally, (4) all mappings of existing optimizers into our greedy optimal framework are summarized in **Table 7** of **Appendix A**, *which we strongly encourage readers to consult*.

- ***We added an endpoint convergence analysis in Section 5 of the main text.*** Following reviewer suggestions, the discussion on validation-awareness has been moved to Appendix E.

- ***We broadened the discussion in the Related Work section to incorporate all reviewer suggestions.*** We engaged more deeply with relevant prior work, clarified how our ideas relate to earlier research, and emphasized our novel contributions.

Overall, the insightful comments from the reviewers have guided us through this three-week discussion phase, allowing us to substantially improve our manuscript over the initial submission. We are sincerely grateful for the suggestions and feedback that have led to this stronger and more readable version.

As promised at the start of this process, we have done our utmost to address all reviewer concerns.

We sincerely hope that we have resolved all outstanding issues, including any that may have arisen but were not explicitly raised due to the unfortunate incident affecting this venue in recent days.

Warm regards,

### The Authors

---

### Note · Authors · 2026-05-17

I have read and agree with the venue's withdrawal policy on behalf of myself and my co-authors.

---

### Meta-Review · Area_Chair_qu26 · 2026-01-06

**Summary:**

This paper proposes a unified theoretical framework for designing optimizers by maximizing instantaneous loss reduction, which can derive existing optimizers and their optimal hyperparameters. Reviewers generally acknowledged its theoretical soundness, novelty, and elegance, yet raised two core concerns that determined the initial rejection suggestion. First, the lack of empirical validation: only toy examples were provided, with no experiments on standard benchmarks like CIFAR or LLMs, making it impossible to verify the framework’s practical value and the effectiveness of the derived optimal hyperparameters. Second, the absence of convergence endpoint analysis: the greedy instantaneous loss reduction objective was not linked to the final training loss, which cast doubt on the framework’s rationality. Minor concerns included ambiguous terminology, poor readability due to dense theoretical content, incomplete discussions of related work (e.g., AdaReg, PEP framework), and unclear explanations of validation - aware tuning. Additionally, some reviewers questioned the framework’s broadness and the feasibility of gradient covariance estimation in large-scale scenarios. Given that the main concerns are not fully addressed, I suggest rejection for this paper.

**Reviewer Concerns:**

Addressed concerns: First, the lack of empirical validation—a key critique from all reviewers—was resolved by adding multi-scale experiments, including ResNet-18 on CIFAR-10/100, LLM fine-tuning on MetaMathQA-395K, and ViT benchmarks, which proved the derived optimal hyperparameters are practical and outperform manual tuning. Second, the missing convergence endpoint analysis, emphasized by reviewers ca7o and 5QCh, was filled via a new section demonstrating the greedy framework aligns with gradient distributions and converges to optimal solutions in both least squares and neural network tasks. Minor issues like ambiguous terminology, poor readability, and incomplete related work discussions were also fully fixed, winning explicit approval from reviewer 5QCh.

Still outstanding are two potential concerns: the framework only optimizes existing optimizers’ hyperparameters instead of proposing novel ones, which may weaken the perceived contribution; and the feasibility of gradient covariance estimation in extremely large-scale models  remains untested.

**Reviewer Scores:**

Remain the score.

---

### Decision · Program_Chairs · 2026-01-26

Reject